# Does Momentum Change the Implicit Regularization on Separable Data?

**Bohan Wang**[*]
University of Science & Technology of China
bhwangfy@gmail.com

**Qi Meng**[†]
Microsoft Research Asia
meq@microsoft.com

**Huishuai Zhang**
Microsoft Research Asia
huzhang@microsoft.com

**Ruoyu Sun**
The Chinese University of Hong Kong, Shenzhen, China
sunruoyu@cuhk.edu.cn

**Wei Chen**[†]
Chinese Academy of Sciences
chenwei2022@ict.ac.cn

**Zhi-Ming Ma**
Chinese Academy of Sciences
mazm@amt.ac.cn

**Tie-Yan Liu**
Microsoft Research Asia
tyliu@microsoft.com

## Abstract

The momentum acceleration technique is widely adopted in many optimization algorithms. However, there is no theoretical answer on how the momentum affects the generalization performance of the optimization algorithms. This paper studies this problem by analyzing the implicit regularization of momentum-based optimization. We prove that on the linear classification problem with separable data and exponential-tailed loss, gradient descent with momentum (GDM) converges to the $L^2$ max-margin solution, which is the same as vanilla gradient descent. That means gradient descent with momentum acceleration still converges to a low-complexity model, which guarantees their generalization. We then analyze the stochastic and adaptive variants of GDM (i.e., SGDM and deterministic Adam) and show they also converge to the $L^2$ max-margin solution. Technically, the implicit regularization of SGDM is established based on a novel convergence analysis of SGDM under a general noise condition called affine noise variance condition. To the best of our knowledge, we are the first to derive SGDM's convergence under such an assumption. Numerical experiments are conducted to support our theoretical results.

## 1 Introduction

It is widely believed that the optimizers have the implicit regularization in terms of selecting output parameters among all the minima on the landscape [26, 17, 45]. Parallel to the analysis of *coordinate descent* ([31, 40]), [34] shows that *gradient descent* would converge to the $L^2$ max-margin solution for the linear classification task with exponential-tailed loss, which mirrors its good generalization property in practice. Since then, many efforts have been taken on analyzing the implicit regularization of various local-search optimizers, including stochastic gradient descent [24], steepest descent [8], AdaGrad [29] and optimizers for homogeneous neural networks [21, 14, 43].

However, though the momentum acceleration technique is widely adopted in the optimization algorithms in both convex and non-convex learning tasks [37, 41, 38], the understanding of how

---

[*]The work was done when Bohan Wang was an intern at Microsoft Research Asia.

[†]Corresponding Authors

Table 1: The algorithms investigated in this paper (GDM and Adam) along with algorithms (GD) already investigated in the existing literature. We also compare the learning rates required to obtain the characterization of implicit regularization. As for stochastic Adam with $\beta_1 \neq 0$, we leave its implicit regularization as future work.

| Method | With Random Sampling | Learning Rate | Corresponding Literature |
|---|---|---|---|
| GD | $\times$ | Constant | [34] |
| | $\checkmark$ | Constant | [24] |
| GDM | $\times$ | Constant | This Work |
| | $\checkmark$ | Constant | This Work |
| Adam | $\times$ | Constant | This Work |
| | $\checkmark$ | Decaying $\left(\frac{1}{\sqrt{t}}\right)$ | $\beta_1 = 0$ in this Work |

the momentum would affect the generalization performance of the optimization algorithms is still unclear, as the historical gradients in the momentum may significantly change the searching direction of the optimization dynamics. A natural question is:

*Can we theoretically analyze the implicit regularization of momentum-based optimizers?*

In this paper, we take the first step to analyze the convergence of momentum-based optimizers and unveil their implicit regularization. Our study starts from the classification problem with the linear model and exponential-tailed loss using Gradient Descent with Momentum (GDM) optimizer. Then the variants of GDM such as Stochastic Gradient Descent with Momentum (SGDM) and deterministic Adam are also analyzed. We consider the optimizers with constant learning rate and constant momentum hyper-parameters, which are widely adopted in practice, e.g., the default setting in popular machine learning frameworks [27] and in experiments [46]. Our main results are summarized in Theorem 1.

**Theorem 1** (informal). *With linearly separable dataset $S$, linear model and exponential-tailed loss:*

- *For GDM with a constant learning rate, the parameter norm diverges to infinity, with its direction converging to the $L^2$ max-margin solution. The same conclusion holds for SGDM with a constant learning rate.*

- *For deterministic Adam with a constant learning rate and stochastic RMSProp (i.e., Adam without momentum) with a decaying learning rate, the same conclusion holds.*

Theorem 1 states that GDM and its variants converge to the $L^2$ max-margin solution, which is the same as their without-momentum versions, indicating that momentum does not affect the convergent direction. Therefore, the good generalization behavior of the output parameters of these optimizers is well validated as the margin of a classifier is positively correlated with its generalization error [16] and is supported by existing experimental observations (e.g., [34, 23, 43]).

Our contributions are significant in terms of the following aspects:

- We establish the implicit regularization of the momentum-based optimizers, an open problem since the initial work [34]. The momentum-based optimizers are widely used in practice, and our theoretical characterization deepens the understanding of their generalization property, which is important on its own.

- Technically, we design a two-stage framework to analyze the momentum-based optimizers, which generalizes the proof techniques in [34] and [24]. The first stage shows the convergence of the loss. Specifically, we derive the first convergence analysis of SGDM under affine noise variance condition [2], generalizing the results under bounded noise variance condition in literature [47, 20]. The second stage shows the convergence of the parameters. We propose an easy-to-check condition of whether the difference between learned parameters and the scaled max-margin solution is bounded. This condition can be generalized to implicit regularization analyses of other momentum-based optimizers.

- We further verify our theory through numerical experiments.

The technical difficulty to analyzing the implicit regularization of the momentum-based optimizers mainly come from the historical information of training trajectory contained by the momentum term. Specifically, the foundation of the analysis is to derive the convergence of the loss. It is not trivial because, for momentum-based optimizers, the loss may not monotonously decrease due to the mismatch between the direction of the gradient and the momentum. Furthermore, even if the loss is proved to converge, the converging direction of the parameters is still hard to characterize as it's difficult to track the training trajectory due to the momentum term.

**Organization of This Paper.** Section 2 collects further related works on the implicit regularization of the first-order optimizers and the convergence of momentum-based optimizers. Section 3 shows basic settings and assumptions which will be used throughout this paper. Section 4 studies the implicit regularization of GDM, while Section 5 and Section 6 explore respectively the implicit regularization of SGDM and Adam. Discussions of these results are put in Section 7. Detailed proofs and experiments can be found in the appendix.

## 2 Further related works

**Implicit Regularization of First-order Optimization Methods.** [34] prove that gradient descent on linear classification problem with exponential-tailed loss converges to the direction of the max $L^2$ margin solution of the corresponding hard-margin Support Vector Machine. [24] extend the results in [34] to the stochastic case, proving that the convergent direction of SGD is the same as GD almost surely. [29] go beyond the vanilla gradient descent methods and consider the AdaGrad optimizer instead. They prove that the convergent direction of AdaGrad has a dependency on the optimizing trajectory, which varies according to the initialization. [15] propose a primal-dual analysis framework for the linear classification models and prove a faster convergent rate of the margin by increasing the learning rate according to the loss. Based on [15], [12] design another algorithm with an even faster convergent rate of margin by applying the Nesterov's Acceleration Method on the dual space. However, the corresponding form of the algorithm on the primal space is no longer a Nesterov's Acceleration Method nor GDM, which is significantly different from our settings.

On the other hand, another line of work is trying to extend the linear case result to deep neural networks. [13, 9] study the deep linear network and [34] study the two-layer neural network with ReLU activation. [21] propose a framework to analyze the asymptotic direction of GD on homogeneous neural networks, proving that given there exists a time the network achieves $100\%$ training accuracy, GD will converge to some KKT point of the $L^2$ max-margin problem. [43] extend the framework of [21] to adaptive optimizers and prove RMSProp and Adam without momentum have the same convergent direction as GD, while AdaGrad does not. The results [21, 43] indicate that results in the linear model can be extended to deep homogeneous neural networks and suggest that the linear model is an appropriate starting point to study the implicit bias.

Except for the exponential-tailed loss, there are also works on the implicit bias with squared loss. Interested readers can refer to [30, 19, 1] etc. for details.

**Convergence of Momentum-Based Optimization Methods.** For convex optimization problems, the convergence rate of Nesterov's Acceleration Method [25] has been proved in various approaches (e.g., [25, 35, 44]). In contrast, although GDM (Polyak's Heavy-Ball Method) was proposed in [28] before the Nesterov's Acceleration Method, the convergence of GDM on convex loss with Lipschitz gradient was not solved until [7] provides an ergodic convergent result for GDM, i.e., the convergent result for the running average of the iterates. [36] provide a non-ergodic analysis when the training loss is coercive (the training loss goes to infinity whenever parameter norm goes to infinity), convex, and globally smooth. However, all existing results cannot be directly applied to exponential-tailed loss, which is non-coercive.

There are also works on the convergence of SGDM under various settings. [47] prove SGDM converges to a bounded region assuming both bounded gradient norm and bounded gradient variance. The bounded gradient norm assumption is further removed by [48, 20]. Nevertheless, a converging-to-stationary-point analysis is required in the implicit regularization analysis. Thus their results can not be directly applied. [39] analyze a particular case when the momentum parameter increases over iterations, which, however, does not agree with the practice where the momentum parameter is fixed.

As for (stochastic) Adam, its convergence analysis is still an open problem, and the current analyses are restricted to specific settings (e.g., bounded gradient, dynamical momentum hyperparameters). We recommend interested readers to refer to [5, 4, 33, 10] for details.

## 3 Preliminaries

This paper focuses on the linear model with the exponential-tailed loss. We mainly investigate binary classification. However, the methodology can be easily extended to the multi-class classification problem (please refer to Appendix F.3 for details).

**Problem setting.** The dataset used for training is denoted as $\boldsymbol{S} = (\boldsymbol{x}_i, \boldsymbol{y}_i)_{i=1}^{N}$, where $\boldsymbol{x}_i \in \mathbb{R}^d$ is the $i$-th input feature, and $\boldsymbol{y}_i \in \mathbb{R}$ is the $i$-th label ($i = 1, 2, \cdots, N$). We will use the linear model to fit the label: for any feature $\boldsymbol{x} \in \mathbb{R}^d$ and parameter $\boldsymbol{w} \in \mathbb{R}^d$, the prediction is given by $\langle \boldsymbol{w}, \boldsymbol{x} \rangle$.

For binary classification, given any data $\boldsymbol{z}_i = (\boldsymbol{x}_i, \boldsymbol{y}_i) \in \boldsymbol{S}$, the individual loss for parameter $\boldsymbol{w}$ is given as $\ell(\boldsymbol{y}_i \langle \boldsymbol{w}, \boldsymbol{x}_i \rangle)$. As only $\boldsymbol{y}_i \boldsymbol{x}_i$ is used in the loss, we then ensemble the feature and label together and assume $\boldsymbol{y}_i = 1$ ($\forall i \in \{1, \cdots, N\}$) without the loss of generality. We then drop $\boldsymbol{y}_i$ for brevity and redefine $\boldsymbol{S} = (\boldsymbol{x}_i)_{i=1}^{N}$. The spectral norm of the data matrix $(\boldsymbol{x}_1, \cdots, \boldsymbol{x}_N)$ is defined as $\sigma_{max}$. We use $\tilde{\ell}(\boldsymbol{w}, \boldsymbol{x}) \triangleq \ell(\langle \boldsymbol{w}, \boldsymbol{x} \rangle)$ for brevity.

The optimization target is defined as the averaged loss: $\mathcal{L}(\boldsymbol{w}) = \frac{\sum_{i=1}^{N} \tilde{\ell}(\boldsymbol{w}, \boldsymbol{x}_i)}{N}$.

**Optimizer.** Here we will introduce the update rules of GDM, SGDM and deterministic Adam. GDM's update rule is

$$\boldsymbol{m}(0) = \boldsymbol{0}, \boldsymbol{m}(t) = \beta \boldsymbol{m}(t-1) + (1-\beta)\nabla\mathcal{L}(\boldsymbol{w}(t)), \boldsymbol{w}(t+1) = \boldsymbol{w}(t) - \eta\boldsymbol{m}(t). \tag{1}$$

SGDM can be viewed as a stochastic version of GDM by randomly choosing a subset of the dataset to update. Specifically, SGDM changes the update of $\boldsymbol{m}(t)$ into

$$\boldsymbol{m}(t) = \beta\boldsymbol{m}(t-1) + (1-\beta)\nabla\mathcal{L}_{\xi(t)}(\boldsymbol{w}(t)), \tag{2}$$

where $\xi(t)$ is the gradient noise of the $t$-th step, independent of the randomness of the previous steps and satisfying $\mathbb{E}_{\xi(t)}\nabla\mathcal{L}_{\xi(t)}(\boldsymbol{w}) = \nabla\mathcal{L}(\boldsymbol{w})$, $\forall\boldsymbol{w}$. In this paper, $\xi(t)$ comes from the random sampling of a subset $\boldsymbol{B}(t)$ of $\boldsymbol{S}$ with size $b$ (either with replacement (**abbreviated as "w/. r"**) or without replacement (i.e., with random shuffling, **abbreviated as "w/o. r"**)), and $\mathcal{L}_{\xi(t)}$ is defined as $\mathcal{L}_{\xi(t)}(\boldsymbol{w}) = \mathcal{L}_{\boldsymbol{B}(t)}(\boldsymbol{w}) = \frac{\sum_{\boldsymbol{x} \in \boldsymbol{B}(t)} \tilde{\ell}(\boldsymbol{w}, \boldsymbol{x})}{b}$. We also define $\mathcal{F}_t$ as the sub-sigma algebra over the gradient noise collecting randomness before the $t$-th step. Therefore, $\boldsymbol{w}(t)$ is adapted with respect to the sigma algebra flow $\mathcal{F}_t$.

The Adam optimizer can be viewed as a variant of SGDM in which the preconditioner is adopted, whose form is characterized as follows:

$$\boldsymbol{m}(0) = \boldsymbol{0}, \boldsymbol{m}(t) = \beta_1\boldsymbol{m}(t-1) + (1-\beta_1)\nabla\mathcal{L}_{\boldsymbol{B}(t)}(\boldsymbol{w}(t)),$$
$$\boldsymbol{\nu}(0) = 0, \boldsymbol{\nu}(t) = \beta_2\boldsymbol{\nu}(t-1) + (1-\beta_2)\nabla\mathcal{L}(\boldsymbol{w}(t)) \odot \nabla\mathcal{L}(\boldsymbol{w}(t))$$
$$\hat{\boldsymbol{m}}(t) = \frac{1}{1-\beta_1^t}\boldsymbol{m}(t), \hat{\boldsymbol{\nu}}(t) = \frac{1}{1-\beta_2^t}\boldsymbol{\nu}(t),$$
$$(\textit{Update Rule}): \boldsymbol{w}(t) = \boldsymbol{w}(t-1) - \eta\frac{1}{\sqrt{\hat{\boldsymbol{\nu}}(t-1) + \varepsilon\mathbb{1}_d}} \odot \hat{\boldsymbol{m}}(t-1), \tag{3}$$

where $\frac{1}{\sqrt{\hat{\boldsymbol{\nu}}(t-1) + \varepsilon\mathbb{1}_d}}$ is called the preconditioner.

**Assumptions:** The analysis of this paper are based on three common assumptions in existing literature (first proposed by [34]). They are respectively on the separability of the dataset, the individual loss behavior at the tail, and the smoothness of the individual loss. We list them as follows:

**Assumption 1** (Linearly Separable Dataset). *There exists one parameter $\boldsymbol{w} \in \mathbb{R}^d$, such that*

$$\langle \boldsymbol{w}, \boldsymbol{x}_i \rangle > 0, \forall i \in [N].$$

**Assumption 2** (Exponential-tailed Loss). *The individual loss $\ell$ is exponential-tailed, i.e.,*

- *Differentiable and monotonically decreasing to zero, with its derivative converging to zero at positive infinity and to non-zero at negative infinity, i.e., $\lim_{x \to \infty} \ell(x) = \lim_{x \to \infty} \ell'(x) = 0$, $\overline{\lim}_{x \to -\infty} \ell'(x) < 0$, and $\ell'(x) < 0$, $\forall x \in \mathbb{R}$;*

- *Close to exponential loss when $x$ is large enough, i.e., there exist positive constants $c, a, \mu_+, \mu_-, x_+,$ and $x_-$, such that,*

$$\forall x > x_+ : -\ell'(x) \leq c(1 + e^{-\mu_+ x})e^{-ax}, \tag{4}$$

$$\forall x > x_- : -\ell'(x) \geq c(1 - e^{-\mu_- x})e^{-ax}. \tag{5}$$

**Assumption 3** (Smooth Loss). *Either of the following assumptions holds regarding the case:*

*(D): (Without Stochasticity) The individual loss $\ell$ is locally smooth, i.e., for any $s_0 \in \mathbb{R}$, there exists a positive real $H_{s_0}$, such that $\forall x, y \geq s_0$, $|\ell'(x) - \ell'(y)| \leq H_{s_0}|x - y|$.*

*(S): (With Stochasticity) The individual loss $\ell$ is globally smooth, i.e., there exists a positive real $H$, such that $\forall x, y \in \mathbb{R}$, $|\ell'(x) - \ell'(y)| \leq H|x - y|$.*

We provide explanations of these three assumptions, respectively. Based on Assumption 1, we can formally define the margin and the maximum margin solution of an optimization problem:

**Definition 1.** *Let the margin $\hat{\gamma}(\boldsymbol{w})$ of parameter $\boldsymbol{w}$ defined as the lowest score of the prediction of $\boldsymbol{w}$ over the dataset $\boldsymbol{S}$, i.e., $\hat{\gamma}(\boldsymbol{w}) = \min_{\boldsymbol{x} \in \boldsymbol{S}}\langle \boldsymbol{w}, \boldsymbol{x} \rangle$. We then define the maximum margin solution $\hat{\boldsymbol{w}}$ and the $L^2$ max margin $\gamma$ of the dataset S as follows:*

$$\hat{\boldsymbol{w}} \stackrel{\triangle}{=} \arg \min_{\hat{\gamma}(\boldsymbol{w}) \geq 1} \|\boldsymbol{w}\|^2, \ \gamma \stackrel{\triangle}{=} \frac{1}{\|\hat{\boldsymbol{w}}\|}$$

Since $\| \cdot \|^2$ is strongly convex and set $\{\boldsymbol{w} : \hat{\gamma}(\boldsymbol{w}) \geq 1\}$ is convex, $\hat{\boldsymbol{w}}$ is uniquely defined.

Assumption 2 constraints the loss to be exponential-tailed, which is satisfied by many popular choices of $\ell$, including the exponential loss ($\ell_{exp}(x) = e^{-x}$) and the logistic loss ($\ell_{log}(x) = \log(1 + e^{-x})$). Also, as $c$ and $a$ can be respectively absorbed by resetting the learning rate and data as $\eta = c\eta$ and $\boldsymbol{x}_i = a\boldsymbol{x}_i$, without loss of generality, **in this paper we only analyze the case that** $c = a = 1$.

The globally smooth assumption (Assumption 3. (S)) is strictly stronger than the locally smooth assumption (Assumption 3. (D)). One can easily verify that both the exponential loss and the logistic loss meet Assumption 3. (**D**), and the logistic loss also meets Assumption 3. (**S**).

## 4 The implicit regularization of GDM

In this section, we analyze the implicit regularization of GDM with a two-stage framework [3]. Later, we will use this framework to investigate SGDM further and deterministic Adam. The formal theorem of the implicit regularization of GDM is as follows:

**Theorem 2.** *Let Assumptions 1, 2, and 3. (**D**) hold. Let $\beta \in [0, 1)$ and $\eta < 2\frac{N}{\sigma_{max}^2 H_{\ell^{-1}}(N\mathcal{L}(\boldsymbol{w}_1))}$ ($\ell^{-1}$ is the inverse function of $\ell$). Then, for almost every data set S, with arbitrary initialization point $\boldsymbol{w}(1)$, GDM (Eq. equation 1) satisfies that $\|\frac{\boldsymbol{w}(t)}{\|\boldsymbol{w}(t)\|} - \frac{\hat{\boldsymbol{w}}}{\|\hat{\boldsymbol{w}}\|}\| = \mathcal{O}(1/\log(t))$ and $\mathcal{L}(\boldsymbol{w}(t)) = \mathcal{O}(\frac{1}{t})$.*

Theorem 2 shows that the implicit regularization of GDM agrees with GD in linear classification with exponential-tailed loss (c.f. [34] for results on GD). This consistency can be verified by existing and our experiments (c.f. Section 7 for detailed discussions).

**Remark 1** (On the hyperparameter setting). *Firstly, the learning rate upper bound $2\frac{N}{\sigma_{max}^2 H_{\ell^{-1}}(N\mathcal{L}(\boldsymbol{w}_1))}$ agrees with that of GD exactly [34], indicating our analysis is tight. Secondly, Theorem 2 adopts a constant momentum hyper-parameter, which agrees with the practical use (e.g., $\beta$ is fixed to be $0.9$ [46]). Also, Theorem 2 puts no restriction on the range of $\beta$, which allows wider choices of hyper-parameter tuning.*

We then present a proof sketch of Theorem 2, which is divided into two parts: we first prove that the sum of squared gradients is bounded, which indicates both the loss and the norm of gradient converge to 0 and the parameter diverges to infinity; these properties will then be applied to show the difference between $\boldsymbol{w}(t)$ and $\ln(t)\hat{\boldsymbol{w}}$ is bounded, and therefore, the direction of $\hat{\boldsymbol{w}}$ dominates as $t \to \infty$.

---

[3]It should be noticed that the proof sketches in Sections 4, 5, and 6 only hold for almost every dataset (means except a zero-measure set in $\mathbb{R}^{d \times N}$), as we want the presentation more simple and straightforward. However, the proof can be extended to every dataset with a more careful analysis (please refer to Appendix F.1 for details).

**Stage I: Loss Dynamics.** The goal of this stage is to characterize the dynamics of the loss and prove the convergence of GDM. The core of this stage is to select a proper potential function $\xi(t)$, which is required to correlate with the training loss $\mathcal{L}$ and be non-increasing along the optimization trajectory. For GD, since $\mathcal{L}$ is non-increasing with a properly chosen learning rate, we can pick $\xi(t) = \mathcal{L}(t)$. However, as the update of GDM does not align with the direction of the negative gradient, training loss $\mathcal{L}(t)$ in GDM is no longer monotonously decreasing, and the potential function requires special construction. Inspired by [36], we choose the following $\xi(t)$:

**Lemma 1.** *Let all conditions in Theorem 2 hold. Define $\xi(t) \triangleq \mathcal{L}(\boldsymbol{w}(t)) + \frac{1-\beta}{2\beta\eta}\|\boldsymbol{w}(t) - \boldsymbol{w}(t-1)\|^2$.*[4] *Define $C_1$ as a positive real with $C_1 \triangleq \frac{\sigma_{max}^2 H_{\ell-1}(N\mathcal{L}(\boldsymbol{w}_1))}{2N}\eta$. We then have*

$$\xi(t) \geq \xi(t+1) + \frac{1-C_1}{\eta}\|\boldsymbol{w}(t+1) - \boldsymbol{w}(t)\|^2. \tag{6}$$

**Remark 2.** *Although this potential function is obtained by [36] by directly examining Taylor's expansion at $\boldsymbol{w}(t)$, the proof here is non-trivial as we only require the loss to be locally smooth instead of globally smooth in [36]. We need to prove that the smoothness parameter along the trajectory is upper bounded. We defer the detailed proof to Appendix C.1.1.*

By Lemma 1, we have that $\xi(t)$ is monotonously decreasing by gap $\frac{1-C_1}{\eta}\|\boldsymbol{w}(t+1) - \boldsymbol{w}(t)\|^2$. As $\xi(1) = \mathcal{L}(\boldsymbol{w}(1))$ is a finite number, we have $\sum_{t=1}^{\infty}\|\boldsymbol{w}(t+1) - \boldsymbol{w}(t)\|^2 < \infty$. By that $(1-\beta)\eta\nabla\mathcal{L}(\boldsymbol{w}(t)) = (\boldsymbol{w}(t+1) - \boldsymbol{w}(t)) - \beta(\boldsymbol{w}(t) - \boldsymbol{w}(t-1))$, it immediately follows that $\sum_{t=1}^{\infty}\|\nabla\mathcal{L}(\boldsymbol{w}(t))\|^2 < \infty$.

**Stage II. Parameter Dynamics.** The goal of this stage is to characterize the dynamics of the parameter and show that GDM asymptotically converges (in direction) to the max-margin solution $\hat{\boldsymbol{w}}$. To see this, we define a residual term $\boldsymbol{r}(t) \triangleq \boldsymbol{w}(t) - \ln(t)\hat{\boldsymbol{w}} - \tilde{\boldsymbol{w}}$ with some constant vector $\tilde{\boldsymbol{w}}$ (specified in Appendix C.1.2). If we can show the norm of $\boldsymbol{r}(t)$ is bounded over the iterations, we complete the proof as $\ln(t)\hat{\boldsymbol{w}}$ will then dominates the dynamics of $\boldsymbol{w}(t)$.

For simplicity, we use the continuous dynamics approximation of GDM [36] to demonstrate why $\boldsymbol{r}(t)$ is bounded:

$$\frac{\beta}{1-\beta}\frac{\mathrm{d}^2\boldsymbol{w}(t)}{\mathrm{d}t^2} + \frac{\mathrm{d}\boldsymbol{w}(t)}{\mathrm{d}t} + \nabla\mathcal{L}(\boldsymbol{w}(t)) = 0. \tag{7}$$

We start by directly examining the evolution of $\|\boldsymbol{r}(t)\|$, i.e.,

$$\frac{1}{2}\|\boldsymbol{r}(T)\|^2 - \frac{1}{2}\|\boldsymbol{r}(1)\|^2 = \int_1^T \frac{1}{2}\frac{\mathrm{d}\|\boldsymbol{r}(s)\|^2}{\mathrm{d}s}\mathrm{d}s$$

$$= \int_1^T \left\langle \boldsymbol{r}(s), -\nabla\mathcal{L}(\boldsymbol{w}(s)) - \frac{1}{s}\hat{\boldsymbol{w}} \right\rangle \mathrm{d}s + \int_1^T \frac{\beta}{1-\beta}\left\langle \boldsymbol{r}(s), -\frac{\mathrm{d}^2\boldsymbol{w}(s)}{\mathrm{d}s^2} \right\rangle \mathrm{d}s,$$

which by integration by part leads to

$$\text{RHS} = \int_1^T \left\langle \boldsymbol{r}(s), -\nabla\mathcal{L}(\boldsymbol{w}(s)) - \frac{1}{s}\hat{\boldsymbol{w}} \right\rangle \mathrm{d}s + \frac{\beta}{1-\beta}\left( \left\langle \boldsymbol{r}(T), -\frac{\mathrm{d}\boldsymbol{w}(T)}{\mathrm{d}t} \right\rangle - \left\langle \boldsymbol{r}(1), -\frac{\mathrm{d}\boldsymbol{w}(1)}{\mathrm{d}t} \right\rangle + \int_1^T \left\langle \frac{\mathrm{d}\boldsymbol{r}(s)}{\mathrm{d}s}, \frac{\mathrm{d}\boldsymbol{w}(s)}{\mathrm{d}s} \right\rangle \mathrm{d}s \right),$$

We then check the terms one by one:

- $\int_1^T \left\langle \boldsymbol{r}(s), -\nabla\mathcal{L}(\boldsymbol{w}(s)) - \frac{1}{s}\hat{\boldsymbol{w}} \right\rangle \mathrm{d}s$: This term also occurs in the analysis of GD [34], the analysis of which can be generalized as it does not depend on the form of $\boldsymbol{w}(s)$;

- $\left\langle \boldsymbol{r}(T), -\frac{\mathrm{d}\boldsymbol{w}(T)}{\mathrm{d}t} \right\rangle$: as shown in Stage I, $\frac{\mathrm{d}\boldsymbol{w}(T)}{\mathrm{d}t} \to 0$ (i.e., $\boldsymbol{w}(T) - \boldsymbol{w}(T-1) \to 0$ in the discrete case) as $T \to \infty$. Thus, this term is $o(\|\boldsymbol{r}(T)\|)$;

- $\int_1^T \left\langle \frac{\mathrm{d}\boldsymbol{r}(s)}{\mathrm{d}s}, \frac{\mathrm{d}\boldsymbol{w}(s)}{\mathrm{d}s} \right\rangle \mathrm{d}s$: finite due to $\frac{\mathrm{d}\boldsymbol{r}(s)}{\mathrm{d}s} = \frac{\mathrm{d}\boldsymbol{w}(s)}{\mathrm{d}s} - \frac{1}{s}\hat{\boldsymbol{w}}$, $\int_1^{\infty}\|\frac{\mathrm{d}\boldsymbol{w}(s)}{\mathrm{d}s}\|^2\mathrm{d}s$ is finite [5] (i.e., $\sum_{t=1}^{\infty}\|\boldsymbol{w}(t+1) - \boldsymbol{w}(t)\|^2 < \infty$ in the discrete case) by Stage I, and the mean-value inequality.

---

[4]Please note that $\xi(t) \geq 0$ regardless of $t$.

[5]The derivation of this property is in the same manner as in the discrete case: we choose the potential function as $\xi(t) = \mathcal{L}(w(t)) + \frac{\beta}{2(1-\beta)}\|\frac{\mathrm{d}w(t)}{\mathrm{d}t}\|^2$. The derivative of $\xi(t)$ is $\frac{\mathrm{d}\xi(t)}{\mathrm{d}t} = \langle\nabla\mathcal{L}(\boldsymbol{w}(t)), \frac{\mathrm{d}w(t)}{\mathrm{d}t}\rangle + \frac{\beta}{1-\beta}\langle\frac{\mathrm{d}^2w(t)}{\mathrm{d}t^2}, \frac{\mathrm{d}w(t)}{\mathrm{d}t}\rangle = -\|\frac{\mathrm{d}w(t)}{\mathrm{d}t}\|^2$. Therefore, $\xi(0) - \xi(T) = \int_0^T \|\frac{\mathrm{d}w(t)}{\mathrm{d}t}\|^2\mathrm{d}t$. Then, taking $T \to \infty$ yields the conclusion.

Putting them together, we show that $\|\boldsymbol{r}(T)\|^2 + \boldsymbol{o}(\|\boldsymbol{r}(T)\|)$ is upper bounded over the iterations, which immediately leads to that $\|\boldsymbol{r}(T)\|$ is bounded. Applying similar methodology to the discrete update rule, we have the following lemma (the proof can be found in Appendix C.1.2).

**Lemma 2.** *Define potential function $g : \mathbb{Z}^+ \to \mathbb{R}$ as*

$$g(t) \triangleq \frac{1}{2}\|\boldsymbol{r}(t)\|^2 + \frac{\beta}{1-\beta}\langle \boldsymbol{r}(t), \boldsymbol{w}(t) - \boldsymbol{w}(t-1)\rangle.$$

*$g(t)$ is upper bounded, which further indicates $\|\boldsymbol{r}(t)\|$ is upper bounded.*

**Remark 3.** *Our technique for analyzing GDM here is essentially more complex and elaborate than that for GD in [34] due to the historical information of gradients GDM. The approach in [34] cannot be directly applied. It is worth mentioning that we provide a more easy-to-check condition for whether $\boldsymbol{r}(t)$ is bounded, i.e., "is $g(t)$ upper-bounded?". This condition can be generalized for other momentum-based implicit regularization analyses. E.g., SGDM and Adam later in this paper.*

## 5 Tackle the difficulty brought by random sampling

In this section, we analyze the implicit regularization of SGDM. Parallel to GDM, we establish the following implicit regularization result for SGDM:

**Theorem 3.** *Let Assumption 1, 2, and 3. (S) hold. Let $\beta \in [0,1)$ and $\eta < \frac{1}{\frac{H\beta\sigma_{max}^3}{\sqrt{N}b\gamma(1-\beta)} + \frac{H\sigma_{max}^4}{2b\gamma^2}}$. Then, with arbitrary initialization point $\boldsymbol{w}(1)$, SGDM (w/. r) satisfies $\|\frac{\boldsymbol{w}(t)}{\|\boldsymbol{w}(t)\|} - \frac{\hat{\boldsymbol{w}}}{\|\hat{\boldsymbol{w}}\|}\| = \mathcal{O}(1/\log(t))$ and $\mathcal{L}(\boldsymbol{w}(t)) = \mathcal{O}(\frac{1}{t})$, almost surely (a.s.).*

Similar to the GDM case, Theorem 3 shows that the implicit regularization of SGDM under this setting is consistent with SGD (c.f. [24] for the implicit regularization of SGD). This matches the observations in practice (c.f. Section 7 for details), and is later supported by our experiments (e.g., Figure 1). We add two remarks on the learning rate upper bound and extension to SGDM (w/. r).

**Remark 4** (On the learning rate). *Firstly, our learning rate upper bound $1/(\frac{H\beta\sigma_{max}^3}{\sqrt{N}b\gamma(1-\beta)} + \frac{H\sigma_{max}^4}{2b\gamma^2})$ exactly matches that of SGD $2\frac{b\gamma^2}{H\sigma_{max}^4}$ [24] when $\beta = 0$, and matches that of SGD in terms of the order of $\sigma_{max}$, $H$, and $b$ when $\beta \neq 0$. This indicates our analysis is tight. Secondly, as the bound is monotonously increasing with respect to batch size $b$, Theorem 3 also sheds light on the learning rate tuning, i.e., the larger the batch size is, the larger the learning rate is.*

**Remark 5.** *(On SGDM (w/o. r)) Theorem 3 can be similarly extended to SGDM (w/o. r). We defer the detailed description of the corresponding theorem together with the proof to Appendix F.2.*

Next, we show the proof sketch for Theorem 3. The proof also contains two stages, where Stage II is similar to that for GDM. However, we highlight that **Stage I for SGDM is not a trivial extension of that for GDM**. The methodology used to construct GDM's potential function fails for SGDM: roughly speaking, the change of $\xi(t)$ (defined in Lemma 1) can be spitted into the "decrease term" and the "error term". The "decrease term" comes from the alignment between directions of the update of GDM and the negative gradient, and contributes to the decrease of $\xi(t)$. However, a gap remains between the directions of the update of GDM and the negative gradient. The "error term" measures this gap and potentially contributes to the increase of $\xi(t)$. For GDM, both the terms are in the order of $\|\nabla\mathcal{L}(\boldsymbol{w}(t))\|^2$ while the "decrease term" has a larger coefficient, leading to the decrease of $\xi(t)$. For SGDM, the "decrease term" is in the order of $(\mathbb{E}\|\nabla\mathcal{L}(\boldsymbol{w}(t))\|)^2$, while the "error term" is in the order of $\mathbb{E}(\|\nabla\mathcal{L}(\boldsymbol{w}(t))\|^2)$, bigger than the "decrease term" in an unbounded magnitude. Therefore, $\xi(t)$ may no longer decrease. We defer a detailed discussion to Appendix C.2.3.

On the other hand, we find the stochastic gradient $\nabla\mathcal{L}_{\boldsymbol{B}(t)}(\boldsymbol{w})$ is special, in the sense that it satisfies the strong growth condition [32] as follows.

**Lemma 3.** *Let all the assumptions in Theorem 3 hold. Then, for every $t$, $\nabla\mathcal{L}_{\boldsymbol{B}(t)}(\boldsymbol{w})$ satisfies the strong growth condition, i.e., $\forall \boldsymbol{w}$, $\mathbb{E}_{\boldsymbol{B}(t)}\|\nabla\mathcal{L}_{\boldsymbol{B}(t)}(\boldsymbol{w})\|^2 \leq C_2\|\nabla\mathcal{L}(\boldsymbol{w})\|^2$, where $C_2 \triangleq \frac{N\sigma_{max}^2}{\gamma^2 b}$.*

The proof is deferred to Appendix B.2. SGD has been known to converge faster when the strong growth condition holds compared to the case when the variance of the stochastic gradient norm is

bounded [32, 42]. However, to the best of our knowledge, convergence of SGDM under the strong growth condition has not been established. We then bridge this gap by proposing the following theorem.

**Theorem 4.** *Consider SGDM defined in Eq. (2) with general $\mathcal{L}$ and $\xi(t)$. Let $\nabla\mathcal{L}_{\xi(t)}(\boldsymbol{w})$ satisfy the following properties:*

- *(Affine noise variance condition). $\forall \boldsymbol{w}$, $\mathbb{E}_{\xi(t)}\|\nabla\mathcal{L}_{\xi(t)}(\boldsymbol{w})\|^2 \leq \sigma_1\|\nabla\mathcal{L}(\boldsymbol{w})\|^2 + \sigma_0$.*

- *(Bounded smoothness condition). $\forall \boldsymbol{w}_1, \boldsymbol{w}_2$, $\|\nabla\mathcal{L}(\boldsymbol{w}_1) - \nabla\mathcal{L}(\boldsymbol{w}_2)\| \leq L\|\boldsymbol{w}_1 - \boldsymbol{w}_2\|$.*

*Then, with learning rate $\eta \leq \frac{1}{2\frac{L\beta\sqrt{\sigma_1}}{1-\beta}+L\sigma_1}$, we have*

$$\frac{1}{T}\sum_{t=1}^{T}\mathbb{E}\|\nabla\mathcal{L}(\boldsymbol{w}(t))\|^2 \leq \mathcal{O}\left(\frac{1}{T\eta}\right) + \mathcal{O}(\eta\sigma_0).$$

To the best of our knowledge, Theorem 4 is **the first to establish the convergence of SGDM under the affine noise variance condition [6]**, which generalizes the strong growth condition and the bounded gradient variance condition. Note that the affine noise variance condition with $\sigma_0 = 0$ is exact the strong growth condition, in which case Theorem 4 shows that $\frac{1}{T}\sum_{t=1}^{T}\mathbb{E}\|\nabla\mathcal{L}(\boldsymbol{w}(t))\|^2$ decays with rate $\frac{1}{T}$. Combining Lemma 3 and Theorem 4 completes the proof of Stage I for SGDM.

We then briefly state the proof idea of Theorem 4. Inspired by SGD's simple update rule, we rearrange the update rule of SGDM such that only the gradient information of the current step is contained, i.e.,

$$\frac{\boldsymbol{w}(t+1) - \beta\boldsymbol{w}(t)}{1-\beta} = \frac{\boldsymbol{w}(t) - \beta\boldsymbol{w}(t-1)}{1-\beta} - \eta\nabla\mathcal{L}_{\xi(t)}(\boldsymbol{w}(t)).$$

By defining $\boldsymbol{u}(t) \triangleq \frac{\boldsymbol{w}(t)-\beta\boldsymbol{w}(t-1)}{1-\beta} = \boldsymbol{w}(t) + \frac{\beta}{1-\beta}(\boldsymbol{w}(t) - \boldsymbol{w}(t-1))$, we have that $\boldsymbol{u}(t)$ is close to $\boldsymbol{w}(t)$ (differs by order of one-step update $\boldsymbol{w}(t) - \boldsymbol{w}(t-1)$), and the update rule of $\boldsymbol{u}(t)$ only contains the current-step-gradient information $\nabla\mathcal{L}(\boldsymbol{w}(t))$. We then select potential function as $\mathcal{L}(\boldsymbol{u}(t))$, and a simple Taylor's expansion directly leads to:

$$\mathbb{E}[\mathcal{L}(\boldsymbol{u}(t+1))|\mathcal{F}_t] \approx \mathcal{L}(\boldsymbol{u}(t)) - \eta\langle\nabla\mathcal{L}(\boldsymbol{w}(t)), \nabla\mathcal{L}(\boldsymbol{u}(t))\rangle \approx \mathcal{L}(\boldsymbol{u}(t)) - \eta\|\nabla\mathcal{L}(\boldsymbol{w}(t))\|^2,$$

i.e., $\mathcal{L}(\boldsymbol{u}(t))$ is a proper potential function. Summing the above equation across iterations completes the proof.

## 6 Analyze the effect of preconditioners

### 6.1 Implicit regularization of deterministic Adam

This section presents the implicit regularization of deterministic Adam, i.e., Adam without random sampling.

**Theorem 5.** *Let Assumption 1, 2, and 3. (D) hold. Let $1 > \beta_2 > \beta_1^4 \geq 0$, and the learning rate $\eta$ is a small enough constant (The upper bound of learning rate is complex, and we defer it to Appendix D.1). Then, with arbitrary initialization point $\boldsymbol{w}(1)$, deterministic Adam (Eq. 3) satisfies that $\|\frac{\boldsymbol{w}(t)}{\|\boldsymbol{w}(t)\|} - \frac{\hat{\boldsymbol{w}}}{\|\hat{\boldsymbol{w}}\|}\| = \mathcal{O}(1/\log(t))$ and $\mathcal{L}(\boldsymbol{w}(t)) = \mathcal{O}(\frac{1}{t})$.*

**Remark 6** (On the $\beta_1$ and $\beta_2$ range). *Almost all existing literature assume a time-decaying hyper-parameter choice of $\beta_1$ or $\beta_2$ (c.f., [18, 3]). On the other hand, our result proves that deterministic Adam converges with constant settings of $\beta_1$ and $\beta_2$, which agrees with the practical use.*

**Remark 7** ((Discussion on the results in [34])). *[34] observe that, on a synthetic dataset, the direction of the output parameter by Adam still does not converge to the max-margin direction after $2 \times 10^6$ iterations (but is getting closer). At the same time, GD seems to converge to the max-margin direction. However, this example does not contradict Theorem 5, as we study the asymptotic behavior. In the example, it can be observed that the angle gap keeps decreasing after $10^5$ iteration, which stands with our result. Furthermore, in Figure 3 and 4 of our paper, we reproduce the experiment and find that the angle gap still keeps decreasing after extending the training time. Secondly, this synthetic dataset is ill-posed, which has large singular value and makes the training rather slow, and it can be*

*observed that the training loss of Adam does not vary much until $10^4$ iteration. On the well-posed dataset adopted by (Figure 1, [34]), we observe that Adam will converge to the max-margin solution rapidly (please refer to Figure 1 for details).*

We simply introduce the proof idea here and put the full proof in Appendix D. The proof of deterministic Adam is very similar to that of GDM with minor changes in potential functions. Specifically, $\xi(t)$ in Lemma 1 is changed to $\xi(t) \triangleq \mathcal{L}(\boldsymbol{w}(t)) + \frac{1}{2}\frac{1-\beta_1^{t-1}}{\eta(1-\beta_1)}\| \sqrt[4]{\varepsilon \mathbb{1}_d + \hat{\boldsymbol{\nu}}(t-1)} \odot (\boldsymbol{w}(t) - \boldsymbol{w}(t-1))\|^2$, and $g(t)$ in Lemma 2 is changed to

$$g(t) \triangleq \left\langle \boldsymbol{r}(t), (1-\beta_1^{t-1})\sqrt{\varepsilon \mathbb{1}_d + \hat{\boldsymbol{\nu}}(t-1)} \odot (\boldsymbol{w}(t) - \boldsymbol{w}(t-1)) \right\rangle \frac{\beta_1}{1-\beta_1} + \frac{\sqrt{\varepsilon}}{2}\|\boldsymbol{r}(t)\|^2.$$

The rest of the proof then flows similarly to the GDM case.

## 6.2 What if random sampling is added?

We have obtained the implicit regularization for GDM, SGDM, and deterministic Adam. One may wonder whether the implicit regularization of stochastic Adam can be obtained. Unfortunately, the gap can not be closed yet. This is because an implicit regularization analysis requires the knowledge of the loss dynamics, little of which, however, has been ever known even for stochastic RMSProp (i.e., Adam with $\beta_1 = 0$ in Eq. (3)) with constant learning rates. Specifically, the main difficulty lies in bounding the change of conditioner $\frac{1}{\sqrt{\varepsilon \mathbb{1}_d + \hat{\boldsymbol{\nu}}(t)}}$ across iterations, which is required to make the drift term $\langle \nabla \mathcal{L}(\boldsymbol{w}(t)), \mathbb{E}(\boldsymbol{w}(t+1) - \boldsymbol{w}(t))\rangle$ (derived by Taylor's expansion of the epoch start from $Kt$) negative to ensure a non-increasing loss.

On the other hand, if we adopt decaying learning rates $\eta_t = \frac{\eta_1}{\sqrt{t}}$, [33] shows $\beta_2$ close enough to 1, the following equation holds for stochastic RMSProp (w/o. r) (recall that $K \triangleq \frac{N}{b}$ is the epoch size)

$$\sum_{t=0}^{T} \frac{1}{\sqrt{t+1}}\|\nabla \mathcal{L}(\boldsymbol{w}(Kt))\| = \mathcal{O}(\ln T). \tag{8}$$

Based on this result, we have the following theorem for stochastic RMSProp (w/o. r):

**Theorem 6.** *Let Assumptions 1, 2, and 3. (S) hold. Let $\beta_2$ be close enough to 1. Then, with arbitrary initialization point $\boldsymbol{w}(1)$ and decaying learning rate $\eta_t = \frac{\eta_1}{\sqrt{t}}$, stochastic RMSProp satisfies* $\|\frac{\boldsymbol{w}(t)}{\|\boldsymbol{w}(t)\|} - \frac{\hat{\boldsymbol{w}}}{\|\hat{\boldsymbol{w}}\|}\| = \mathcal{O}(1/\log(t))$ *and* $\mathcal{L}(\boldsymbol{w}(t)) = \mathcal{O}(\frac{1}{\sqrt{t}})$.

**Remark 8** (On the decaying learning rate). *The decaying learning rate is a "stronger" setting compared to the constant learning rate, both in the sense that GDM, SGDM, and deterministic Adam can be shown to converge to the max-margin solution following the same routine as Theorems 2, 3, and 5, and in the sense that we usually adopt constant learning rate in practice.*

The proof is on the grounds of a novel characterization of the loss convergence rate derived from Eq. (8), and readers can find the details in Appendix E.

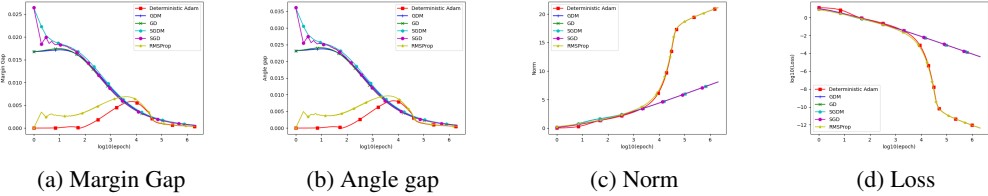

| (a) Margin Gap | (b) Angle gap | (c) Norm | (d) Loss |

Figure 1: Comparison of the implicit regularization of the optimizers. We use the synthetic dataset in [34] with learning rate $\frac{1}{\sigma_{max}^2}$. Figure (b) shows (1). all the optimizers converge to the max margin solution, and (2). the asymptotic behaviors with & without momentum are similar. The experimental observation support our theoretical results.

# 7 Discussions

**Consistency with the Experimental Results.** We conduct experiments to verify our theoretical findings. Specifically, we (1). run GD, GDM, SGD, SGDM, and Adam on a synthetic dataset to observe their implicit regularization; (2). run GD and Adam on ill-posed dataset proposed in [34] to verify Theorem 5; (3). run SGD and SGDM on neural networks to classify the MNIST dataset and compare their implicit regularization. The experimental observations stand with our theoretical results. Furthermore, it is worth mentioning that experimental phenomenons that adding momentum will not change the implicit regularization have also been observed by existing literature [34, 23, 43].

**Influence of hyperparameters on convergence rates.** Our results can be further extended to provide a precise characterization of the influence of the hyperparameters $\eta$ and $\beta$ on the convergence rate of (S)GDM. Specifically, in Appendix F.4, we show that the asymptotic convergence rate of (S)GDM is $C\frac{1}{\eta}\frac{1}{t}$, where $C$ is some constant independent of $\beta$ and $\eta$. Therefore, increase $\eta$ can lead to a faster convergence rate (with learning rate requirements in Theorems 2 and 3 satisfied). However, changing $\beta$ does not affect the convergence rate, which is also observed in our experiments (e.g., Figure 1). Furthermore, to the best of our knowledge, there still lacks the theoretical justification of the acceleration effect of the momentum except the strongly convex case. On the other hand, we still can not provide a precise characterization of the exact convergence behavior of Adam in this case due to the ever-changing preconditioner. As this work mainly focuses on the linear model, whether momentum and preconditioner can accelerate the training of non-linear model with the non-convex landscape is an exciting future direction and deserves further investigation.

**Generalization Behavior of Adam.** Our results characterize the implicit regularization behavior of optimizers when the time is large enough, which can be changed by early stopping, especially for Adam. This is because the proof of Theorem 5 relies on that when the training time is large enough, the gradient is small and the adaptive learning rate $\frac{1}{\sqrt{\nu(t)+\varepsilon\mathbb{1}_d}}$ is dominated by $\frac{1}{\varepsilon}\mathbb{1}_d$, and then Adam behaves like GDM and thus converges to the max margin solution. This can also be verified by Figure 1, as the angle gap of deterministic Adam and RMSProp only start to decrease late in the training.

**Gap Between The Linear Model and Deep Neural Networks.** While our results only hold for the linear classification problem, extending the results to the deep neural networks is possible. Specifically, existing literature [21, 43] provide a framework for deriving implicit regularization for deep homogeneous neural networks. However, the approach in [21, 43] can not be trivially applied to the momentum-based optimizers, as their proofs require the specific gradient-based updates to lower bound a smoothed margin (c.f., Theorem 4.1, [21]). It remains an exciting work to see how our results can be expanded to GDM and Adam for deep neural networks.

# 8 Conclusion

This paper studies the implicit regularization of momentum-based optimizers in linear classification with exponential-tailed loss. Our results indicate that for SGD and the deterministic version of Adam, adding momentum will not influence the implicit regularization, and the direction of the parameter converges to the $L^2$ max-margin solution. Our theoretical results stand with existing experimental observations, and developed techniques such as the potential functions may inspire the analyses on other momentum-based optimizers. Motivated by the results and techniques for linear cases in this paper, it has the potential to extend them to the homogeneous neural network in the future. Another topic left for future work is to derive the implicit regularization of constant learning rate stochastic Adam. As discussed in Section 6.2, this topic is non-trivial, and it requires new techniques and assumptions to be developed.

# 9 Acknowledgement

This research is supported in part by CAS Project for Young Scientists in Basic Research under Grant No. YSBR-034 and Innovation Project of ICT CAS under Grants No. E261090.

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
