## A  Additional Related Works

[11] and [49] also investigate the generalization behavior of GDM and Adam but focus on a different setting from ours. Specifically, they work on a two-layer convolutional neural network with fixed and untrained second layer and cubic activation, while we work on the linear classifier. Also, except linear separability, they further pose additional specific requirements for the dataset (e.g., all data shares a same (scaled) patch). Based on the settings, [11] shows that the output hypothesis by GDM provably generalizes better than GD, while [49] proves that the output hypothesis by GD generalizes better than Adam. These results, however, do not contradict our findings due to the difference in the settings. We believe both the works and this paper have its own merits to unveil the mystery of the generalization of momentum-based optimizers.

## B  Preparations

This section collect definitions and lemmas which will be used throughout the proofs.

### B.1  Characterization of the max-margin solution

This section collects several commonly-used characterization of the max-margin solution from [24] and [34].

To start with, we define support vectors and support set, which are two common terms in margin analysis. Recall that in the main text, we assume that without the loss of generality, $y_i = 1$, $\forall i \in \{1, \cdots, N\}$.

**Definition 2** (Support vectors and support set). *For any $i \in [N]$, $\boldsymbol{x}_i$ is called a support vector of the dataset $\boldsymbol{S}$, if*

$$\langle \boldsymbol{x}_i, \hat{\boldsymbol{w}} \rangle = 1.$$

*Correspondingly, $\boldsymbol{x}_i$ is called a non-support vector if $\langle \boldsymbol{x}_i, \hat{\boldsymbol{w}} \rangle > 1$. The support set of $\boldsymbol{S}$ is then defined as*

$$\boldsymbol{S}_s = \{\boldsymbol{x} \in \boldsymbol{S} : \langle \boldsymbol{x}, \hat{\boldsymbol{w}} \rangle = 1\}.$$

The following lemma delivers $\hat{\boldsymbol{w}}$ as an linear combination of support vectors.

**Lemma 4** (Lemma 12, [34]). *For almost every datasets $S$, there exists a unique vector $\boldsymbol{v} = (\boldsymbol{v}_1, \cdots, \boldsymbol{v}_N)$, such that $\hat{\boldsymbol{w}}$ can be represented as*

$$\hat{\boldsymbol{w}} = \sum_{i=1}^{N} \boldsymbol{v}_i \boldsymbol{x}_i, \tag{9}$$

*where $\boldsymbol{v}$ satisfies $\boldsymbol{v}_i = 0$ if $\boldsymbol{x}_i \notin \boldsymbol{S}_s$, and $\boldsymbol{v}_i > 0$ if $\boldsymbol{x}_i \in \boldsymbol{S}_s$. Furthermore, the size of $\tilde{\boldsymbol{S}}_s$ is at most $d$.*

By Lemma 4, we further have the following corollary:

**Corollary 1.** *For almost every datasets $S$, the unique $\boldsymbol{v}$ given by Lemma 4 further satisfies that for any positive constant $C_3$, there exists a non-zero vector $\tilde{\boldsymbol{w}}$, such that, $\boldsymbol{x}_i \in \boldsymbol{S}_s$, we have*

$$C_3 e^{-\langle \boldsymbol{x}_i, \bar{\boldsymbol{w}} \rangle} = \boldsymbol{v}_i. \tag{10}$$

*Proof.* For almost every datasets $\boldsymbol{S}$, any subsets with size $d$ of $\boldsymbol{S}$ is linearly independent. Since $\tilde{\boldsymbol{S}}_s$ has size no larger than $d$ (by Lemma 4), and Eq. (10) is equivalent to linear equations, the proof is completed. □

For the stochastic case, we will also need the following lemma when we calculate the form of parameter at time $t$.

**Lemma 5** (Lemma 5, [24])**.** *Let $B(s)$ be the random subset used in SGDM (w/. r). Almost surely, there exists a vector $\check{\boldsymbol{w}}$*

$$\frac{N}{b} \sum_{s=1}^{t-1} \frac{1}{s} \sum_{\boldsymbol{x}_i \in B(s) \cap S_s} v_i \boldsymbol{x}_i = \ln\left(\frac{bt}{N}\right) \hat{\boldsymbol{w}} + \boldsymbol{n}(t) + \check{\boldsymbol{w}},$$

*where $\boldsymbol{n}(t)$ satisfies $\|\boldsymbol{n}(t)\| = \boldsymbol{o}(t^{-0.5+\varepsilon})$ for any $\varepsilon > 0$, and $\|\boldsymbol{n}(t+1) - \boldsymbol{n}(t)\| = O(t^{-1})$. As for SGDM (w/o. r), the a.s. condition can be removed.*

### B.2 Preparations of the optimization analysis

This section collects technical lemmas which will be used in latter proofs. We begin with a lemma bounding the smooth constants if the loss is bounded.

**Lemma 6.** *If loss $\ell$ satisfies (D) in Assumption 3, then for any $\boldsymbol{w}_0$, if $\mathcal{L}(\boldsymbol{w}) \leq \mathcal{L}(\boldsymbol{w}_0)$, then we have $\mathcal{L}$ is $\sigma_{max}^2 H_{s_0}$ smooth at point $\boldsymbol{w}$, where $s_0 = \ell^{-1}(N\mathcal{L}(\boldsymbol{w}_0))$. Furthermore, $\mathcal{L}$ is globally $\sigma_{max}^2 H_{s_0}$ smooth over the set $\{\boldsymbol{w} : \mathcal{L}(\boldsymbol{w}) \leq \mathcal{L}(\boldsymbol{w}_0)\}$.*

*Proof.* Since $\ell$ is positive, we have $\forall i \in [N]$,

$$\frac{\tilde{\ell}(\boldsymbol{w}, \boldsymbol{x}_i)}{N} < \frac{\sum_{j=1}^N \tilde{\ell}(\boldsymbol{w}, \boldsymbol{x}_j)}{N} = \mathcal{L}(\boldsymbol{w}) \leq \mathcal{L}(\boldsymbol{w}_0),$$

which leads to $\tilde{\ell}(\boldsymbol{w}, \boldsymbol{x}_i) < N\mathcal{L}(\boldsymbol{w}_0)$, and $\ell$ is $H_{s_0}$ smooth at $\langle \boldsymbol{w}, \boldsymbol{x}_i \rangle$.

Furthermore, since $\nabla_{\boldsymbol{w}} \tilde{\ell}(\boldsymbol{w}, \boldsymbol{x}_i) = \nabla_{\boldsymbol{w}} \ell(\langle \boldsymbol{w}, \boldsymbol{x}_i \rangle) = \ell'(\langle \boldsymbol{w}, \boldsymbol{x}_i \rangle) \boldsymbol{x}_i$, for any two parameters $\boldsymbol{w}_1$ and $\boldsymbol{w}_2$ close enough to $\boldsymbol{w}$,

$$\|\nabla_{\boldsymbol{w}} \mathcal{L}(\boldsymbol{w}_1) - \nabla_{\boldsymbol{w}} \mathcal{L}(\boldsymbol{w}_2)\| = \left\| \sum_{\boldsymbol{x} \in S} (\ell'(\langle \boldsymbol{w}_1, \boldsymbol{x} \rangle) - \ell'(\langle \boldsymbol{w}_2, \boldsymbol{x} \rangle)) \boldsymbol{x} \right\|$$

$$\leq \sigma_{max} \sqrt{\sum_{\boldsymbol{x} \in S} (\ell'(\langle \boldsymbol{w}_1, \boldsymbol{x} \rangle) - \ell'(\langle \boldsymbol{w}_2, \boldsymbol{x} \rangle))^2} \leq \sigma_{max} H_{s_0} \sqrt{\sum_{\boldsymbol{x} \in S} (\langle \boldsymbol{w}_1 - \boldsymbol{w}_2, \boldsymbol{x} \rangle)^2}$$

$$\leq \sigma_{max}^2 H_{s_0} \|\boldsymbol{w}_1 - \boldsymbol{w}_2\|.$$

Now if $\boldsymbol{w}_1$ and $\boldsymbol{w}_2$ both belong to $\{\boldsymbol{w} : \mathcal{L}(\boldsymbol{w}) \leq \mathcal{L}(\boldsymbol{w}_0)\}$, we have for any $\boldsymbol{x}_i \in S$, $\langle \boldsymbol{w}_1, \boldsymbol{x}_i \rangle > \ell^{-1}(N\mathcal{L}(\boldsymbol{w}_0))$, and $\langle \boldsymbol{w}_2, \boldsymbol{x}_i \rangle > \ell^{-1}(N\mathcal{L}(\boldsymbol{w}_0))$. Following the same routine as the locally smooth proof, we complete the second argument.

The proof is completed. $\qquad\square$

Based on Assumption 2, we also have the following lemma characterizing the relationship between loss $\ell$ and its derivative $\ell'$ when $x$ is large enough.

**Lemma 7.** *Let loss $\ell$ satisfy Assumption 2. Then, there exists an large enough $x_0$ and a positive real $K$, such that, $\forall x > x_0$, we have*

$$-\frac{1}{4}\ell'(x) \leq \ell(x) \leq -4\ell'(x).$$

*Proof.* By Assumption 2, there exists a large enough $x_0$, such that $\forall x > x_0$, we have

$$\frac{1}{2}e^{-x} \leq -\ell'(x) \leq 2e^{-x}. \tag{11}$$

On the other hand, as $\lim_{t \to \infty} \ell(x) = 0$, we have

$$\ell(x) = \int_{s=x}^{\infty} -\ell'(s)\mathrm{d}s,$$

which by Eq. (11) leads to

$$\frac{1}{2}e^{-x} = \frac{1}{2}\int_x^{\infty} e^{-s}\mathrm{d}s \leq \ell(x) \leq 2\int_x^{\infty} e^{-s}\mathrm{d}s = 2e^{-x}.$$

The proof is completed. $\qquad\square$

By Lemma 7, we immediately get the following corollary:

**Corollary 2.** *Let loss $\ell$ satisfy Assumption 2. Then, there exist positive reals $C_g$ and $C_l$, such that, for any $\boldsymbol{w} \in \mathbb{R}^d$ satisfying either $\|\nabla\mathcal{L}(\boldsymbol{w})\| \leq C_g$ or $\mathcal{L}(\boldsymbol{w}) \leq C_l$, we have*

$$\frac{\gamma}{4}\mathcal{L}(\boldsymbol{w}) \leq \|\nabla\mathcal{L}(\boldsymbol{w})\| \leq 4\mathcal{L}(\boldsymbol{w}).$$

*Proof.* We start with the case $\|\nabla\mathcal{L}(\boldsymbol{w})\| \leq C_g$. By simple calculation, we have

$$\|\nabla\mathcal{L}(\boldsymbol{w})\| = \frac{1}{N}\left\|\sum_{i=1}^{N}\ell'(\langle\boldsymbol{w},\boldsymbol{x}_i\rangle)\boldsymbol{x}_i\right\| \leq -\frac{\sigma_{max}}{N}\sum_{i=1}^{N}\ell'(\langle\boldsymbol{w},\boldsymbol{x}_i\rangle), \tag{12}$$

and

$$\|\nabla\mathcal{L}(\boldsymbol{w})\|\|\hat{\boldsymbol{w}}\| \geq \langle\nabla\mathcal{L}(\boldsymbol{w}),\hat{\boldsymbol{w}}\rangle \geq -\frac{1}{N}\sum_{i=1}^{N}\ell'(\langle\boldsymbol{w},\boldsymbol{x}_i\rangle). \tag{13}$$

By Assumption 2, we have there exists a constant $C_g'$, s.t., any $x$ with $-\ell'(x) > C_g'$ satisfies $x > x_0$. Let $C_g = \frac{C_g'\gamma}{N}$. We then have if $\|\nabla\mathcal{L}(\boldsymbol{w})\| \leq C_g$, then $\langle\boldsymbol{w},\boldsymbol{x}_i\rangle > x_0$ $(\forall i)$, and thus $4\ell(\langle\boldsymbol{w},\boldsymbol{x}_i\rangle) \geq -\ell'(\langle\boldsymbol{w},\boldsymbol{x}_i\rangle) \geq \frac{1}{4}\ell(\langle\boldsymbol{w},\boldsymbol{x}_i\rangle)$. Combing Eqs. (12) and (13), we then have

$$4\mathcal{L}(\boldsymbol{w}) = 4\sum_{i=1}^{N}\ell(\langle\boldsymbol{w},\boldsymbol{x}_i\rangle) \geq \|\nabla\mathcal{L}(\boldsymbol{w})\| \geq \frac{\gamma}{4N}\sum_{i=1}^{N}\ell(\langle\boldsymbol{w},\boldsymbol{x}_i\rangle) = \frac{\gamma}{4}\mathcal{L}(\boldsymbol{w}).$$

Similarly, as for the case $\mathcal{L}(\boldsymbol{w}) \leq C_l$, we have there exists a constant $C_l'$, s.t., any $x$ with $\ell(x) < C_l'$ satisfies $x > x_0$. Let $C_l = \frac{C_l\sigma_{max}}{N}$ and the rest of the proof follows the same routine as the first case.

The proof is completed. $\square$

The following lemma bridges the second moment of $\nabla\mathcal{L}_{\boldsymbol{B}(t)}$ with its squared first moment.

**Lemma 8** (Lemma 3, restated). *Let the dataset $\boldsymbol{S}$ satisfies the separable assumption 1. Let $\boldsymbol{B}$ be a random subset of $\boldsymbol{S}$ with size $b$ sampled independently and uniformly without replacement. Then, at any point $\boldsymbol{w}$, we have*

$$\|\nabla\mathcal{L}(\boldsymbol{w})\|^2 \leq \mathbb{E}_{\boldsymbol{B}}\left[\|\nabla\mathcal{L}_{\boldsymbol{B}}(\boldsymbol{w})\|^2\right] \leq \frac{N\sigma_{max}^2}{\gamma^2 b}\|\nabla\mathcal{L}(\boldsymbol{w})\|^2.$$

*Proof.* To start with, notice that

$$\|\nabla\mathcal{L}(\boldsymbol{w})\| = \|\mathbb{E}_{\boldsymbol{B}}\nabla\mathcal{L}_{\boldsymbol{B}}(\boldsymbol{w})\| \leq \mathbb{E}_{\boldsymbol{B}}\|\nabla\mathcal{L}_{\boldsymbol{B}}(\boldsymbol{w})\|.$$

Therefore, the first inequality can be directly obtained by Cauchy-Schwartz's inequality. To prove the second inequality, we first calculate the explicit form of $\nabla\mathcal{L}_{\boldsymbol{B}}(\boldsymbol{w})$.

$$\|\nabla\mathcal{L}_{\boldsymbol{B}}(\boldsymbol{w})\|^2 = \frac{1}{b^2}\left\|\sum_{\boldsymbol{x}\in\boldsymbol{B}}\nabla\tilde{\ell}(\boldsymbol{w},\boldsymbol{x})\right\|^2 = \frac{1}{b^2}\left\|\sum_{\boldsymbol{x}\in\boldsymbol{B}}\ell'(\langle\boldsymbol{w},\boldsymbol{x}\rangle)\boldsymbol{x}\right\|^2 \leq \frac{\sigma_{max}^2}{b^2}\sum_{\boldsymbol{x}\in\boldsymbol{B}}\ell'(\langle\boldsymbol{w},\boldsymbol{x}\rangle)^2.$$

Therefore,

$$\mathbb{E}_{\boldsymbol{B}}\|\nabla\mathcal{L}_{\boldsymbol{B}}(\boldsymbol{w})\|^2 \leq \frac{\sigma_{max}^2}{Nb}\sum_{\boldsymbol{x}\in\boldsymbol{S}}\ell'(\langle\boldsymbol{w},\boldsymbol{x}\rangle)^2 \leq \frac{\sigma_{max}^2}{Nb}\left(\sum_{\boldsymbol{x}\in\boldsymbol{S}}\ell'(\langle\boldsymbol{w},\boldsymbol{x}\rangle)\right)^2. \tag{14}$$

On the other hand,

$$\|\nabla\mathcal{L}(\boldsymbol{w})\| = \frac{1}{N}\left\|\sum_{\boldsymbol{x}\in\boldsymbol{S}}\ell'(\langle\boldsymbol{w},\boldsymbol{x}\rangle)\boldsymbol{x}\right\|$$

$$\geq \frac{1}{N}\left\langle\sum_{\boldsymbol{x}\in\boldsymbol{S}}\ell'(\langle\boldsymbol{w},\boldsymbol{x}\rangle)\boldsymbol{x}, -\frac{\hat{\boldsymbol{w}}}{\|\hat{\boldsymbol{w}}\|}\right\rangle \overset{(\star)}{\geq} \frac{\gamma}{N}\sum_{\boldsymbol{x}\in\boldsymbol{S}}\ell'(\langle\boldsymbol{w},\boldsymbol{x}\rangle)$$

where Eq. ($\star$) is due to $\forall \boldsymbol{x} \in \boldsymbol{S}$, $\langle \boldsymbol{x}, -\hat{\boldsymbol{w}} \rangle \geq 1$ and $\ell' < 0$.

Therefore,

$$\|\nabla \mathcal{L}(\boldsymbol{w})\|^2 \geq \frac{\gamma^2}{N^2} \left( \sum_{\boldsymbol{x} \in \boldsymbol{S}} \ell'(\langle \boldsymbol{w}, \boldsymbol{x} \rangle) \right)^2. \tag{15}$$

The proof is completed by putting Eqs. (14) and (15) together. $\qquad\square$

In the following lemma, we show the updates of GDM, Adam, and SGDM are all non-zero.

**Lemma 9.** *Regardless of GDM, Adam, or SGDM, the updates of all steps are non-zero, i.e.,*

$$\|\boldsymbol{w}(t+1) - \boldsymbol{w}(t)\| > 0, \forall t > 1.$$

*Proof.* We start with the alternative forms of the update rule of GDM, Adam, and SGDM using the gradients along the trajectory respectively. For GDM, by Eq. (1), the update rule can be written as

$$\boldsymbol{w}(t+1) - \boldsymbol{w}(t) = -\eta(1-\beta) \left( \sum_{s=1}^{t} \beta^{t-s} \nabla \mathcal{L}(\boldsymbol{w}(s)) \right). \tag{16}$$

Similarly, the update rule of SGDM can be written as

$$\boldsymbol{w}(t+1) - \boldsymbol{w}(t) = -\eta(1-\beta) \left( \sum_{s=1}^{t} \beta^{t-s} \nabla \mathcal{L}_{\boldsymbol{B}(s)}(\boldsymbol{w}(s)) \right), \tag{17}$$

while the update rule of Adam can be given as

$$\boldsymbol{w}(t+1) - \boldsymbol{w}(t) = -\eta \frac{\sum_{s=1}^{t} \frac{1-\beta_1}{1-\beta_1^s} \beta_1^{t-s} \nabla \mathcal{L}(\boldsymbol{w}(s))}{\sqrt{\varepsilon \mathbf{1}_d + \sum_{s=1}^{t} \frac{1-\beta_2}{1-\beta_2^s} \beta_2^{t-s} (\nabla \mathcal{L}(\boldsymbol{w}(s)))^2}}. \tag{18}$$

On the other hand, by the definition of empirical risk $\mathcal{L}$, the gradient of $\mathcal{L}$ at point $\boldsymbol{w}$ can be given as

$$\nabla \mathcal{L}(\boldsymbol{w}) = \frac{\sum_{i=1}^{N} \ell'(\langle \boldsymbol{w}, \boldsymbol{x}_i \rangle) \boldsymbol{x}_i}{N}. \tag{19}$$

By Eq. (19) and Eq. (16), we further have for GDM,

$$\boldsymbol{w}(t+1) - \boldsymbol{w}(t) = -\eta(1-\beta) \left( \sum_{s=1}^{t} \beta^{t-s} \frac{\sum_{i=1}^{N} \ell'(\langle \boldsymbol{w}(s), \boldsymbol{x}_i \rangle) \boldsymbol{x}_i}{N} \right). \tag{20}$$

By Assumption 1, there exists a non-zero parameter $\hat{\boldsymbol{w}}$, such that, $\langle \hat{\boldsymbol{w}}, \boldsymbol{x}_i \rangle > 0, \forall i$. Therefore, by executing inner product between Eq. (20) and $\hat{\boldsymbol{w}}$, we have

$$\|\boldsymbol{w}(t+1) - \boldsymbol{w}(t)\| \|\hat{\boldsymbol{w}}\| \geq \langle \boldsymbol{w}(t+1) - \boldsymbol{w}(t), \hat{\boldsymbol{w}} \rangle$$
$$= -(1-\beta)\eta \left( \sum_{s=1}^{t} \beta^{t-s} \frac{\sum_{i=1}^{N} \ell'(\langle \boldsymbol{w}(s), \boldsymbol{x}_i \rangle) \langle \boldsymbol{x}_i, \hat{\boldsymbol{w}} \rangle}{N} \right) \overset{(*)}{>} 0,$$

where Eq. ($*$) is due to $\ell' < 0$. This complete the proof for GDM.

Similarly, for SGDM, we have

$$\|\boldsymbol{w}(t+1) - \boldsymbol{w}(t)\| \|\hat{\boldsymbol{w}}\| \geq -\eta(1-\beta) \left( \sum_{s=1}^{t} \beta^{t-s} \frac{\sum_{(\boldsymbol{x},\boldsymbol{y}) \in \boldsymbol{B}} \ell'(\langle \boldsymbol{w}(s), \boldsymbol{y}\boldsymbol{x} \rangle) \langle \boldsymbol{y}\boldsymbol{x}, \hat{\boldsymbol{w}} \rangle}{b} \right) > 0,$$

which completes the proof of SGDM.

For Adam, we have

$$\|\boldsymbol{w}(t+1) - \boldsymbol{w}(t)\| \left\| \hat{\boldsymbol{w}} \odot \sqrt{\varepsilon \mathbf{1}_d + \sum_{s=1}^t \frac{1-\beta_2}{1-\beta_2^s} \beta_2^{t-s} (\nabla \mathcal{L}(\boldsymbol{w}(s)))^2} \right\|$$

$$\geq - \left\langle \hat{\boldsymbol{w}} \odot \sqrt{\varepsilon \mathbf{1}_d + \sum_{s=1}^t \frac{1-\beta_2}{1-\beta_2^s} \beta_2^{t-s} (\nabla \mathcal{L}(\boldsymbol{w}(s)))^2}, \eta \frac{\sum_{s=1}^t \frac{1-\beta_1}{1-\beta_1^s} \beta_1^{t-s} \nabla \mathcal{L}(\boldsymbol{w}(s))}{\sqrt{\varepsilon \mathbf{1}_d + \sum_{s=1}^t \frac{1-\beta_2}{1-\beta_2^s} \beta_2^{t-s} (\nabla \mathcal{L}(\boldsymbol{w}(s)))^2}} \right\rangle$$

$$= \left\langle \hat{\boldsymbol{w}}, \eta \sum_{s=1}^t \frac{1-\beta_1}{1-\beta_1^s} \beta_1^{t-s} \nabla \mathcal{L}(\boldsymbol{w}(s)) \right\rangle$$

$$= - \eta \left( \sum_{s=1}^t \frac{1-\beta_1}{1-\beta_1^s} \beta_1^{t-s} \frac{\sum_{i=1}^N \ell'(\langle \boldsymbol{w}(s), \boldsymbol{x}_i \rangle) \langle \boldsymbol{x}_i, \hat{\boldsymbol{w}} \rangle}{N} \right) > 0,$$

which completes the proof of Adam.

The proof is completed. □

## C    Implicit regularization of GD/SGD with momentum

This section collects the proof of the implicit regularization of gradient descent with momentum and stochastic gradient descent with momentum. The analyses of this section hold for almost every dataset, and the "almost every" constraint is further moved in Section F.1.

### C.1    Implicit regularization of GD with Momentum

This section collects the proof of Theorem 2.

#### C.1.1    Proof of the sum of squared gradients converges

To begin with, we will prove the sum of squared norm of gradients along the trajectory is finite for gradient descent with momentum. To see this, we first define the continuous-time update rule as

$$\boldsymbol{w}(t+\alpha) - \boldsymbol{w}(t) = \alpha(\boldsymbol{w}(t+1) - \boldsymbol{w}(t)), \forall t \in \mathbb{Z}^+, \forall \alpha \in [0,1].$$

We then prove a generalized case of Lemma 1 for any $\boldsymbol{w}(t+\alpha)$.

**Lemma 10** (Lemma 1, extended). *Let all conditions in Theorem 2 hold. We then have*

$$\mathcal{L}(\boldsymbol{w}(t)) + \frac{\beta}{2\eta(1-\beta)} \|\boldsymbol{w}(t) - \boldsymbol{w}(t-1)\|^2 \geq \mathcal{L}(\boldsymbol{w}(t+\alpha)) + \frac{\beta}{2\eta(1-\beta)} \alpha^2 \|\boldsymbol{w}(t+1) - \boldsymbol{w}(t)\|^2$$

$$+ \frac{(1-C_1)\alpha^2}{\eta} \|\boldsymbol{w}(t+1) - \boldsymbol{w}(t)\|^2, \qquad (21)$$

*where $C_1$ is a positive real such that $\eta = 2 \frac{N}{H_{s_0} \sigma_{max}^2} C_1$ and $s_0 \triangleq \ell^{-1}(N\mathcal{L}(\boldsymbol{w}_1))$.*

*Proof of Lemma 10.* We prove this lemma by reduction to absurdity.

Concretely, let $t^*$ be the smallest positive integer time such that there exists an $\alpha \in [0,1]$, such that Eq. (21) doesn't hold. Let $\alpha^* = \inf\{\alpha \in [0,1] : Eq. (21) \ doesn't \ hold \ for \ (t^*, \alpha)\}$. By continuity, Eq. (21) holds for $(t^*, \alpha^*)$.

We further divide the proof into two cases depending on the value of $\alpha^*$.

**Case 1:** $\alpha^* = 0$: For any $t^* > t \geq 1$, we have Eq. (21) holds for $(t, 1)$. Specifically, we have

$$\mathcal{L}(\boldsymbol{w}(t)) + \frac{\beta}{2(1-\beta)\eta} \|\boldsymbol{w}(t) - \boldsymbol{w}(t-1)\|^2 \geq \mathcal{L}(\boldsymbol{w}(t+1)) + \frac{\beta}{2(1-\beta)\eta} \|\boldsymbol{w}(t+1) - \boldsymbol{w}(t)\|^2,$$

which further leads to

$$\mathcal{L}(\boldsymbol{w}(1)) = \mathcal{L}(\boldsymbol{w}(1)) + \frac{\beta}{2(1-\beta)\eta}\|\boldsymbol{w}(1) - \boldsymbol{w}(0)\|^2 \geq \mathcal{L}(\boldsymbol{w}(t^*)) + \frac{\beta}{2(1-\beta)\eta}\|\boldsymbol{w}(t^*) - \boldsymbol{w}(t^*-1)\|^2.$$

Since $\frac{\beta}{2\eta}\|\boldsymbol{w}(t^*) - \boldsymbol{w}(t^*-1)\|^2$ is non-negative, we have

$$\mathcal{L}(\boldsymbol{w}(1)) \geq \mathcal{L}(\boldsymbol{w}(t^*)).$$

By Lemma 6, we have $\mathcal{L}$ is $H_{s_0}$ smooth at $\boldsymbol{w}(t^*)$. Therefore, by Taylor's expansion for $\mathcal{L}$ at point $\boldsymbol{w}(t^*)$, we have for small enough $\alpha > 0$

$$\mathcal{L}(\boldsymbol{w}(t^* + \alpha))$$

$$\leq \mathcal{L}(\boldsymbol{w}(t^*)) + \langle \nabla \mathcal{L}(\boldsymbol{w}(t^*)), \boldsymbol{w}(t^* + \alpha) - \boldsymbol{w}(t^*)\rangle + \frac{H_{s_0}\sigma_{max}^2}{2N}\|\boldsymbol{w}(t^* + \alpha) - \boldsymbol{w}(t^*)\|^2$$

$$= \mathcal{L}(\boldsymbol{w}(t^*)) + \alpha \langle \nabla \mathcal{L}(\boldsymbol{w}(t^*)), \boldsymbol{w}(t^* + 1) - \boldsymbol{w}(t^*)\rangle + \frac{H_{s_0}\alpha^2\sigma_{max}^2}{2N}\|\boldsymbol{w}(t^* + 1) - \boldsymbol{w}(t^*)\|^2$$

$$\overset{(*)}{=} \mathcal{L}(\boldsymbol{w}(t^*)) + \alpha \left\langle \frac{1}{(1-\beta)\eta}(\beta(\boldsymbol{w}(t^*) - \boldsymbol{w}(t^*-1)) - (\boldsymbol{w}(t^*+1) - \boldsymbol{w}(t^*))), \boldsymbol{w}(t^*+1) - \boldsymbol{w}(t^*) \right\rangle$$

$$+ \frac{H_{s_0}\alpha^2\sigma_{max}^2}{2N}\|\boldsymbol{w}(t^*+1) - \boldsymbol{w}(t^*)\|^2$$

$$= \mathcal{L}(\boldsymbol{w}(t^*)) + \frac{\alpha\beta}{(1-\beta)\eta}\langle(\boldsymbol{w}(t^*) - \boldsymbol{w}(t^*-1), \boldsymbol{w}(t^*+1) - \boldsymbol{w}(t^*)\rangle + \left(\frac{H_{s_0}\alpha^2\sigma_{max}}{2N} - \frac{\alpha}{(1-\beta)\eta}\right)\|\boldsymbol{w}(t^*+1) - \boldsymbol{w}(t^*)\|^2$$

$$\overset{(**)}{\leq} \mathcal{L}(\boldsymbol{w}(t^*)) + \frac{\alpha\beta}{2(1-\beta)\eta}\|(\boldsymbol{w}(t^*) - \boldsymbol{w}(t^*-1)\|^2 + \frac{\alpha\beta}{2(1-\beta)\eta}\|(\boldsymbol{w}(t^*+1) - \boldsymbol{w}(t^*)\|^2$$

$$+ \left(\frac{H_{s_0}\alpha^2\sigma_{max}^2}{2N} - \frac{\alpha}{(1-\beta)\eta}\right)\|\boldsymbol{w}(t^*+1) - \boldsymbol{w}(t^*)\|^2$$

$$= \mathcal{L}(\boldsymbol{w}(t^*)) + \frac{\alpha\beta}{2(1-\beta)\eta}\|(\boldsymbol{w}(t^*) - \boldsymbol{w}(t^*-1)\|^2 + \left(\frac{\alpha\beta}{2(1-\beta)\eta} - \frac{\alpha}{(1-\beta)\eta} + \frac{H_{s_0}\alpha^2\sigma_{max}^2}{2N}\right)\|(\boldsymbol{w}(t^*+1) - \boldsymbol{w}(t^*)\|^2$$

$$= \mathcal{L}(\boldsymbol{w}(t^*)) + \frac{\beta}{2(1-\beta)\eta}\|(\boldsymbol{w}(t^*) - \boldsymbol{w}(t^*-1)\|^2 - \frac{(1-\alpha)\beta}{2(1-\beta)\eta}\|\boldsymbol{w}(t^*) - \boldsymbol{w}(t^*-1)\|^2$$

$$+ \left(\frac{\alpha\beta}{2(1-\beta)\eta} - \frac{\alpha}{(1-\beta)\eta} + \frac{H_{s_0}\alpha^2\sigma_{max}^2}{2N}\right)\|(\boldsymbol{w}(t^*+1) - \boldsymbol{w}(t^*)\|^2$$

$$\overset{(\diamond)}{\leq} \mathcal{L}(\boldsymbol{w}(t^*)) + \frac{\beta}{2(1-\beta)\eta}\|\boldsymbol{w}(t^*) - \boldsymbol{w}(t^*-1)\|^2 - \frac{\beta}{2(1-\beta)\eta}\alpha^2\|\boldsymbol{w}(t^*+1) - \boldsymbol{w}(t^*)\|^2$$

$$- \frac{(1-C_1)\alpha^2}{\eta}\|\boldsymbol{w}(t^*+1) - \boldsymbol{w}(t^*)\|^2, \tag{22}$$

where Eq. $(*)$ is due to a simple rearrangement of the update rule of gradient descent with momentum (Eq. (1)), i.e.,

$$\nabla \mathcal{L}(\boldsymbol{w}(t)) = \frac{1}{(1-\beta)\eta}(\beta(\boldsymbol{w}(t) - \boldsymbol{w}(t-1)) - (\boldsymbol{w}(t+1) - \boldsymbol{w}(t))), \forall t \geq 1, \tag{23}$$

Inequality $(**)$ is due to Cauchy Schwarz's inequality and arithmetic-geometric average inequality, and Inequality $(\diamond)$ is due to

$$- \frac{(1-\alpha)\beta}{2(1-\beta)\eta}\|(\boldsymbol{w}(t^*) - \boldsymbol{w}(t^*-1)\|^2 + \left(\frac{\alpha\beta}{2(1-\beta)\eta} - \frac{\alpha}{(1-\beta)\eta} + \frac{H_{s_0}\alpha^2\sigma_{max}^2}{2N}\right)\|(\boldsymbol{w}(t^*+1) - \boldsymbol{w}(t^*)\|^2$$

$$= - \frac{(1-\alpha)\beta}{2(1-\beta)\eta}\|(\boldsymbol{w}(t^*) - \boldsymbol{w}(t^*-1)\|^2 + \mathcal{O}(\alpha)$$

$$\leq - \frac{\beta}{2(1-\beta)\eta}\alpha^2\|\boldsymbol{w}(t^*+1) - \boldsymbol{w}(t^*)\|^2 - \frac{(1-C_1)\alpha^2}{\eta}\|\boldsymbol{w}(t^*+1) - \boldsymbol{w}(t^*)\|^2.$$

Here the inequality is due to that $- \frac{(1-\alpha)\beta}{2\eta}\|(\boldsymbol{w}(t^*) - \boldsymbol{w}(t^*-1)\|^2$ tend to $- \frac{\beta}{2\eta}\|(\boldsymbol{w}(t^*) - \boldsymbol{w}(t^*-1)\|^2$ as $\alpha$ tend to zero, which is a negative constant by Lemma 9, and $- \frac{\beta}{2(1-\beta)\eta}\alpha^2\|\boldsymbol{w}(t^*+1) - \boldsymbol{w}(t^*)\|^2 - \frac{(1-C_1)\alpha^2}{\eta}\|\boldsymbol{w}(t^*+1) - \boldsymbol{w}(t^*)\|^2$ is $\mathcal{O}(\alpha^2)$.

Eq. (22) indicates Eq. (21) holds at $(t^*, \alpha)$ for $\alpha > 0$ is small enough, which contradicts to $\alpha^* = 0$.

**Case 2:** $\alpha^* \neq 0$: Same as **Case 1**, we have for any $1 \leq t < t^*$,

$$\mathcal{L}(\boldsymbol{w}(t)) + \frac{\beta}{2(1-\beta)\eta}\|\boldsymbol{w}(t) - \boldsymbol{w}(t-1)\|^2 \geq \mathcal{L}(\boldsymbol{w}(t+1)) + \frac{\beta}{2(1-\beta)\eta}\|\boldsymbol{w}(t+1) - \boldsymbol{w}(t)\|^2,$$

which further leads to

$$\mathcal{L}(\boldsymbol{w}(1)) \geq \mathcal{L}(\boldsymbol{w}(t^*)) + \frac{\beta}{2\eta}\|\boldsymbol{w}(t^*) - \boldsymbol{w}(t^*-1)\|^2. \tag{24}$$

On the other hand, by the definition of $\alpha^*$, we have for any $0 \leq \alpha < \alpha^*$, we have Eq. (21) holds for $(t^*, \alpha)$, which by continuity further leads to Eq. (21) holds for $(t^*, \alpha^*)$. Therefore, $\alpha^* < 1$, otherwise, Eq. (21) holds for $(t^*, \alpha), \forall \alpha \in [0, 1]$ which contradicts the definition of $t^*$.

Combining Eq. (21) with $(t^*, \alpha)$ and Eq. (24), we further have

$$\mathcal{L}(\boldsymbol{w}(1)) \geq \mathcal{L}(\boldsymbol{w}(t^* + \alpha)) + \frac{\beta}{2(1-\beta)\eta}\alpha^2\|\boldsymbol{w}(t+1) - \boldsymbol{w}(t)\|^2 + \frac{1-C_1}{2C_1}\alpha^2\|\boldsymbol{w}(t+1) - \boldsymbol{w}(t)\|^2,$$

Consequently, for any $\alpha \in [0, \alpha^*]$

$$\mathcal{L}(\boldsymbol{w}(1)) \geq \mathcal{L}(\boldsymbol{w}(t^* + \alpha)),$$

and by Lemma 6, we then have $\mathcal{L}$ is $\frac{H_{s_0}\sigma_{max}^2}{N}$ smooth at $\boldsymbol{w}(t^* + \alpha)$, which further by Taylor's expansion leads to

$$\mathcal{L}(\boldsymbol{w}(t^* + \alpha^*))$$

$$\leq \mathcal{L}(\boldsymbol{w}(t^*)) + \langle\nabla\mathcal{L}(\boldsymbol{w}(t^*)), \boldsymbol{w}(t^* + \alpha^*) - \boldsymbol{w}(t^*)\rangle + \frac{H_{s_0}\sigma_{max}^2}{2N}\|\boldsymbol{w}(t^* + \alpha^*) - \boldsymbol{w}(t^*)\|^2$$

$$\overset{(\circ)}{\leq} \mathcal{L}(\boldsymbol{w}(t^*)) + \frac{\alpha^*\beta}{2(1-\beta)\eta}\|\boldsymbol{w}(t^*) - \boldsymbol{w}(t^*-1)\|^2 + \frac{\alpha^*\beta}{2(1-\beta)\eta}\|\boldsymbol{w}(t^*+1) - \boldsymbol{w}(t^*)\|^2$$

$$+ \left(\frac{H_{s_0}(\alpha^*)^2\sigma_{max}^2}{2N} - \frac{\alpha^*}{(1-\beta)\eta}\right)\|\boldsymbol{w}(t^*+1) - \boldsymbol{w}(t^*)\|^2$$

$$= \mathcal{L}(\boldsymbol{w}(t^*)) + \frac{\alpha^*\beta}{2(1-\beta)\eta}\|\boldsymbol{w}(t^*) - \boldsymbol{w}(t^*-1)\|^2 + \left(\frac{H_{s_0}(\alpha^*)^2\sigma_{max}^2}{2N} - \frac{\alpha^*(2-\beta)}{2(1-\beta)\eta}\right)\|\boldsymbol{w}(t^*+1) - \boldsymbol{w}(t^*)\|^2$$

$$\overset{(\bullet)}{=} \mathcal{L}(\boldsymbol{w}(t^*)) + \frac{\alpha^*\beta}{2(1-\beta)\eta}\|\boldsymbol{w}(t^*) - \boldsymbol{w}(t^*-1)\|^2 + \left(\frac{C_1(\alpha^*)^2}{\eta} - \frac{\alpha^*(2-\beta)}{2(1-\beta)\eta}\right)\|\boldsymbol{w}(t^*+1) - \boldsymbol{w}(t^*)\|^2$$

$$\overset{(*)}{<} \mathcal{L}(\boldsymbol{w}(t^*)) + \frac{\beta}{2(1-\beta)\eta}\|\boldsymbol{w}(t^*) - \boldsymbol{w}(t^*-1)\|^2 - \frac{(\alpha^*)^2\beta}{2(1-\beta)\eta}\|\boldsymbol{w}(t^*+1) - \boldsymbol{w}(t^*)\|^2$$

$$- \frac{(1-C_1)(\alpha^*)^2}{\eta}\|\boldsymbol{w}(t^*+1) - \boldsymbol{w}(t^*)\|^2$$

where Eq. ($\circ$) follows the same routine as **Case 1**, Eq. ($\bullet$) is due to the definition of $\eta$ and $C_1$, and Eq. ($*$) is due to $\alpha^* < 1$, and $\|\boldsymbol{w}(t^*+1) - \boldsymbol{w}(t^*)\|^2 > 0$ (given by Lemma 9).

By the continuity of $\mathcal{L}$, for any small enough $\delta > 0$, Eq. (21) holds for $(t^*, \alpha^* + \delta)$, which contradicts to the definition of $\alpha^*$.

The proof is completed. $\qquad\qquad\qquad\qquad\qquad\qquad\qquad\qquad\qquad\qquad\qquad\qquad\qquad\quad$ $\square$

By Lemma 1, one can easily obtain the sum of the squared norms of the updates across the trajectory converges.

**Corollary 3.** *Let all conditions in Theorem 2 hold. We have*

$$\sum_{t=1}^{\infty}\|\boldsymbol{w}(t+1) - \boldsymbol{w}(t)\|^2 < \infty. \tag{25}$$

*Consequentially, we have*

$$\|\boldsymbol{w}(t)\| = \mathcal{O}(\sqrt{t}).$$

*Proof.* By Lemma 1, we have

$$\mathcal{L}(\boldsymbol{w}(t)) + \frac{\beta}{2(1-\beta)\eta}\|\boldsymbol{w}(t) - \boldsymbol{w}(t-1)\|^2 - \left(\mathcal{L}(\boldsymbol{w}(t+1)) + \frac{\beta}{2(1-\beta)\eta}\|\boldsymbol{w}(t+1) - \boldsymbol{w}(t)\|^2\right)$$
$$\geq \frac{1-C_1}{\eta}\|\boldsymbol{w}(t+1) - \boldsymbol{w}(t)\|^2,$$

which by summing over $t$ further leads to

$$\mathcal{L}(\boldsymbol{w}(1)) \geq \mathcal{L}(\boldsymbol{w}(1)) - \left(\mathcal{L}(\boldsymbol{w}(t+1)) + \frac{\beta}{2(1-\beta)\eta}\|\boldsymbol{w}(t+1) - \boldsymbol{w}(t)\|^2\right) \geq \frac{1-C_1}{\eta}\sum_{s=1}^{t}\|\boldsymbol{w}(s+1) - \boldsymbol{w}(s)\|^2.$$

Taking $t \to \infty$ leads to

$$\sum_{s=1}^{\infty}\|\boldsymbol{w}(s+1) - \boldsymbol{w}(s)\|^2 < \infty.$$

By triangle inequality, we further have

$$\|\boldsymbol{w}(t)\| \leq \sum_{s=1}^{t}\|\boldsymbol{w}(s+1) - \boldsymbol{w}(s)\| + \|\boldsymbol{w}(1)\|$$
$$\overset{(\star)}{\leq} \sqrt{t\left(\sum_{s=1}^{t}\|\boldsymbol{w}(s+1) - \boldsymbol{w}(s)\|^2\right)} + \|\boldsymbol{w}(1)\| = \mathcal{O}(\sqrt{t}),$$

where Eq. $(\star)$ is due to Cauchy-Schwartz's inequality.

The proof is completed. $\qquad\square$

By the negative derivative of the loss and the separable data, we can finally prove the sum of squared gradient converges.

**Corollary 4.** *Let all conditions in Theorem 2 hold. We have, $\sum_{t=1}^{\infty}\|\nabla\mathcal{L}(\boldsymbol{w}(t))\|^2 < \infty$.*

*Proof.* By Eq. (20), we have

$$\|\boldsymbol{w}(t+1) - \boldsymbol{w}(t)\|^2 = \eta^2\left\|(1-\beta)\sum_{s=1}^{t}\beta^{t-s}\frac{\sum_{i=1}^{N}\ell'(\langle\boldsymbol{w}(s),\boldsymbol{x}_i\rangle)\boldsymbol{x}_i}{N}\right\|^2$$

$$= \eta^2(1-\beta)^2\left\|\sum_{s=1}^{t}\beta^{t-s}\frac{\sum_{i=1}^{N}\ell'(\langle\boldsymbol{w}(s),\boldsymbol{x}_i\rangle)\boldsymbol{x}_i}{N}\right\|^2\frac{\|\hat{\boldsymbol{w}}\|^2}{\|\hat{\boldsymbol{w}}\|^2}$$

$$\overset{(*)}{\geq} \eta^2\gamma^2(1-\beta)\left\langle\hat{\boldsymbol{w}}, \sum_{s=1}^{t}\beta^{t-s}\frac{\sum_{i=1}^{N}\ell'(\langle\boldsymbol{w}(s),\boldsymbol{x}_i\rangle)\boldsymbol{x}_i}{N}\right\rangle^2$$

$$\overset{(**)}{\geq} \eta^2\gamma^2(1-\beta)^2\left(\sum_{s=1}^{t}\beta^{t-s}\frac{\sum_{i=1}^{N}\ell'(\langle\boldsymbol{w}(s),\boldsymbol{x}_i\rangle)}{N}\right)^2$$

$$\geq \eta^2\gamma^2(1-\beta)^2\left(\frac{\sum_{i=1}^{N}\ell'(\langle\boldsymbol{w}(t),\boldsymbol{x}_i\rangle)}{N}\right)^2$$

$$\overset{(\bullet)}{\geq} \frac{\eta^2\gamma^2(1-\beta)^2}{\sigma_{max}^2}\left\|\frac{\sum_{i=1}^{N}\ell'(\langle\boldsymbol{w}(t),\boldsymbol{x}_i\rangle)\boldsymbol{x}_i}{N}\right\|^2$$

$$= \frac{\eta^2\gamma^2(1-\beta)^2}{\sigma_{max}^2}\|\nabla\mathcal{L}(\boldsymbol{w}(t))\|^2, \tag{26}$$

where Inequality $(*)$ is due to Cauchy-Schwartz's inequality, Inequality $(**)$ is due to $\ell'(s) < 0$, $\forall s \in \mathbb{R}$ and $\langle\hat{\boldsymbol{w}}, \boldsymbol{x}_i\rangle \geq \gamma$, $\forall i \in [N]$, and Inequality $(\bullet)$ is due to the definition of $\sigma_{max}$. By combining Eq. (25) and Eq. (26), we complete the proof. $\qquad\square$

By the exponential-tailed assumption of the loss (Assumption 2), we further have the following corollary.

**Corollary 5.** *Let all conditions in Theorem 2 hold. Then, $\lim_{t\to\infty} \|\nabla\mathcal{L}(\boldsymbol{w}(t))\| = 0$, and*

$$\lim_{t\to\infty} \langle \boldsymbol{w}(t), \boldsymbol{x}_i \rangle = \infty, \forall i.$$

*Consequently, there exists an large enough time $t_0$, such that, $\forall t > t_0$, $\forall i$, we have $\langle \boldsymbol{w}(t), \boldsymbol{x}_i \rangle > 0$, and*

$$-\ell'(\langle \boldsymbol{w}(t), \boldsymbol{x}_i \rangle) \leq (1 + e^{-\mu_+ \langle \boldsymbol{w}(t), \boldsymbol{x}_i \rangle}) e^{-\langle \boldsymbol{w}(t), \boldsymbol{x}_i \rangle},$$
$$-\ell'(\langle \boldsymbol{w}(t), \boldsymbol{x}_i \rangle) \geq (1 - e^{-\mu_- \langle \boldsymbol{w}(t), \boldsymbol{x}_i \rangle}) e^{-\langle \boldsymbol{w}(t), \boldsymbol{x}_i \rangle}.$$

### C.1.2 Parameter dynamics

To prove Theorem 2, we only need to show $\boldsymbol{w}(t) - \ln(t)\hat{\boldsymbol{w}}$ ($t \geq 1$) has bounded norm for any iteration $t > 0$. Letting $C_3 = \frac{\eta}{N}$ in Corollary 1, we obtain an constant vector $\tilde{\boldsymbol{w}}$ satisfying Eq. (10). Define

$$\boldsymbol{r}(t) \triangleq \boldsymbol{w}(t) - \ln(t)\hat{\boldsymbol{w}} - \tilde{\boldsymbol{w}}. \tag{27}$$

As $\tilde{\boldsymbol{w}}$ is a constant vector, that $\boldsymbol{w}(t) - \ln(t)\hat{\boldsymbol{w}}$ ($t \geq 1$) has bounded norm is equivalent to $\boldsymbol{r}(t)$ has bounded norm. As discussed in the main body of the paper, we then propose an equivalent proposition of $\|\boldsymbol{r}(t)\|$ is bounded, and further prove this proposition is fulfilled. Specifically, we have

**Lemma 11.** *Let all conditions in Theorem 2 hold. Then, $\|\boldsymbol{r}(t)\|$ is bounded if and only if the function $g(t)$ is upper bounded, where $g : \mathbb{Z}^+ \to \mathbb{R}$ is defined as*

$$g(t) \triangleq \frac{1}{2}\|\boldsymbol{r}(t)\|^2 + \frac{\beta}{1-\beta}\langle \boldsymbol{r}(t), \boldsymbol{w}(t) - \boldsymbol{w}(t-1) \rangle - \frac{\beta}{1-\beta}\sum_{\tau=2}^{t}\langle \boldsymbol{r}(\tau) - \boldsymbol{r}(\tau-1), \boldsymbol{w}(\tau) - \boldsymbol{w}(\tau-1) \rangle. \tag{28}$$

*Furthermore, for almost every dataset, we have $\sum_{t=1}^{\infty}(g(t+1) - g(t))$ is upper bounded.*

As the proof is rather complex, we separate it into two sub-lemmas. We first prove $\|\boldsymbol{r}(t)\|$ is bounded if and only if function $g(t)$ is upper bounded.

**Lemma 12** (First argument in Lemma 11). *Let all conditions in Theorem 2 hold. Then, $\|\boldsymbol{r}(t)\|$ is bounded if and only if function $g(t)$ is upper bounded.*

*Proof.* We start the proof by showing that $A_1(t) \triangleq \sum_{\tau=2}^{t}\langle \boldsymbol{r}(\tau) - \boldsymbol{r}(\tau-1), \boldsymbol{w}(\tau) - \boldsymbol{w}(\tau-1) \rangle$ has bounded absolute value.

By the definition of $\boldsymbol{r}(t)$, we have

$$\boldsymbol{r}(t) - \boldsymbol{r}(t-1) = \boldsymbol{w}(t) - \boldsymbol{w}(t-1) - \ln\left(\frac{t}{t-1}\right)\hat{\boldsymbol{w}},$$

which further indicates

$$A_1(t) = \sum_{\tau=2}^{t}\left\langle \boldsymbol{w}(\tau) - \boldsymbol{w}(\tau-1) - \ln\left(\frac{\tau}{\tau-1}\right)\hat{\boldsymbol{w}}, \boldsymbol{w}(\tau) - \boldsymbol{w}(\tau-1) \right\rangle.$$

Therefore, the absolute value of $A_1(t)$ can be bounded as

$$
\begin{aligned}
|A_1(t)| &= \left| \sum_{\tau=2}^{t} \left\langle \boldsymbol{w}(\tau) - \boldsymbol{w}(\tau-1) - \ln\left(\frac{\tau}{\tau-1}\right)\hat{\boldsymbol{w}}, \boldsymbol{w}(\tau) - \boldsymbol{w}(\tau-1) \right\rangle \right| \\
&\leq \sum_{\tau=2}^{t} \left| \left\langle \boldsymbol{w}(\tau) - \boldsymbol{w}(\tau-1) - \ln\left(\frac{\tau}{\tau-1}\right)\hat{\boldsymbol{w}}, \boldsymbol{w}(\tau) - \boldsymbol{w}(\tau-1) \right\rangle \right| \\
&\leq \sum_{\tau=2}^{t} \|\boldsymbol{w}(\tau) - \boldsymbol{w}(\tau-1)\|^2 + \sum_{\tau=2}^{t} \left| \left\langle \ln\left(\frac{\tau}{\tau-1}\right)\hat{\boldsymbol{w}}, \boldsymbol{w}(\tau) - \boldsymbol{w}(\tau-1) \right\rangle \right| \\
&\leq \sum_{\tau=2}^{t} \|\boldsymbol{w}(\tau) - \boldsymbol{w}(\tau-1)\|^2 + \sum_{\tau=2}^{t} \left\| \ln\left(\frac{\tau}{\tau-1}\right)\hat{\boldsymbol{w}} \right\| \|\boldsymbol{w}(\tau) - \boldsymbol{w}(\tau-1)\| \\
&\overset{(\star)}{\leq} \frac{3}{2} \sum_{\tau=2}^{t} \|\boldsymbol{w}(\tau) - \boldsymbol{w}(\tau-1)\|^2 + \frac{1}{2}\sum_{\tau=2}^{t} \left\| \ln\left(\frac{\tau}{\tau-1}\right)\hat{\boldsymbol{w}} \right\|^2 \\
&\overset{(\circ)}{<} \infty,
\end{aligned}
$$

where Inequality $(\star)$ is due to the Inequality of arithmetic and geometric means, and Inequality $(\circ)$ is due to Corollary 3 and $\ln\frac{\tau}{\tau-1} = \boldsymbol{O}(\frac{1}{\tau})$.

Therefore, $g(t)$ is upper bounded is then equivalent to $\frac{1}{2}\|\boldsymbol{r}(t)\|^2 + \frac{\beta}{1-\beta}\langle \boldsymbol{r}(t), \boldsymbol{w}(t) - \boldsymbol{w}(t-1)\rangle$ is upper bounded. Now if $\frac{1}{2}\|\boldsymbol{r}(t)\|^2 + \frac{\beta}{1-\beta}\langle \boldsymbol{r}(t), \boldsymbol{w}(t) - \boldsymbol{w}(t-1)\rangle$ is upper bounded, we will prove $\|\boldsymbol{r}(t)\|$ is bounded by reduction to absurdity.

Suppose that $\|\boldsymbol{r}(t)\|$ has unbounded norm. By Corollary 3, we have $\lim_{t\to\infty}\|\boldsymbol{w}(t) - \boldsymbol{w}(t-1)\| = 0$, and there exists a large enough time $T$, such that $\|\boldsymbol{w}(t) - \boldsymbol{w}(t-1)\| < 1$ for any $t \geq T$. On the other hand, since $\boldsymbol{r}(t)$ is unbounded from above, there exists an increasing time sequence $k_i > T$, $i \in \mathbb{Z}^+$, such that

$$
\lim_{i\to\infty} \|\boldsymbol{r}(k_i)\| = \infty.
$$

Therefore, we have

$$
\lim_{i\to\infty} \frac{1}{2}\|\boldsymbol{r}(k_i)\|^2 + \frac{\beta}{1-\beta}\langle \boldsymbol{r}(k_i), \boldsymbol{w}(k_i) - \boldsymbol{w}(k_i-1)\rangle
$$

$$
\geq \lim_{i\to\infty} \frac{1}{2}\|\boldsymbol{r}(k_i)\|^2 - \frac{\beta}{1-\beta}\|\boldsymbol{r}(k_i)\| \|\boldsymbol{w}(k_i) - \boldsymbol{w}(k_i-1)\|
$$

$$
\geq \lim_{i\to\infty} \frac{1}{2}\|\boldsymbol{r}(k_i)\|^2 - \frac{\beta}{1-\beta}\|\boldsymbol{r}(k_i)\| = \infty,
$$

which leads to contradictory, and completes the proof of necessity.

On the other hand, if $\|\boldsymbol{r}(t)\|$ is upper bounded, since $\|\boldsymbol{w}(t) - \boldsymbol{w}(t-1)\|$ is also upper bounded, we have $\frac{1}{2}\|\boldsymbol{r}(t)\|^2 + \frac{\beta}{1-\beta}\langle \boldsymbol{r}(t), \boldsymbol{w}(t) - \boldsymbol{w}(t-1)\rangle$ is upper bounded, which completes the proof of sufficiency.

The proof is completed. $\qquad\square$

Therefore, the last piece of this puzzle is to prove $g(t)$ is upper bounded $\forall t > 0$.

**Lemma 13** (Second argument in Lemma 11). *Let all conditions in Theorem 2 hold. Then, for almost every dataset, we have that $g(t)$ is upper bounded.*

*Proof.* We start the proof by calculating $g(t+1) - g(t)$. For any $t \geq 2$, we have

$$g(t+1) - g(t) = \frac{1}{2}\|\boldsymbol{r}(t+1) - \boldsymbol{r}(t)\|^2 + \langle \boldsymbol{r}(t), \boldsymbol{r}(t+1) - \boldsymbol{r}(t)\rangle + \frac{\beta}{1-\beta}\langle \boldsymbol{r}(t+1), \boldsymbol{w}(t+1) - \boldsymbol{w}(t)\rangle$$

$$- \frac{\beta}{1-\beta}\langle \boldsymbol{r}(t), \boldsymbol{w}(t) - \boldsymbol{w}(t-1)\rangle - \frac{\beta}{1-\beta}\langle \boldsymbol{r}(t+1) - \boldsymbol{r}(t), \boldsymbol{w}(t+1) - \boldsymbol{w}(t)\rangle$$

$$= \frac{1}{2}\|\boldsymbol{r}(t+1) - \boldsymbol{r}(t)\|^2 + \langle \boldsymbol{r}(t), \boldsymbol{r}(t+1) - \boldsymbol{r}(t)\rangle + \frac{\beta}{1-\beta}\langle \boldsymbol{r}(t), \boldsymbol{w}(t+1) + \boldsymbol{w}(t-1) - 2\boldsymbol{w}(t)\rangle.$$

On the other hand, by simply rearranging the update rule Eq. (1), we have

$$\frac{\beta}{1-\beta}(\boldsymbol{w}(t+1) + \boldsymbol{w}(t-1) - 2\boldsymbol{w}(t)) = -\eta\nabla\mathcal{L}(\boldsymbol{w}(t)) - (\boldsymbol{w}(t+1) - \boldsymbol{w}(t)), \qquad (29)$$

which further indicates

$$g(t+1) - g(t)$$
$$= \frac{1}{2}\|\boldsymbol{r}(t+1) - \boldsymbol{r}(t)\|^2 + \langle \boldsymbol{r}(t), \boldsymbol{r}(t+1) - \boldsymbol{r}(t)\rangle + \langle \boldsymbol{r}(t), -\eta\nabla\mathcal{L}(\boldsymbol{w}(t)) - (\boldsymbol{w}(t+1) - \boldsymbol{w}(t))\rangle$$
$$= \frac{1}{2}\|\boldsymbol{r}(t+1) - \boldsymbol{r}(t)\|^2 + \left\langle \boldsymbol{r}(t), -\ln\left(\frac{t+1}{t}\right)\hat{\boldsymbol{w}} - \eta\nabla\mathcal{L}(\boldsymbol{w}(t))\right\rangle.$$

Denote $A_2(t) = \|\boldsymbol{r}(t+1) - \boldsymbol{r}(t)\|^2$, and $A_3(t) = \langle \boldsymbol{r}(t), -\ln\left(\frac{t+1}{t}\right)\hat{\boldsymbol{w}} - \eta\nabla\mathcal{L}(\boldsymbol{w}(t))\rangle$. We then prove respectively $\sum_{t=1}^{\infty} A_2(t)$ and $\sum_{t=1}^{\infty} A_3(t)$ are upper bounded.

First of all, by definition of $\boldsymbol{r}(t)$ Eq.(27), we have

$$\sum_{t=1}^{\infty} A_2(t) = \sum_{t=1}^{\infty}\left(\|\boldsymbol{w}(t+1) - \boldsymbol{w}(t)\|^2 + \ln\left(\frac{t+1}{t}\right)^2\|\hat{\boldsymbol{w}}\|^2 - 2\ln\left(\frac{t+1}{t}\right)\langle \boldsymbol{w}(t+1) - \boldsymbol{w}(t), \hat{\boldsymbol{w}}\rangle\right)$$

$$\leq 2\sum_{t=1}^{\infty}\left(\|\boldsymbol{w}(t+1) - \boldsymbol{w}(t)\|^2 + \ln\left(\frac{t+1}{t}\right)^2\|\hat{\boldsymbol{w}}\|^2\right) \overset{(\bullet)}{<} \infty, \qquad (30)$$

where Eq. ($\bullet$) is due to Lemma 3 and $\ln\left(\frac{t+1}{t}\right) = \mathcal{O}(\frac{1}{t})$.

Then we only need to prove $\sum_{t=1}^{\infty} A_3(t) < \infty$.

To begin with, by adding one additional term $\frac{1}{t}\hat{\boldsymbol{w}}$ into $A_3$, we have

$$A_3(t) = \left\langle \boldsymbol{r}(t), \frac{1}{t}\hat{\boldsymbol{w}} - \ln\left(\frac{t+1}{t}\right)\hat{\boldsymbol{w}}\right\rangle + \left\langle \boldsymbol{r}(t), -\frac{1}{t}\hat{\boldsymbol{w}} - \eta\nabla\mathcal{L}(\boldsymbol{w}(t))\right\rangle.$$

On the one hand, by Corollary 3, $\|\boldsymbol{w}(t)\| = \mathcal{O}(\sqrt{t})$, which further leads to

$$\|\boldsymbol{r}(t)\| = \|\boldsymbol{w}(t)\| + \ln(t)\|\hat{\boldsymbol{w}}\| + \|\hat{\boldsymbol{w}}\| = \mathcal{O}(\sqrt{t})$$

By $\frac{1}{t} - \ln\frac{t+1}{t} = \mathcal{O}\left(\frac{1}{t^2}\right)$, we have

$$\left\langle \boldsymbol{r}(t), \frac{1}{t}\hat{\boldsymbol{w}} - \ln\left(\frac{t+1}{t}\right)\hat{\boldsymbol{w}}\right\rangle = \mathcal{O}\left(\frac{1}{t^{\frac{3}{2}}}\right). \qquad (31)$$

On the other hand, by direct calculation of the gradient, we have

$$\left\langle \boldsymbol{r}(t), -\frac{1}{t}\hat{\boldsymbol{w}} - \eta\nabla\mathcal{L}(\boldsymbol{w}(t))\right\rangle$$

$$= \left\langle \boldsymbol{r}(t), -\frac{1}{t}\hat{\boldsymbol{w}} - \frac{\eta}{N}\sum_{i=1}^{N}\ell'(\langle \boldsymbol{w}(t), \boldsymbol{x}_i\rangle)\boldsymbol{x}_i\right\rangle$$

$$\overset{(\star)}{=} \frac{1}{N}\left\langle \boldsymbol{r}(t), -\frac{1}{t}\eta\sum_{\boldsymbol{x}_i \in S_s} e^{-\langle \tilde{\boldsymbol{w}}, \boldsymbol{x}_i\rangle}\boldsymbol{x}_i - \eta\sum_{i=1}^{N}\ell'(\langle \boldsymbol{w}, \boldsymbol{x}_i\rangle)\boldsymbol{x}_i\right\rangle$$

$$= \frac{1}{N}\left\langle \boldsymbol{r}(t), -\eta\sum_{\boldsymbol{x}_i \in S_s}\left(\frac{1}{t}e^{-\langle \tilde{\boldsymbol{w}}, \boldsymbol{x}_i\rangle} + \ell'(\langle \boldsymbol{w}(t), \boldsymbol{x}_i\rangle)\right)\boldsymbol{x}_i\right\rangle - \frac{1}{N}\left\langle \boldsymbol{r}(t), \eta\sum_{\boldsymbol{x}_i \notin S_s}\ell'(\langle \boldsymbol{w}(t), \boldsymbol{x}_i\rangle)\boldsymbol{x}_i\right\rangle,$$

where Eq. ($\star$) is due to the definition of $\tilde{\boldsymbol{w}}$ (Eq. (10) with $C_3 = \eta/N$).

Denote

$$A_4(t) = -\left\langle \boldsymbol{r}(t), \eta \sum_{\boldsymbol{x}_i \notin \boldsymbol{S}_s} \ell'(\langle \boldsymbol{w}(t), \boldsymbol{x}_i \rangle) \boldsymbol{x}_i \right\rangle,$$

and

$$A_5(t) = \left\langle \boldsymbol{r}(t), -\eta \sum_{\boldsymbol{x}_i \in \boldsymbol{S}_s} \left( \frac{1}{t} e^{-\langle \tilde{\boldsymbol{w}}, \boldsymbol{x}_i \rangle} + \ell'(\langle \boldsymbol{w}(t), \boldsymbol{x}_i \rangle) \right) \boldsymbol{x}_i \right\rangle.$$

We then analysis these two terms respectively. As for $A_4(t)$, due to $\ell' < 0$, we have

$$A_4(t) \leq -\eta \left\langle \boldsymbol{r}(t), \sum_{\boldsymbol{x}_i \notin \boldsymbol{S}_s, \langle \boldsymbol{r}(t), \boldsymbol{x}_i \rangle > 0} \ell'(\langle \boldsymbol{w}(t), \boldsymbol{x}_i \rangle) \boldsymbol{x}_i \right\rangle.$$

By Corollary 5, we further have $\forall t > t_0$

$$-\ell'(\langle \boldsymbol{w}(t), \boldsymbol{x}_i \rangle) \leq (1 + e^{-\mu_+ \langle \boldsymbol{w}(t), \boldsymbol{x}_i \rangle}) e^{-\langle \boldsymbol{w}(t), \boldsymbol{x}_i \rangle} \leq 2e^{-\langle \boldsymbol{w}(t), \boldsymbol{x}_i \rangle},$$

which further indicates

$$A_4(t) \leq -\eta \sum_{\boldsymbol{x}_i \notin \boldsymbol{S}_s, \langle \boldsymbol{r}(t), \boldsymbol{x}_i \rangle > 0} \ell'(\langle \boldsymbol{w}(t), \boldsymbol{x}_i \rangle) \langle \boldsymbol{r}(t), \boldsymbol{x}_i \rangle$$

$$\leq \eta \sum_{\boldsymbol{x}_i \notin \boldsymbol{S}_s, \langle \boldsymbol{r}(t), \boldsymbol{x}_i \rangle > 0} 2e^{-\langle \boldsymbol{w}(t), \boldsymbol{x}_i \rangle} \langle \boldsymbol{r}(t), \boldsymbol{x}_i \rangle$$

$$= \eta \sum_{\boldsymbol{x}_i \notin \boldsymbol{S}_s, \langle \boldsymbol{r}(t), \boldsymbol{x}_i \rangle > 0} 2e^{-\langle \boldsymbol{r}(t) + \ln t \hat{\boldsymbol{w}} + \tilde{\boldsymbol{w}}, \boldsymbol{x}_i \rangle} \langle \boldsymbol{r}(t), \boldsymbol{x}_i \rangle$$

$$\leq \eta \left( \max_i e^{\langle -\tilde{\boldsymbol{w}}, \boldsymbol{x}_i \rangle} \right) \sum_{\boldsymbol{x}_i \notin \boldsymbol{S}_s, \langle \boldsymbol{r}(t), \boldsymbol{x}_i \rangle > 0} 2e^{-\langle \boldsymbol{r}(t) + \ln t \hat{\boldsymbol{w}}, \boldsymbol{x}_i \rangle} \langle \boldsymbol{r}(t), \boldsymbol{x}_i \rangle$$

$$\overset{(\circ)}{\leq} \eta \frac{\left( \max_i e^{\langle -\tilde{\boldsymbol{w}}, \boldsymbol{x}_i \rangle} \right)}{t^\theta} \sum_{\boldsymbol{x}_i \notin \boldsymbol{S}_s, \langle \boldsymbol{r}(t), \boldsymbol{x}_i \rangle > 0} 2e^{-\langle \boldsymbol{r}(t), \boldsymbol{x}_i \rangle} \langle \boldsymbol{r}(t), \boldsymbol{x}_i \rangle$$

$$\overset{(\diamond)}{\leq} \eta \frac{\left( \max_i e^{\langle -\tilde{\boldsymbol{w}}, \boldsymbol{x}_i \rangle} \right)}{t^\theta} 2N,$$

where $\theta$ in Eq. ($\circ$) is defined as

$$\theta = \min_{\boldsymbol{x}_i \notin \boldsymbol{S}_s} \langle \boldsymbol{x}_i, \hat{\boldsymbol{w}} \rangle > 1. \tag{32}$$

As $\sum_{t=1}^{\infty} \frac{1}{t^\theta} < \infty$, we have

$$\sum_{t=1}^{\infty} A_4(t) < \infty^6. \tag{33}$$

For each term $\langle \boldsymbol{r}(t), -\eta \left( \frac{1}{t} e^{-\langle \tilde{\boldsymbol{w}}, \boldsymbol{x}_i \rangle} + \ell'(\langle \boldsymbol{w}, \boldsymbol{x}_i \rangle) \right) \boldsymbol{x}_i \rangle$ ($\boldsymbol{x}_i \notin \boldsymbol{S}_s$) in $A_5(t)$, we divide the analysis into two parts depending on the sign of $\langle \boldsymbol{r}(t), \boldsymbol{x}_i \rangle$.

**Case 1:** $\langle \boldsymbol{r}(t), \boldsymbol{x}_i \rangle \geq 0$. By Corollary 5, we have

$$\left\langle \boldsymbol{r}(t), -\eta \left( \frac{1}{t} e^{-\langle \tilde{\boldsymbol{w}}, \boldsymbol{x}_i \rangle} + \ell'(\langle \boldsymbol{w}, \boldsymbol{x}_i \rangle) \right) \boldsymbol{x}_i \right\rangle$$

$$= -\eta \left( \frac{1}{t} e^{-\langle \tilde{\boldsymbol{w}}, \boldsymbol{x}_i \rangle} + \ell'(\langle \boldsymbol{w}, \boldsymbol{x}_i \rangle) \right) \langle \boldsymbol{r}(t), \boldsymbol{x}_i \rangle$$

$$\leq \eta \left( -\frac{1}{t} e^{-\langle \tilde{\boldsymbol{w}}, \boldsymbol{x}_i \rangle} + (1 + e^{-\mu_+ \langle \boldsymbol{w}(t), \boldsymbol{x}_i \rangle}) e^{-\langle \boldsymbol{w}(t), \boldsymbol{x}_i \rangle} \right) \langle \boldsymbol{r}(t), \boldsymbol{x}_i \rangle$$

$$\overset{(\diamond)}{=} \eta \left( -\frac{1}{t} e^{-\langle \tilde{\boldsymbol{w}}, \boldsymbol{x}_i \rangle} + (1 + e^{-\mu_+ \langle \boldsymbol{r}(t) + \ln t \hat{\boldsymbol{w}} + \tilde{\boldsymbol{w}}, \boldsymbol{x}_i \rangle}) e^{-\langle \boldsymbol{r}(t) + \ln t \hat{\boldsymbol{w}} + \tilde{\boldsymbol{w}}, \boldsymbol{x}_i \rangle} \right) \langle \boldsymbol{r}(t), \boldsymbol{x}_i \rangle,$$

---

[6] In this paper, for a real series $\{r_i\}_{i=1}^{\infty}$, we use $\sum_{i=1}^{\infty} r_i < \infty$ representing $\sum_{i=1}^{T} r_i$ is uniformly upper bounded for any $T$.

where Eq. ($\diamond$) is due to the definition of $r(t)$ (Eq. (27)).

Since $\langle r(t), x_i \rangle \geq 0$, we further have

$$\left\langle r(t), -\eta \left( \frac{1}{t} e^{-\langle \tilde{w}, x_i \rangle} + \ell'(\langle w, x_i \rangle) \right) x_i \right\rangle$$

$$\leq \eta \left( -\frac{1}{t} e^{-\langle \tilde{w}, x_i \rangle} + (1 + e^{-\mu_+ \langle \ln t \hat{w} + \tilde{w}, x_i \rangle}) e^{-\langle r(t) + \ln t \hat{w} + \tilde{w}, x_i \rangle} \right) \langle r(t), x_i \rangle$$

$$\overset{(\square)}{=} \eta \left( -\frac{1}{t} e^{-\langle \tilde{w}, x_i \rangle} + \frac{1}{t} (1 + t^{-\mu_+} e^{-\mu_+ \langle \tilde{w}, x_i \rangle}) e^{-\langle r(t) + \tilde{w}, x_i \rangle} \right) \langle r(t), x_i \rangle$$

$$= \eta \frac{1}{t} e^{-\langle \tilde{w}, x_i \rangle} \left( -1 + (1 + t^{-\mu_+} e^{-\mu_+ \langle \tilde{w}, x_i \rangle}) e^{-\langle r(t), x_i \rangle} \right) \langle r(t), x_i \rangle,$$

where Eq. ($\square$) is due to $\langle \hat{w}, x_i \rangle = 1, \forall x_i \in S_s$.

Specifically,

$$-1 + (1 + t^{-\mu_+} e^{-\mu_+ \langle \tilde{w}, x_i \rangle}) e^{-\langle r(t), x_i \rangle}$$

$$= -1 + e^{-\langle r(t), x_i \rangle} + t^{-\mu_+} e^{-\mu_+ \langle \tilde{w}, x_i \rangle} e^{-\langle r(t), x_i \rangle}$$

$$\leq t^{-\mu_+} e^{-\mu_+ \langle \tilde{w}, x_i \rangle} e^{-\langle r(t), x_i \rangle}.$$

Therefore,

$$\eta \frac{1}{t} e^{-\langle \tilde{w}, x_i \rangle} \left( -1 + (1 + t^{-\mu_+} e^{-\mu_+ \langle \tilde{w}, x_i \rangle}) e^{-\langle r(t), x_i \rangle} \right) \langle r(t), x_i \rangle$$

$$\leq \eta \frac{1}{t} e^{-\langle \tilde{w}, x_i \rangle} \left( t^{-\mu_+} e^{-\mu_+ \langle \tilde{w}, x_i \rangle} e^{-\langle r(t), x_i \rangle} \right) \langle r(t), x_i \rangle$$

$$\leq \frac{\eta}{(1-\beta)e} \frac{1}{t^{1+\mu_+}} e^{-(1+\mu_+)\langle \tilde{w}, x_i \rangle} = \mathcal{O}\left( \frac{1}{t^{1+\mu_+}} \right).$$

**Case 2:** $\langle r(t), x_i \rangle < 0$. Similar to **Case 1.**, in this case we have

$$\left\langle r(t), -\eta \left( \frac{1}{t} e^{-\langle \tilde{w}, x_i \rangle} + \ell'(\langle w, x_i \rangle) \right) x_i \right\rangle$$

$$\leq \eta \left( -\frac{1}{t} e^{-\langle \tilde{w}, x_i \rangle} + \left( 1 - e^{-\mu_- \langle w(t), x_i \rangle} \right) e^{-\langle w(t), x_i \rangle} \right) \langle r(t), x_i \rangle$$

$$= \eta \left( -\frac{1}{t} e^{-\langle \tilde{w}, x_i \rangle} + \left( 1 - e^{-\mu_- \langle r(t) + \ln t \hat{w} + \tilde{w}, x_i \rangle} \right) e^{-\langle r(t) + \ln t \hat{w} + \tilde{w}, x_i \rangle} \right) \langle r(t), x_i \rangle$$

$$= \eta \frac{1}{t} e^{-\langle \tilde{w}, x_i \rangle} \left( -1 + \left( 1 - e^{-\mu_- \langle r(t) + \ln t \hat{w} + \tilde{w}, x_i \rangle} \right) e^{-\langle r(t), x_i \rangle} \right) \langle r(t), x_i \rangle.$$

Specifically, if $\langle r(t), x_i \rangle \geq -t^{-0.5\mu_-}$,

$$\left| \eta \frac{1}{t} e^{-\langle \tilde{w}, x_i \rangle} \left( -1 + (1 - e^{-\mu_- \langle r(t) + \ln t \hat{w} + \tilde{w}, x_i \rangle}) e^{-\langle r(t), x_i \rangle} \right) \langle r(t), x_i \rangle \right|$$

$$= \left| \eta \frac{1}{t} e^{-\langle \tilde{w}, x_i \rangle} \left( -1 + (1 - t^{-\mu_-} e^{-\mu_- \langle r(t) + \tilde{w}, x_i \rangle}) e^{-\langle r(t), x_i \rangle} \right) \langle r(t), x_i \rangle \right|$$

$$\leq \eta \frac{1}{t^{1+0.5\mu_-}} e^{-\langle \tilde{w}, x_i \rangle} \left| -1 + \left( 1 - t^{-\mu_-} e^{-\mu_- \langle r(t) + \tilde{w}, x_i \rangle} \right) e^{-\langle r(t), x_i \rangle} \right|$$

$$\overset{(\dagger)}{=} \mathcal{O}\left( \frac{1}{t^{1+0.5\mu_-}} \right),$$

where Eq. ($\dagger$) is due to if $\langle r(t), x_i \rangle \geq -t^{-0.5\mu_-}$,

$$\lim_{t \to \infty} \left| -1 + \left( 1 - t^{-\mu_-} e^{-\mu_- \langle r(t) + \tilde{w}, x_i \rangle} \right) e^{-\langle r(t), x_i \rangle} \right| = 0.$$

If $-2 \le \langle \boldsymbol{r}(t), \boldsymbol{x}_i \rangle < -t^{-0.5\mu_-}$, we have

$$\eta \frac{1}{t} e^{-\langle \tilde{\boldsymbol{w}}, \boldsymbol{x}_i \rangle} \left( -1 + \left( 1 - e^{-\mu_- \langle \boldsymbol{r}(t) + \ln t \hat{\boldsymbol{w}} + \tilde{\boldsymbol{w}}, \boldsymbol{x}_i \rangle} \right) e^{-\langle \boldsymbol{r}(t), \boldsymbol{x}_i \rangle} \right) \langle \boldsymbol{r}(t), \boldsymbol{x}_i \rangle$$

$$= \eta \frac{1}{t} e^{-\langle \tilde{\boldsymbol{w}}, \boldsymbol{x}_i \rangle} \left( -1 + \left( 1 - \frac{1}{t^{\mu_-}} e^{-\mu_- \langle \boldsymbol{r}(t) + \tilde{\boldsymbol{w}}, \boldsymbol{x}_i \rangle} \right) e^{-\langle \boldsymbol{r}(t), \boldsymbol{x}_i \rangle} \right) \langle \boldsymbol{r}(t), \boldsymbol{x}_i \rangle$$

$$\le \eta \frac{1}{t} e^{-\langle \tilde{\boldsymbol{w}}, \boldsymbol{x}_i \rangle} \left( -1 + \left( 1 - \frac{e^{2\mu_-}}{t^{\mu_-}} e^{-\mu_- \langle \tilde{\boldsymbol{w}}, \boldsymbol{x}_i \rangle} \right) e^{-\langle \boldsymbol{r}(t), \boldsymbol{x}_i \rangle} \right) \langle \boldsymbol{r}(t), \boldsymbol{x}_i \rangle .$$

Therefore, when $t$ is large enough, $1 - \frac{e^{2\mu_-}}{t^{\mu_-}} e^{-\mu_- \langle \tilde{\boldsymbol{w}}, \boldsymbol{x}_i \rangle} > 0$, which by $e^{-\langle \boldsymbol{r}(t), \boldsymbol{x}_i \rangle} \ge 1 - \langle \boldsymbol{r}(t), \boldsymbol{x}_i \rangle$ leads to

$$\eta \frac{1}{t} e^{-\langle \tilde{\boldsymbol{w}}, \boldsymbol{x}_i \rangle} \left( -1 + \left( 1 - \frac{e^{2\mu_-}}{t^{\mu_-}} e^{-\mu_- \langle \tilde{\boldsymbol{w}}, \boldsymbol{x}_i \rangle} \right) e^{-\langle \boldsymbol{r}(t), \boldsymbol{x}_i \rangle} \right) \langle \boldsymbol{r}(t), \boldsymbol{x}_i \rangle$$

$$\le \eta \frac{1}{t} e^{-\langle \tilde{\boldsymbol{w}}, \boldsymbol{x}_i \rangle} \left( -1 + \left( 1 - \frac{e^{2\mu_-}}{t^{\mu_-}} e^{-\mu_- \langle \tilde{\boldsymbol{w}}, \boldsymbol{x}_i \rangle} \right) (1 - \langle \boldsymbol{r}(t), \boldsymbol{x}_i \rangle) \right) \langle \boldsymbol{r}(t), \boldsymbol{x}_i \rangle$$

$$\le \eta \frac{1}{t} e^{-\langle \tilde{\boldsymbol{w}}, \boldsymbol{x}_i \rangle} \left( -1 + \left( 1 - \frac{e^{2\mu_-}}{t^{\mu_-}} e^{-\mu_- \langle \tilde{\boldsymbol{w}}, \boldsymbol{x}_i \rangle} \right) \left( 1 + \frac{1}{t^{0.5\mu_-}} \right) \right) \langle \boldsymbol{r}(t), \boldsymbol{x}_i \rangle$$

$$= \eta \frac{1}{t} e^{-\langle \tilde{\boldsymbol{w}}, \boldsymbol{x}_i \rangle} \left( \frac{1}{t^{0.5\mu_-}} + \boldsymbol{o} \left( \frac{1}{t^{0.5\mu_-}} \right) \right) \langle \boldsymbol{r}(t), \boldsymbol{x}_i \rangle < 0.$$

If $-2 > \langle \boldsymbol{r}(t), \boldsymbol{x}_i \rangle$,

$$\eta \frac{1}{t} e^{-\langle \tilde{\boldsymbol{w}}, \boldsymbol{x}_i \rangle} \left( -1 + \left( 1 - e^{-\mu_- \langle \boldsymbol{r}(t) + \ln t \hat{\boldsymbol{w}} + \tilde{\boldsymbol{w}}, \boldsymbol{x}_i \rangle} \right) e^{-\langle \boldsymbol{r}(t), \boldsymbol{x}_i \rangle} \right) \langle \boldsymbol{r}(t), \boldsymbol{x}_i \rangle$$

$$= \eta \frac{1}{t} e^{-\langle \tilde{\boldsymbol{w}}, \boldsymbol{x}_i \rangle} \left( -1 + \left( 1 - e^{-\mu_- \langle \boldsymbol{w}(t), \boldsymbol{x}_i \rangle} \right) e^{-\langle \boldsymbol{r}(t), \boldsymbol{x}_i \rangle} \right) \langle \boldsymbol{r}(t), \boldsymbol{x}_i \rangle .$$

For large enough $t$, $1 - e^{-\mu_- \langle \boldsymbol{w}(t), \boldsymbol{x}_i \rangle} > \frac{1}{2}$, and

$$\eta \frac{1}{t} e^{-\langle \tilde{\boldsymbol{w}}, \boldsymbol{x}_i \rangle} \left( -1 + \left( 1 - e^{-\mu_- \langle \boldsymbol{w}(t), \boldsymbol{x}_i \rangle} \right) e^{-\langle \boldsymbol{r}(t), \boldsymbol{x}_i \rangle} \right) \langle \boldsymbol{r}(t), \boldsymbol{x}_i \rangle$$

$$\le \eta \frac{1}{t} e^{-\langle \tilde{\boldsymbol{w}}, \boldsymbol{x}_i \rangle} \left( -1 + \left( 1 - e^{-\mu_- \langle \boldsymbol{w}(t), \boldsymbol{x}_i \rangle} \right) e^2 \right) \langle \boldsymbol{r}(t), \boldsymbol{x}_i \rangle$$

$$\le \eta \frac{1}{t} e^{-\langle \tilde{\boldsymbol{w}}, \boldsymbol{x}_i \rangle} \left( -1 + \frac{e^2}{2} \right) \langle \boldsymbol{r}(t), \boldsymbol{x}_i \rangle < 0.$$

Therefore, in **Case 2.**, for large enough $t$, we have

$$\left\langle \boldsymbol{r}(t), -\eta \left( \frac{1}{t} e^{-\langle \tilde{\boldsymbol{w}}, \boldsymbol{x}_i \rangle} + \ell'(\langle \boldsymbol{w}, \boldsymbol{x}_i \rangle) \right) \boldsymbol{x}_i \right\rangle \le \mathcal{O} \left( \frac{1}{t^{1+0.5\mu_-}} \right).$$

Combining **Case 1.** and **Case 2.**, we conclude that

$$A_5(t) \le \mathcal{O} \left( \frac{1}{t^{1+0.5\mu_+}} \right),$$

which further yields

$$\sum_{t=1}^{\infty} A_5(t) < \infty. \tag{34}$$

Combining Eq. (33) and Eq. (34), we conclude that $\sum_{t=1}^{\infty} A_3(t) < \infty$, which together with Eq. (30) yields $\sum_{t=2}^{\infty} g(t+1) - g(t) < \infty$, and completes the proof. $\square$

We are now ready to prove Theorem 2.

*Proof of Theorem 2.* By Lemma 13, we have $g(t)$ is upper bounded. Therefore, by Lemma 11, we have $\|\boldsymbol{r}(t)\|$ is bounded, which further indicates $\|\boldsymbol{w}(t) - \ln(t)\hat{\boldsymbol{w}}\|$ is bounded.

Therefore, the direction of $\boldsymbol{w}(t)$ can be calculated as

$$\frac{\boldsymbol{w}(t)}{\|\boldsymbol{w}(t)\|} = \frac{\ln(t)\hat{\boldsymbol{w}}}{\|\boldsymbol{w}(t)\|} + \frac{\boldsymbol{w}(t) - \ln(t)\hat{\boldsymbol{w}}}{\|\boldsymbol{w}(t)\|} = \frac{\ln(t)\hat{\boldsymbol{w}}}{\|\ln(t)\hat{\boldsymbol{w}} + \boldsymbol{w}(t) - \ln(t)\hat{\boldsymbol{w}}\|} + \frac{\boldsymbol{w}(t) - \ln(t)\hat{\boldsymbol{w}}}{\|\boldsymbol{w}(t)\|}$$

$$= \frac{\hat{\boldsymbol{w}}}{\left\|\hat{\boldsymbol{w}} + \frac{\boldsymbol{w}(t) - \ln(t)\hat{\boldsymbol{w}}}{\ln t}\right\|} + \frac{\boldsymbol{w}(t) - \ln(t)\hat{\boldsymbol{w}}}{\|\boldsymbol{w}(t)\|} \to \frac{\hat{\boldsymbol{w}}}{\|\hat{\boldsymbol{w}}\|} \ (as \ t \to \infty).$$

The proof is completed.

$\square$

## C.2 Implicit regularization of SGDM

This section collects the proof of Theorem 3. Following the same framework as Appendix C.1, we will first prove that the sum of the squared gradient norms along the trajectory is finite. One may expect $\mathcal{L}(\boldsymbol{w}(t)) + \frac{\beta}{2\eta}\|\boldsymbol{w}(t) - \boldsymbol{w}(t-1)\|^2$ is a Lyapunov function of SGDM. However, due to the randomness of the update rule of SGDM, $\mathcal{L}(\boldsymbol{w}(t)) + \frac{\beta}{2\eta}\|\boldsymbol{w}(t) - \boldsymbol{w}(t-1)\|^2$ may no longer decrease (we will show this in the end of Appendix C.2, please see Appendix C.2.3 for explanation).

### C.2.1 Loss dynamics

Recall that in the main text, we define $\boldsymbol{u}(t)$ as

$$\boldsymbol{u}(t) = \frac{\boldsymbol{w}(t) - \beta\boldsymbol{w}(t-1)}{1 - \beta}, \tag{35}$$

where the update of $\boldsymbol{u}(t)$ is given by $\boldsymbol{u}(t+1) = \boldsymbol{u}(t) - \eta\nabla\mathcal{L}_{\boldsymbol{B}(t)}(\boldsymbol{w}(t))$. We then prove Theorem 4.

*Proof of Theorem 4.* By the bounded smoothness assumption, we have that

$$\mathcal{L}(\boldsymbol{u}(t+1)) \leq \mathcal{L}(\boldsymbol{u}(t)) + \langle\boldsymbol{u}(t+1) - \boldsymbol{u}(t), \nabla\mathcal{L}(\boldsymbol{u}(t))\rangle + \frac{L}{2}\|\boldsymbol{u}(t+1) - \boldsymbol{u}(t)\|^2,$$

which by Eq. (35) leads to

$$\mathcal{L}(\boldsymbol{u}(t+1)) \leq \mathcal{L}(\boldsymbol{u}(t)) - \eta\langle\nabla\mathcal{L}_{\xi(t)}(\boldsymbol{w}(t)), \nabla\mathcal{L}(\boldsymbol{u}(t))\rangle + \frac{L\eta^2}{2}\|\nabla\mathcal{L}_{\xi(t)}(\boldsymbol{w}(t))\|^2. \tag{36}$$

Taking the expectation of Eq. (36) with respect to $\boldsymbol{w}(t+1)$ conditioning on $\mathcal{F}_t$ (recall that $\mathcal{F}_t$ is the sub-sigma algebra over the mini-batch sampling, such that $\forall t \in \mathbb{N}$, $\boldsymbol{w}(t)$ is adapted with respect to the sigma algebra flow $\mathcal{F}_t$), we have

$$\mathbb{E}[\mathcal{L}(\boldsymbol{u}(t+1))|\mathcal{F}_t]$$

$$\overset{(\star)}{=} \mathbb{E}_{\xi(t)}[\mathcal{L}(\boldsymbol{u}(t+1))]$$

$$\leq \mathbb{E}_{\xi(t)}\left[\mathcal{L}(\boldsymbol{u}(t)) - \eta\langle\nabla\mathcal{L}_{\xi(t)}(\boldsymbol{w}(t)), \nabla\mathcal{L}(\boldsymbol{u}(t))\rangle + \frac{L\eta^2}{2}\|\nabla\mathcal{L}_{\xi(t)}(\boldsymbol{w}(t))\|^2\right]$$

$$\overset{(\circ)}{=} \mathcal{L}(\boldsymbol{u}(t)) - \eta\langle\nabla\mathcal{L}(\boldsymbol{w}(t)), \nabla\mathcal{L}(\boldsymbol{u}(t))\rangle + \frac{L\eta^2}{2}\mathbb{E}_{\xi(t)}\left[\|\nabla\mathcal{L}_{\xi(t)}(\boldsymbol{w}(t))\|^2\right]$$

$$\overset{(\bullet)}{\leq} \mathcal{L}(\boldsymbol{u}(t)) - \eta\langle\nabla\mathcal{L}(\boldsymbol{w}(t)), \nabla\mathcal{L}(\boldsymbol{u}(t))\rangle + \frac{L\eta^2}{2}(\sigma_1\|\nabla\mathcal{L}(\boldsymbol{w}(t))\|^2 + \sigma_0), \tag{37}$$

where Eq. $(\star)$ is due to that $\boldsymbol{w}(t+1)$ is uniquely determined by $\boldsymbol{B}(t)$ given $\{\boldsymbol{w}(s)\}_{s=1}^t$, Eq. $(\circ)$ is due to $\boldsymbol{u}(t)$ is uniquely determined by $\{\boldsymbol{w}(s)\}_{s=1}^t$, and Inequality. $(\bullet)$ is due to the affine variance assumption .

Therefore, we have

$$\mathbb{E}[\mathcal{L}(\boldsymbol{u}(t+1))|\mathcal{F}_t]$$

$$\leq \mathcal{L}(\boldsymbol{u}(t)) - \eta\langle\nabla\mathcal{L}(\boldsymbol{w}(t)),\nabla\mathcal{L}(\boldsymbol{u}(t))\rangle + \frac{L\eta^2}{2}(\sigma_1\|\nabla\mathcal{L}(\boldsymbol{w}(t))\|^2 + \sigma_0)$$

$$= \mathcal{L}(\boldsymbol{u}(t)) - \eta\langle\nabla\mathcal{L}(\boldsymbol{w}(t)),\nabla\mathcal{L}(\boldsymbol{w}(t))\rangle + \eta\langle\nabla\mathcal{L}(\boldsymbol{w}(t)),\nabla\mathcal{L}(\boldsymbol{w}(t)) - \nabla\mathcal{L}(\boldsymbol{u}(t))\rangle$$

$$\quad + \frac{L\eta^2}{2}(\sigma_1\|\nabla\mathcal{L}(\boldsymbol{w}(t))\|^2 + \sigma_0)$$

$$= \mathcal{L}(\boldsymbol{u}(t)) - \eta\left(1 - \frac{L\eta\sigma_1}{2}\right)\|\nabla\mathcal{L}(\boldsymbol{w}(t))\|^2 + \langle\eta\nabla\mathcal{L}(\boldsymbol{w}(t)),\nabla\mathcal{L}(\boldsymbol{w}(t)) - \nabla\mathcal{L}(\boldsymbol{u}(t))\rangle + \frac{L\eta^2\sigma_0}{2}$$

$$\leq \mathcal{L}(\boldsymbol{u}(t)) - \eta\left(1 - \frac{L\eta\sigma_1}{2}\right)\|\nabla\mathcal{L}(\boldsymbol{w}(t))\|^2 + \frac{1}{2\lambda}\|\eta\nabla\mathcal{L}(\boldsymbol{w}(t))\|^2 + \frac{\lambda}{2}\|\nabla\mathcal{L}(\boldsymbol{w}(t)) - \nabla\mathcal{L}(\boldsymbol{u}(t))\|^2$$

$$\quad + \frac{L\eta^2\sigma_0}{2}$$

$$= \mathcal{L}(\boldsymbol{u}(t)) - \eta\left(1 - \left(\frac{1}{2\lambda} + \frac{L\sigma_1}{2}\right)\eta\right)\|\nabla\mathcal{L}(\boldsymbol{w}(t))\|^2 + \frac{\lambda}{2}\|\nabla\mathcal{L}(\boldsymbol{w}(t)) - \nabla\mathcal{L}(\boldsymbol{u}(t))\|^2 + \frac{L\eta^2\sigma_0}{2},$$

where $\lambda$ is a positive constant that will be specified latter.

Reusing the bounded smoothness assumption leads to

$$\|\nabla\mathcal{L}(\boldsymbol{w}(t)) - \nabla\mathcal{L}(\boldsymbol{u}(t))\|^2$$

$$\leq L^2\|\boldsymbol{w}(t) - \boldsymbol{u}(t)\|^2 \overset{(\square)}{=} \frac{\beta^2 L^2}{(1-\beta)^2}\|\boldsymbol{w}(t) - \boldsymbol{w}(t-1)\|^2$$

$$= \beta^2 L^2 \left\|\sum_{s=1}^{t-1}\eta\beta^{t-1-s}\nabla\mathcal{L}_{\xi(s)}(\boldsymbol{w}(s))\right\|^2 \overset{(\diamond)}{\leq} \beta^2 L^2\eta^2\left(\sum_{s=1}^{t-1}\beta^{t-1-s}\|\nabla\mathcal{L}_{\xi(s)}(\boldsymbol{w}(s))\|\right)^2$$

$$\overset{(\clubsuit)}{\leq} \beta^2 L^2\eta^2\left(\sum_{s=1}^{t-1}\beta^{t-1-s}\|\nabla\mathcal{L}_{\xi(s)}(\boldsymbol{w}(s))\|^2\right)\left(\sum_{s=1}^{t-1}\beta^{t-1-s}\right)$$

$$\leq \frac{\beta^2 L^2\eta^2}{1-\beta}\left(\sum_{s=1}^{t-1}\beta^{t-1-s}\|\nabla\mathcal{L}_{\xi(s)}(\boldsymbol{w}(s))\|^2\right), \tag{38}$$

where Inequality ($\square$) is due to $\beta(\boldsymbol{w}(t) - \boldsymbol{w}(t-1)) = (1-\beta)(\boldsymbol{u}(t) - \boldsymbol{w}(t))$ by Eq. (35), Inequality ($\diamond$) is due to triangular inequality, and Inequality ($\clubsuit$) is due to Cauchy-Schwartz Inequality.

Combining Eqs. (37) and (38), we have

$$\mathbb{E}[\mathcal{L}(\boldsymbol{u}(t+1))|\mathcal{F}_t]$$

$$\leq \mathcal{L}(\boldsymbol{u}(t)) - \eta\left(1 - \left(\frac{1}{2\lambda} + \frac{L\sigma_1}{2}\right)\eta\right)\|\nabla\mathcal{L}(\boldsymbol{w}(t))\|^2 + \frac{\lambda\beta^2 L^2\eta^2}{2(1-\beta)}\left(\sum_{s=1}^{t-1}\beta^{t-1-s}\|\nabla\mathcal{L}_{\boldsymbol{B}(s)}(\boldsymbol{w}(s))\|^2\right)$$

$$\quad + \frac{L\eta^2\sigma_0}{2},$$

which by taking expectation with respect to $\mathcal{F}_t$ leads to

$$\mathbb{E}[\mathcal{L}(\boldsymbol{u}(t+1))]$$

$$\leq \mathbb{E}\mathcal{L}(\boldsymbol{u}(t)) - \eta\left(1 - \left(\frac{1}{2\lambda} + \frac{L\sigma_1}{2}\right)\eta\right)\mathbb{E}\|\nabla\mathcal{L}(\boldsymbol{w}(t))\|^2 + \mathbb{E}\frac{\lambda\beta^2 L^2\eta^2}{2(1-\beta)}\left(\sum_{s=1}^{t-1}\beta^{t-1-s}\|\nabla\mathcal{L}_{\xi(s)}(\boldsymbol{w}(s))\|^2\right)$$

$$\quad + \frac{L\eta^2\sigma_0}{2}$$

$$\leq \mathbb{E}\mathcal{L}(\boldsymbol{u}(t)) - \eta\left(1 - \left(\frac{1}{2\lambda} + \frac{L\sigma_1}{2}\right)\eta\right)\mathbb{E}\|\nabla\mathcal{L}(\boldsymbol{w}(t))\|^2 + \frac{\lambda\beta^2 L^2\eta^2}{2(1-\beta)}\sigma_1\left(\sum_{s=1}^{t-1}\beta^{t-1-s}\mathbb{E}\|\nabla\mathcal{L}(\boldsymbol{w}(s))\|^2\right)$$

$$\quad + \frac{\lambda\beta^2 L^2\eta^2}{2(1-\beta)^2}\sigma_0 + \frac{L\eta^2\sigma_0}{2},$$

where the last inequality is due to the affine variance assumption. Letting $\lambda = \frac{1-\beta}{L\beta\sqrt{\sigma_1}}$ then leads to

$$\mathbb{E}[\mathcal{L}(\boldsymbol{u}(t+1))]$$

$$\leq \mathbb{E}\mathcal{L}(\boldsymbol{u}(t)) - \eta\left(1 - \left(\frac{L\beta\sqrt{\sigma_1}}{2(1-\beta)} + \frac{L\sigma_1}{2}\right)\eta\right)\mathbb{E}\|\nabla\mathcal{L}(\boldsymbol{w}(t))\|^2 + \frac{L\beta\eta^2}{2}\sqrt{\sigma_1}\left(\sum_{s=1}^{t-1}\beta^{t-1-s}\mathbb{E}\|\nabla\mathcal{L}(\boldsymbol{w}(s))\|^2\right)$$

$$+ \frac{\beta L\eta^2}{2(1-\beta)\sqrt{\sigma_1}}\sigma_0 + \frac{L\eta^2\sigma_0}{2}.$$

By the learning rate upper bound $\eta \leq \frac{1}{2\frac{L\beta\sqrt{\sigma_1}}{1-\beta} + L\sigma_1}$, summing the above inequality over $t$ then leads to

$$\mathbb{E}[\mathcal{L}(\boldsymbol{u}(T+1))]$$

$$\leq \mathcal{L}(\boldsymbol{u}(1)) - \frac{\eta}{2}\sum_{t=1}^{T}\mathbb{E}\|\nabla\mathcal{L}(\boldsymbol{w}(t))\|^2 + T\left(\frac{\beta L\eta^2}{2(1-\beta)\sqrt{\sigma_1}}\sigma_0 + \frac{L\eta^2\sigma_0}{2}\right).$$

The proof is completed. $\qquad\square$

As a direct corollary of Theorem 4 and Lemma 3, we have the following corollary.

**Corollary 6.** *Let all conditions in Theorem 3 hold. Then, we have*

$$\sum_{t=1}^{\infty}\mathbb{E}\|\nabla\mathcal{L}(\boldsymbol{w}(t))\|^2 < \infty. \tag{39}$$

*Consequently,*

$$\sum_{t=1}^{\infty}\|\nabla\mathcal{L}(\boldsymbol{w}(t))\|^2 < \infty$$

*and*

$$\langle\boldsymbol{w}(t), \boldsymbol{x}\rangle \to \infty, \forall \boldsymbol{x} \in \tilde{\boldsymbol{S}}$$

*hold almost surely.*

*Proof.* Eq. (39) directly follows from Theorem 4 and Lemma 3. The rest of claims follows immediately by Fubini's Theorem and Assumption 2.

The proof is completed. $\qquad\square$

### C.2.2 Parameter dynamics

Similar to the case of GDM, we define $\tilde{\boldsymbol{w}}$ as the solution of Eq. (10) with $C_3 = \frac{\eta}{(1-\beta)N}$. We also let $\boldsymbol{n}(t)$ be given by Lemma 5, and define $\boldsymbol{r}(t)$ in this case as

$$\boldsymbol{r}(t) \triangleq \boldsymbol{w}(t) - \ln(t)\hat{\boldsymbol{w}} - \tilde{\boldsymbol{w}} - \boldsymbol{n}(t). \tag{40}$$

As $\tilde{\boldsymbol{w}}$ is a constant vector, and $\|\boldsymbol{n}(t)\| \to 0$ as $t \to \infty$, we have $\boldsymbol{w}(t) - \ln(t)\hat{\boldsymbol{w}}$ has bounded norm if and only if $\|\boldsymbol{r}(t)\|$ is upper bounded. Similar to the GDM case, we have the following equivalent condition of that $\|\boldsymbol{r}(t)\|$ is bounded.

**Lemma 14.** *Let all conditions in Theorem 3 hold. Then, $\|\boldsymbol{r}(t)\|$ is bounded almost surely if and only if function $g(t)$ is upper bounded almost surely, where $g : \mathbb{Z}^+ \to \mathbb{R}$ is defined as*

$$g(t) \triangleq \frac{1}{2}\|\boldsymbol{r}(t)\|^2 + \frac{\beta}{1-\beta}\langle\boldsymbol{r}(t), \boldsymbol{w}(t) - \boldsymbol{w}(t-1)\rangle - \frac{\beta}{1-\beta}\sum_{\tau=2}^{t}\langle\boldsymbol{r}(\tau) - \boldsymbol{r}(\tau-1), \boldsymbol{w}(\tau) - \boldsymbol{w}(\tau-1)\rangle. \tag{41}$$

*Proof.* To begin with, we prove that almost surely $|\sum_{\tau=2}^{t}\langle\boldsymbol{r}(\tau) - \boldsymbol{r}(\tau-1), \boldsymbol{w}(\tau) - \boldsymbol{w}(\tau-1)\rangle|$ is upper bounded for any $t$. By Corollary 6, we have almost surly

$$\sum_{t=1}^{\infty}\|\nabla\mathcal{L}(\boldsymbol{w}(t))\|^2 < \infty.$$

On the other hand, for any $\boldsymbol{w}$, we have

$$\|\nabla\mathcal{L}_{\boldsymbol{B}(t)}(\boldsymbol{w})\| = \frac{1}{b}\left\|\sum_{\boldsymbol{x}\in\boldsymbol{B}(t)} \ell'(\langle\boldsymbol{w},\boldsymbol{x}\rangle)\boldsymbol{x}\right\|$$

$$\leq -\frac{\sigma_{max}}{b}\sum_{\boldsymbol{x}\in\boldsymbol{B}(t)} \ell'(\langle\boldsymbol{w},\boldsymbol{x}\rangle) < -\frac{\sigma_{max}}{b}\sum_{\boldsymbol{x}\in\boldsymbol{S}} \ell'(\langle\boldsymbol{w},\boldsymbol{x}\rangle)$$

$$\leq -\frac{\sigma_{max}}{b}\sum_{\boldsymbol{x}\in\boldsymbol{S}} \ell'(\langle\boldsymbol{w},\boldsymbol{x}\rangle)\langle\hat{\boldsymbol{w}},\boldsymbol{x}\rangle \leq \frac{N\sigma_{max}}{b}\left\|\frac{1}{N}\sum_{\boldsymbol{x}\in\boldsymbol{S}} \ell'(\langle\boldsymbol{w},\boldsymbol{x}\rangle)\boldsymbol{x}\right\|\|\hat{\boldsymbol{w}}\|$$

$$= \frac{N\sigma_{max}}{b\gamma}\|\nabla\mathcal{L}(\boldsymbol{w})\|.$$

Therefore, we have almost surely,

$$\sum_{t=1}^{\infty}\left\|\nabla\mathcal{L}_{\boldsymbol{B}(t)}(\boldsymbol{w}(t))\right\|^2 < \infty,$$

which further leads to almost surely

$$\sum_{t=1}^{\infty}\|\boldsymbol{w}(t+1) - \boldsymbol{w}(t)\|^2 \leq \eta^2(1-\beta)^2\sum_{t=1}^{\infty}\left\|\sum_{s=1}^{t}\beta^{t-s}\nabla\mathcal{L}_{\boldsymbol{B}(s)}(\boldsymbol{w}(s))\right\|^2$$

$$\leq \eta^2(1-\beta)^2\sum_{t=1}^{\infty}\left(\sum_{s=1}^{t}\beta^{t-s}\left\|\nabla\mathcal{L}_{\boldsymbol{B}(s)}(\boldsymbol{w}(s))\right\|\right)^2$$

$$\leq \eta^2(1-\beta)^2\sum_{t=1}^{\infty}\left(\sum_{s=1}^{t}\beta^{t-s}\left\|\nabla\mathcal{L}_{\boldsymbol{B}(s)}(\boldsymbol{w}(s))\right\|^2\right)\left(\sum_{s=1}^{t}\beta^{t-s}\right)$$

$$\leq \eta^2\sum_{s=1}^{\infty}\left\|\nabla\mathcal{L}_{\boldsymbol{B}(s)}(\boldsymbol{w}(s))\right\|^2 < \infty.$$

By the definition of $\boldsymbol{r}(t)$ (Eq. (40)), we further have

$$\left|\sum_{\tau=2}^{t}\langle\boldsymbol{r}(\tau) - \boldsymbol{r}(\tau-1), \boldsymbol{w}(\tau) - \boldsymbol{w}(\tau-1)\rangle\right|$$

$$\leq \sum_{\tau=2}^{t}|\langle\boldsymbol{r}(\tau) - \boldsymbol{r}(\tau-1), \boldsymbol{w}(\tau) - \boldsymbol{w}(\tau-1)\rangle|$$

$$= \sum_{\tau=2}^{t}\left|\left\langle\boldsymbol{w}(\tau) - \boldsymbol{w}(\tau-1) - \ln\left(\frac{\tau+1}{\tau}\right) - (\boldsymbol{n}(\tau) - \boldsymbol{n}(\tau-1)), \boldsymbol{w}(\tau) - \boldsymbol{w}(\tau-1)\right\rangle\right|$$

$$\leq \sum_{\tau=2}^{t}\|\boldsymbol{w}(\tau) - \boldsymbol{w}(\tau-1)\|^2 + \sum_{\tau=2}^{t}\left|\left\langle-\ln\left(\frac{\tau+1}{\tau}\right) - (\boldsymbol{n}(\tau) - \boldsymbol{n}(\tau-1)), \boldsymbol{w}(\tau) - \boldsymbol{w}(\tau-1)\right\rangle\right|$$

$$\leq \frac{3}{2}\sum_{\tau=2}^{t}\|\boldsymbol{w}(\tau) - \boldsymbol{w}(\tau-1)\|^2 + \frac{1}{2}\sum_{\tau=2}^{t}\left\|-\ln\left(\frac{\tau+1}{\tau}\right) - (\boldsymbol{n}(\tau) - \boldsymbol{n}(\tau-1))\right\|^2$$

$$\overset{(\star)}{\leq} \frac{3}{2}\sum_{\tau=2}^{t}\|\boldsymbol{w}(\tau) - \boldsymbol{w}(\tau-1)\|^2 + \frac{1}{2}\sum_{\tau=2}^{t}\mathcal{O}\left(\frac{1}{\tau}\right)^2 < \infty,$$

where Inequality $(\star)$ is due to $\|\boldsymbol{n}(\tau) - \boldsymbol{n}(\tau-1)\| = \mathcal{O}(\frac{1}{\tau})$ and $\ln\frac{\tau+1}{\tau} = \mathcal{O}(\frac{1}{\tau})$.

Therefore, $g(t)$ is upper bounded almost surely is equivalent to $\frac{1}{2}\|\boldsymbol{r}(t)\|^2 + \frac{\beta}{1-\beta}\langle\boldsymbol{r}(t), \boldsymbol{w}(t) - \boldsymbol{w}(t-1)\rangle$ is upper bounded, which can be shown to be equivalent with $\|\boldsymbol{r}(t)\|$ is bounded following the same routine as Lemma 11.

The proof is completed. $\qquad\square$

As the case of GDM, we only need to prove $g(t)$ is upper bounded to complete the proof of Theorem 3.

**Lemma 15.** *Let all conditions in Theorem 3 hold. Then, for almost every dataset, we have $g(t)$ is upper bounded.*

*Proof.* Following the same routine as Lemma 11, we have

$$g(t+1) - g(t)$$
$$= \frac{1}{2}\|\boldsymbol{r}(t+1) - \boldsymbol{r}(t)\|^2 + \langle \boldsymbol{r}(t), \boldsymbol{r}(t+1) - \boldsymbol{r}(t) \rangle + \langle \boldsymbol{r}(t), -\eta\nabla\mathcal{L}_{\boldsymbol{B}(t)}(\boldsymbol{w}(t)) - (\boldsymbol{w}(t+1) - \boldsymbol{w}(t)) \rangle,$$

where $\sum_{t=1}^{\infty}\|\boldsymbol{r}(t+1) - \boldsymbol{r}(t)\|^2$ is upper bounded.

On the other hand, by the definition of $\boldsymbol{r}(t)$ (Eq. (40)), we have

$$\boldsymbol{r}(t+1) - \boldsymbol{r}(t)$$
$$= \boldsymbol{w}(t+1) - \boldsymbol{w}(t) - \ln\left(\frac{t+1}{t}\right)\hat{\boldsymbol{w}} - \boldsymbol{n}(t+1) + \boldsymbol{n}(t),$$

while by Lemma 5,

$$\frac{N}{b}\frac{1}{t}\sum_{i:\boldsymbol{x}_i \in \boldsymbol{B}(t)\cap\boldsymbol{S}_s} \boldsymbol{v}_i\boldsymbol{x}_i = \ln\left(\frac{t+1}{t}\right)\hat{\boldsymbol{w}} + \boldsymbol{n}(t+1) - \boldsymbol{n}(t).$$

Combining the above two equations, we further have

$$\boldsymbol{r}(t+1) - \boldsymbol{r}(t) = \boldsymbol{w}(t+1) - \boldsymbol{w}(t) - \frac{N}{bt}\sum_{i:\boldsymbol{x}_i \in \boldsymbol{B}(t)\cap\boldsymbol{S}_s} \boldsymbol{v}_i\boldsymbol{x}_i,$$

which further indicates

$$g(t+1) - g(t) = \frac{1}{2}\|\boldsymbol{r}(t+1) - \boldsymbol{r}(t)\|^2 + \left\langle \boldsymbol{r}(t), -\eta\nabla\mathcal{L}_{\boldsymbol{B}(t)}(\boldsymbol{w}(t)) - \frac{N}{bt}\sum_{i:\boldsymbol{x}_i \in \boldsymbol{B}(t)\cap\boldsymbol{S}_s} \boldsymbol{v}_i\boldsymbol{x}_i \right\rangle.$$

Therefore, we only need to prove $\sum_{t=1}^{\infty}\langle \boldsymbol{r}(t), -\eta\nabla\mathcal{L}(\boldsymbol{w}(t)) - \frac{N}{bt}\sum_{i:\boldsymbol{x}_i \in \boldsymbol{B}(t)\cap\boldsymbol{S}_s} \boldsymbol{v}_i\boldsymbol{x}_i \rangle < \infty$. By directly applying the form of $\nabla\mathcal{L}(\boldsymbol{w}(t))$, we have

$$\left\langle \boldsymbol{r}(t), -\eta\nabla\mathcal{L}_{\boldsymbol{B}(t)}(\boldsymbol{w}(t)) - \frac{N}{bt}\sum_{i:\boldsymbol{x}_i \in \boldsymbol{B}(s)\cap\boldsymbol{S}_s} \boldsymbol{v}_i\boldsymbol{x}_i \right\rangle$$

$$= \left\langle \boldsymbol{r}(t), -\frac{\eta}{(1-\beta)b}\sum_{i:\boldsymbol{x}_i \in \boldsymbol{B}(s)} \ell'(\langle \boldsymbol{w}(t), \boldsymbol{x}_i \rangle)\boldsymbol{x}_i - \frac{N}{bt}\sum_{i:\boldsymbol{x}_i \in \boldsymbol{B}(s)\cap\boldsymbol{S}_s} \boldsymbol{v}_i\boldsymbol{x}_i \right\rangle$$

$$= \frac{\eta}{(1-\beta)b}\left\langle \boldsymbol{r}(t), -\sum_{i:\boldsymbol{x}_i \in \boldsymbol{B}(s)} \ell'(\langle \boldsymbol{w}(t), \boldsymbol{x}_i \rangle)\boldsymbol{x}_i - \frac{N(1-\beta)}{\eta}\frac{1}{t}\sum_{i:\boldsymbol{x}_i \in \boldsymbol{B}(s)\cap\boldsymbol{S}_s} \boldsymbol{v}_i\boldsymbol{x}_i \right\rangle$$

$$= \frac{\eta}{(1-\beta)b}\left\langle \boldsymbol{r}(t), -\sum_{i:\boldsymbol{x}_i \in \boldsymbol{B}(s)} \ell'(\langle \boldsymbol{w}(t), \boldsymbol{x}_i \rangle)\boldsymbol{x}_i - \frac{1}{t}\sum_{i:\boldsymbol{x}_i \in \boldsymbol{B}(s)\cap\boldsymbol{S}_s} e^{-\langle \tilde{\boldsymbol{w}}, \boldsymbol{x}_i \rangle}\boldsymbol{x}_i \right\rangle$$

$$= \frac{\eta}{(1-\beta)b}\sum_{i:\boldsymbol{x}_i \in \boldsymbol{B}(s)\cap\boldsymbol{S}_s} \left(-\ell'(\langle \boldsymbol{w}(t), \boldsymbol{x}_i \rangle) - \frac{1}{t}e^{-\langle \tilde{\boldsymbol{w}}, \boldsymbol{x}_i \rangle}\right)\langle \boldsymbol{r}(t), \boldsymbol{x}_i \rangle$$

$$+ \frac{\eta}{(1-\beta)b}\sum_{i:\boldsymbol{x}_i \in \boldsymbol{B}(s)\cap\boldsymbol{S}_s^c} \langle \boldsymbol{r}(t), -\ell'(\langle \boldsymbol{w}(t), \boldsymbol{x}_i \rangle)\boldsymbol{x}_i \rangle.$$

Let $A_6(t) = \sum_{i:\boldsymbol{x}_i \in \boldsymbol{B}(s)\cap\boldsymbol{S}_s} \left(-\ell'(\langle \boldsymbol{w}(t), \boldsymbol{x}_i \rangle) - \frac{1}{t}e^{-\langle \tilde{\boldsymbol{w}}, \boldsymbol{x}_i \rangle}\right)\langle \boldsymbol{r}(t), \boldsymbol{x}_i \rangle$, and $A_7(t) = \sum_{i:\boldsymbol{x}_i \in \boldsymbol{B}(s)\cap\boldsymbol{S}_s^c} \langle \boldsymbol{r}(t), -\ell'(\langle \boldsymbol{w}(t), \boldsymbol{x}_i \rangle)\boldsymbol{x}_i \rangle$. We will investigate these two terms respectively.

As $\langle w(t), x \rangle \to \infty$, $\forall x \in S$, a.s., we have a.s., there exists a large enough time $t_0$, s.t., $\forall t \geq t_0$, $\forall x \in S$,

$$-\ell'(\langle w(t), x \rangle) \leq (1 + e^{-\mu_+ \langle w(t), x \rangle}) e^{-\langle w(t), x \rangle},$$
$$-\ell'(\langle w(t), x_i \rangle) \geq (1 - e^{-\mu_- \langle w(t), x \rangle}) e^{-\langle w(t), x \rangle},$$
$$\langle x, w(t) \rangle > 0.$$

Therefore,

$$
\begin{aligned}
A_7(t) &\leq \sum_{i: x_i \in B(s) \cap S_s^c} -\ell'(\langle w(t), x_i \rangle) \langle r(t), x_i \rangle \mathbb{1}_{\langle r(t), x_i \rangle \geq 0} \\
&\leq \sum_{i: x_i \in B(s) \cap S_s^c} (1 + e^{-\mu_+ \langle w(t), x_i \rangle}) e^{-\langle w(t), x_i \rangle} \langle r(t), x_i \rangle \mathbb{1}_{\langle r(t), x_i \rangle \geq 0} \\
&\leq 2 \sum_{i: x_i \in B(s) \cap S_s^c} e^{-\langle r(t) + \ln(t)\hat{w} + \tilde{w} + n(t), x_i \rangle} \langle r(t), x_i \rangle \mathbb{1}_{\langle r(t), x_i \rangle \geq 0} \\
&\overset{(\star)}{\leq} 2 \sum_{i: x_i \in B(s) \cap S_s^c} \frac{1}{t^\theta} e^{-\langle \tilde{w} + n(t), x_i \rangle} e^{-\langle r(t), x_i \rangle} \langle r(t), x_i \rangle \mathbb{1}_{\langle r(t), x_i \rangle \geq 0} \\
&\overset{(\dagger)}{\leq} \frac{2}{e} \frac{1}{t^\theta} \sum_{i: x_i \in B(s) \cap S_s^c} e^{-\langle \tilde{w} + n(t), x_i \rangle} \mathbb{1}_{\langle r(t), x_i \rangle \geq 0} \\
&\overset{(\circ)}{=} \mathcal{O}\left(\frac{1}{t^\theta}\right),
\end{aligned}
$$

where Inequality. $(\star)$ is due the definition of $\theta$ (Eq. (32)), Inequality. $(\dagger)$ is due to $e^{-\langle r(t), x_i \rangle} \langle r(t), x_i \rangle \leq e^{-1}$, and Eq. $(\circ)$ is due to $\lim_{t \to \infty} e^{-\langle \tilde{w} + n(t), x_i \rangle} = e^{-\langle \tilde{w}, x_i \rangle}$. Thus,

$$\sum_{t=1}^{\infty} A_7(t) < \infty.$$

On the other hand, $A_6(t)$ can be rewritten as

$$
\begin{aligned}
A_6(t) = &\sum_{i: x_i \in B(s) \cap S_s} \left(-\ell'(\langle w(t), x_i \rangle) - \frac{1}{t} e^{-\langle \tilde{w}, x_i \rangle}\right) \langle r(t), x_i \rangle \mathbb{1}_{\langle r(t), x_i \rangle \geq 0} \\
&+ \sum_{i: x_i \in B(s) \cap S_s} \left(-\ell'(\langle w(t), x_i \rangle) - \frac{1}{t} e^{-\langle \tilde{w}, x_i \rangle}\right) \langle r(t), x_i \rangle \mathbb{1}_{\langle r(t), x_i \rangle < 0}.
\end{aligned}
$$

If $\langle \boldsymbol{r}(t), \boldsymbol{x}_i \rangle \geq 0$, we have for $\varepsilon < 0.5$,

$$\left( -\ell'(\langle \boldsymbol{w}(t), \boldsymbol{x}_i \rangle) - \frac{1}{t} e^{-\langle \tilde{\boldsymbol{w}}, \boldsymbol{x}_i \rangle} \right) \langle \boldsymbol{r}(t), \boldsymbol{x}_i \rangle$$

$$\leq \left( \left( 1 + e^{-\mu_+ \langle \boldsymbol{w}(t), \boldsymbol{x}_i \rangle} \right) e^{-\langle \boldsymbol{w}(t), \boldsymbol{x}_i \rangle} - \frac{1}{t} e^{-\langle \tilde{\boldsymbol{w}}, \boldsymbol{x}_i \rangle} \right) \langle \boldsymbol{r}(t), \boldsymbol{x}_i \rangle$$

$$= \left( \left( 1 + e^{-\mu_+ \langle \boldsymbol{r}(t) + \ln(t)\hat{\boldsymbol{w}} + \tilde{\boldsymbol{w}} + \boldsymbol{n}(t), \boldsymbol{x}_i \rangle} \right) e^{-\langle \boldsymbol{r}(t) + \ln(t)\hat{\boldsymbol{w}} + \tilde{\boldsymbol{w}} + \boldsymbol{n}(t), \boldsymbol{x}_i \rangle} - \frac{1}{t} e^{-\langle \tilde{\boldsymbol{w}}, \boldsymbol{x}_i \rangle} \right) \langle \boldsymbol{r}(t), \boldsymbol{x}_i \rangle$$

$$= \left( \left( 1 + e^{-\mu_+ \langle \boldsymbol{r}(t) + \ln(t)\hat{\boldsymbol{w}} + \tilde{\boldsymbol{w}} + \boldsymbol{n}(t), \boldsymbol{x}_i \rangle} \right) e^{-\langle \boldsymbol{r}(t) + \boldsymbol{n}(t), \boldsymbol{x}_i \rangle} - 1 \right) \frac{1}{t} \langle \boldsymbol{r}(t), \boldsymbol{x}_i \rangle e^{-\langle \tilde{\boldsymbol{w}}, \boldsymbol{x}_i \rangle}$$

$$\leq \left( \left( 1 + \frac{1}{t^{\mu_+}} e^{-\mu_+ \langle \tilde{\boldsymbol{w}} + \boldsymbol{n}(t), \boldsymbol{x}_i \rangle} \right) e^{-\langle \boldsymbol{r}(t) + \boldsymbol{n}(t), \boldsymbol{x}_i \rangle} - 1 \right) \frac{1}{t} \langle \boldsymbol{r}(t), \boldsymbol{x}_i \rangle e^{-\langle \tilde{\boldsymbol{w}}, \boldsymbol{x}_i \rangle}$$

$$\overset{(\bullet)}{=} \left( \left( 1 + \mathcal{O}\left( \frac{1}{t^{\mu_+}} \right) \right) \left( 1 + \mathcal{O}\left( \frac{1}{t^{0.5-\varepsilon}} \right) \right) e^{-\langle \boldsymbol{r}(t), \boldsymbol{x}_i \rangle} - 1 \right) \frac{1}{t} \langle \boldsymbol{r}(t), \boldsymbol{x}_i \rangle e^{-\langle \tilde{\boldsymbol{w}}, \boldsymbol{x}_i \rangle}$$

$$= \left( e^{-\langle \boldsymbol{r}(t), \boldsymbol{x}_i \rangle} - 1 \right) \frac{1}{t} \langle \boldsymbol{r}(t), \boldsymbol{x}_i \rangle e^{-\langle \tilde{\boldsymbol{w}}, \boldsymbol{x}_i \rangle} + \frac{1}{t} \mathcal{O}\left( \frac{1}{t^{\min\{\mu_+, 0.5-\varepsilon\}}} \right) e^{-\langle \boldsymbol{r}(t), \boldsymbol{x}_i \rangle} \langle \boldsymbol{r}(t), \boldsymbol{x}_i \rangle e^{-\langle \tilde{\boldsymbol{w}}, \boldsymbol{x}_i \rangle}$$

$$\leq \frac{1}{t} \mathcal{O}\left( \frac{1}{t^{\min\{\mu_+, 0.5-\varepsilon\}}} \right) e^{-\langle \boldsymbol{r}(t), \boldsymbol{x}_i \rangle} \langle \boldsymbol{r}(t), \boldsymbol{x}_i \rangle e^{-\langle \tilde{\boldsymbol{w}}, \boldsymbol{x}_i \rangle}$$

$$\overset{(\diamond)}{=} \mathcal{O}\left( \frac{1}{t^{\min\{1+\mu_+, 1.5-\varepsilon\}}} \right),$$

where Eq. $(\bullet)$ is due to $\boldsymbol{n}(t) = \mathcal{O}(\frac{1}{t^{0.5-\varepsilon}})$, and Eq. $(\diamond)$ is due to $e^{-\langle \boldsymbol{r}(t), \boldsymbol{x}_i \rangle} \langle \boldsymbol{r}(t), \boldsymbol{x}_i \rangle \leq \frac{1}{e}$.
On the other hand, if $\langle \boldsymbol{r}(t), \boldsymbol{x}_i \rangle < 0$, we have

$$\left( -\ell'(\langle \boldsymbol{w}(t), \boldsymbol{x}_i \rangle) - \frac{1}{t} e^{-\langle \tilde{\boldsymbol{w}}, \boldsymbol{x}_i \rangle} \right) \langle \boldsymbol{r}(t), \boldsymbol{x}_i \rangle$$

$$\leq \left( \left( 1 - e^{-\mu_- \langle \boldsymbol{w}(t), \boldsymbol{x}_i \rangle} \right) e^{-\langle \boldsymbol{w}(t), \boldsymbol{x}_i \rangle} - \frac{1}{t} e^{-\langle \tilde{\boldsymbol{w}}, \boldsymbol{x}_i \rangle} \right) \langle \boldsymbol{r}(t), \boldsymbol{x}_i \rangle$$

$$= \frac{1}{t} e^{-\langle \tilde{\boldsymbol{w}}, \boldsymbol{x}_i \rangle} \left( -1 + \left( 1 - e^{-\mu_- \langle \boldsymbol{w}(t), \boldsymbol{x}_i \rangle} \right) e^{-\langle \boldsymbol{r}(t) + \boldsymbol{n}(t), \boldsymbol{x}_i \rangle} \right) \langle \boldsymbol{r}(t), \boldsymbol{x}_i \rangle$$

Specifically, if $\langle \boldsymbol{r}(t), \boldsymbol{x}_i \rangle \geq -t^{-0.5 \min\{\mu_-, 0.5\}}$,

$$\left| \frac{1}{t} e^{-\langle \tilde{\boldsymbol{w}}, \boldsymbol{x}_i \rangle} \left( -1 + \left( 1 - e^{-\mu_- \langle \boldsymbol{w}(t), \boldsymbol{x}_i \rangle} \right) e^{-\langle \boldsymbol{r}(t) + \boldsymbol{n}(t), \boldsymbol{x}_i \rangle} \right) \langle \boldsymbol{r}(t), \boldsymbol{x}_i \rangle \right|$$

$$\leq \frac{1}{t^{1+0.5 \min\{\mu_-, 0.5\}}} e^{-\langle \tilde{\boldsymbol{w}}, \boldsymbol{x}_i \rangle} \left| -1 + \left( 1 - e^{-\mu_- \langle \boldsymbol{w}(t), \boldsymbol{x}_i \rangle} \right) e^{-\langle \boldsymbol{r}(t) + \boldsymbol{n}(t), \boldsymbol{x}_i \rangle} \right|$$

$$\overset{(\Box)}{=} \mathcal{O}\left( \frac{1}{t^{1+0.5 \min\{\mu_-, 0.5\}}} \right),$$

where Eq. $(\Box)$ is due to that as $\langle \boldsymbol{w}(t), \boldsymbol{x}_i \rangle \to \infty$ and $t^{-0.5 \min\{\mu_-, 0.5\}} \to 0$ as $t \to \infty$, there exists a large enough time $T$, s.t., $\forall t > T$, under the circumstance $0 > \langle \boldsymbol{r}(t), \boldsymbol{x}_i \rangle \geq -t^{-0.5 \min\{\mu_-, 0.5\}}$, $e^{-\langle \boldsymbol{r}(t) + \boldsymbol{n}(t), \boldsymbol{x}_i \rangle} < 1$ and $e^{-\mu_- \langle \boldsymbol{w}(t), \boldsymbol{x}_i \rangle} < 1$.

If $-2 \leq \langle \boldsymbol{r}(t), \boldsymbol{x}_i \rangle < -t^{-0.5\min\{\mu_-,0.5\}}$, then, for large enough $t$, $|\langle \boldsymbol{x}_i, \boldsymbol{n}(t) \rangle| < 2$, $1 - \frac{e^{\mu_-(-\langle \tilde{\boldsymbol{w}}, \boldsymbol{x}_i \rangle + 4)}}{t^{\mu_-}} > 0$, and

$$\frac{1}{t}e^{-\langle \tilde{\boldsymbol{w}}, \boldsymbol{x}_i \rangle} \left( -1 + \left( 1 - e^{-\mu_-\langle \boldsymbol{w}(t), \boldsymbol{x}_i \rangle} \right) e^{-\langle \boldsymbol{r}(t)+\boldsymbol{n}(t), \boldsymbol{x}_i \rangle} \right) \langle \boldsymbol{r}(t), \boldsymbol{x}_i \rangle$$

$$= \frac{1}{t}e^{-\langle \tilde{\boldsymbol{w}}, \boldsymbol{x}_i \rangle} \left( -1 + \left( 1 - e^{-\mu_-\langle \boldsymbol{r}(t)+\ln(t)\hat{\boldsymbol{w}}+\tilde{\boldsymbol{w}}+\boldsymbol{n}(t), \boldsymbol{x}_i \rangle} \right) e^{-\langle \boldsymbol{r}(t)+\boldsymbol{n}(t), \boldsymbol{x}_i \rangle} \right) \langle \boldsymbol{r}(t), \boldsymbol{x}_i \rangle$$

$$= \frac{1}{t}e^{-\langle \tilde{\boldsymbol{w}}, \boldsymbol{x}_i \rangle} \left( -1 + \left( 1 - \frac{e^{-\mu_-\langle \tilde{\boldsymbol{w}}, \boldsymbol{x}_i \rangle}}{t^{\mu_-}} e^{-\mu_-\langle \boldsymbol{r}(t)+\boldsymbol{n}(t), \boldsymbol{x}_i \rangle} \right) e^{-\langle \boldsymbol{r}(t)+\boldsymbol{n}(t), \boldsymbol{x}_i \rangle} \right) \langle \boldsymbol{r}(t), \boldsymbol{x}_i \rangle$$

$$\leq \frac{1}{t}e^{-\langle \tilde{\boldsymbol{w}}, \boldsymbol{x}_i \rangle} \left( -1 + \left( 1 - \frac{e^{\mu_-(-\langle \tilde{\boldsymbol{w}}, \boldsymbol{x}_i \rangle + 4)}}{t^{\mu_-}} \right) e^{-\langle \boldsymbol{r}(t)+\boldsymbol{n}(t), \boldsymbol{x}_i \rangle} \right) \langle \boldsymbol{r}(t), \boldsymbol{x}_i \rangle$$

$$\leq \frac{1}{t}e^{-\langle \tilde{\boldsymbol{w}}, \boldsymbol{x}_i \rangle} \left( -1 + \left( 1 - \frac{e^{\mu_-(-\langle \tilde{\boldsymbol{w}}, \boldsymbol{x}_i \rangle + 4)}}{t^{\mu_-}} \right) (1 - \langle \boldsymbol{r}(t) + \boldsymbol{n}(t), \boldsymbol{x}_i \rangle) \right) \langle \boldsymbol{r}(t), \boldsymbol{x}_i \rangle$$

$$\leq \frac{1}{t}e^{-\langle \tilde{\boldsymbol{w}}, \boldsymbol{x}_i \rangle} \left( -1 + \left( 1 - \frac{e^{\mu_-(-\langle \tilde{\boldsymbol{w}}, \boldsymbol{x}_i \rangle + 4)}}{t^{\mu_-}} \right) \left( 1 + t^{-0.5\min\{\mu_-,0.5\}} - \langle \boldsymbol{n}(t), \boldsymbol{x}_i \rangle \right) \right) \langle \boldsymbol{r}(t), \boldsymbol{x}_i \rangle$$

$$= \frac{1}{t}e^{-\langle \tilde{\boldsymbol{w}}, \boldsymbol{x}_i \rangle} \left( -1 + \left( 1 - \frac{e^{\mu_-(-\langle \tilde{\boldsymbol{w}}, \boldsymbol{x}_i \rangle + 4)}}{t^{\mu_-}} \right) \left( 1 + t^{-0.5\min\{\mu_-,0.5\}} + \boldsymbol{o}\left( t^{-0.5\min\{\mu_-,0.5\}} \right) \right) \right) \langle \boldsymbol{r}(t), \boldsymbol{x}_i \rangle$$

$$= \frac{1}{t}e^{-\langle \tilde{\boldsymbol{w}}, \boldsymbol{x}_i \rangle} \left( -1 + 1 + t^{-0.5\min\{\mu_-,0.5\}} + \boldsymbol{o}\left( t^{-0.5\min\{\mu_-,0.5\}} \right) \right) \langle \boldsymbol{r}(t), \boldsymbol{x}_i \rangle < 0.$$

If $-2 > \langle \boldsymbol{r}(t), \boldsymbol{x}_i \rangle$, then for large enough time $t$, $e^{-\langle \boldsymbol{r}(t)+\boldsymbol{n}(t), \boldsymbol{x}_i \rangle} \geq e^{\frac{3}{2}}$, $1 - e^{-\mu_-\langle \boldsymbol{w}(t), \boldsymbol{x}_i \rangle} \geq e^{-\frac{1}{2}}$, and

$$\frac{1}{t}e^{-\langle \tilde{\boldsymbol{w}}, \boldsymbol{x}_i \rangle} \left( -1 + \left( 1 - e^{-\mu_-\langle \boldsymbol{w}(t), \boldsymbol{x}_i \rangle} \right) e^{-\langle \boldsymbol{r}(t)+\boldsymbol{n}(t), \boldsymbol{x}_i \rangle} \right) \langle \boldsymbol{r}(t), \boldsymbol{x}_i \rangle$$

$$\leq \frac{1}{t}e^{-\langle \tilde{\boldsymbol{w}}, \boldsymbol{x}_i \rangle} (-1 + e) \langle \boldsymbol{r}(t), \boldsymbol{x}_i \rangle < 0.$$

Conclusively, if $\langle \boldsymbol{r}(t), \boldsymbol{x}_i \rangle < 0$, for large enough $t$, we have

$$\left( -\ell'(\langle \boldsymbol{w}(t), \boldsymbol{x}_i \rangle) - \frac{1}{t}e^{-\langle \tilde{\boldsymbol{w}}, \boldsymbol{x}_i \rangle} \right) \langle \boldsymbol{r}(t), \boldsymbol{x}_i \rangle \leq \mathcal{O}\left( \frac{1}{t^{1+0.5\min\{\mu_-,0.5\}}} \right),$$

which further indicates, for large enough $t$, we have

$$A_6(t) \leq \max \left\{ \mathcal{O}\left( \frac{1}{t^{1+0.5\min\{\mu_-,0.5\}}} \right), \mathcal{O}\left( \frac{1}{t^{\min\{1+\mu_+,1.5-\varepsilon\}}} \right) \right\},$$

which indicates

$$\sum_{t=1}^{\infty} A_6(t) < \infty.$$

Therefore,

$$\sum_{t=1}^{\infty} (g(t+1) - g(t))$$

$$= \sum_{t=1}^{\infty} \left( \frac{1}{2}\|\boldsymbol{r}(t+1) - \boldsymbol{r}(t)\|^2 + \left\langle \boldsymbol{r}(t), -\eta\nabla\mathcal{L}_{\boldsymbol{B}(t)}(\boldsymbol{w}(t)) - \frac{N}{bt} \sum_{i:\boldsymbol{x}_i \in \boldsymbol{B}(t) \cap \boldsymbol{S}_s} v_i \boldsymbol{x}_i \right\rangle \right)$$

$$= \sum_{t=1}^{\infty} \left( \frac{1}{2}\|\boldsymbol{r}(t+1) - \boldsymbol{r}(t)\|^2 + \eta A_6(t) + \eta A_7(t) \right)$$

$$< \infty.$$

The proof is completed. $\qquad\square$

### C.2.3 Explanation for proper lyapunov function

Based on the success of applying Lyapunov function $\mathcal{L}(\boldsymbol{w}(t)) + \frac{\beta}{2\eta}\|\boldsymbol{w}(t) - \boldsymbol{w}(t-1)\|^2$ to analyze gradient descent with momentum, it is natural to try to extend this routine to analyze stochastic gradient descent with momentum. However, in this section, we will show such Lyapunov function is not proper to analyze SGDM as this will put constraints on the range of the momentum rate $\beta$. Specifically, at any step $t$, since the loss $\mathcal{L}$ is $\frac{H\sigma_{max}^2}{N}$ smooth at $\boldsymbol{w}(t)$, we can expand the loss $\mathcal{L}$ in the same way as the GDM case:

$$\mathcal{L}(\boldsymbol{w}(t+1)) \leq \mathcal{L}(\boldsymbol{w}(t)) + \langle \boldsymbol{w}(t+1) - \boldsymbol{w}(t), \nabla\mathcal{L}(\boldsymbol{w}(t))\rangle + \frac{H\sigma_{max}^2}{2N}\|\boldsymbol{w}(t+1) - \boldsymbol{w}(t)\|^2.$$

By taking expectation with respect to $\boldsymbol{w}(t+1)$ conditioning on $\{\boldsymbol{w}(s)\}_{s=1}^t$ for both sides, we further obtain

$$\mathbb{E}\left[\mathcal{L}(\boldsymbol{w}(t+1))\,|\mathcal{F}_t\right]$$

$$\leq \mathcal{L}(\boldsymbol{w}(t)) + \langle \mathbb{E}\left[\boldsymbol{w}(t+1) - \boldsymbol{w}(t)\,|\mathcal{F}_t\right], \nabla\mathcal{L}(\boldsymbol{w}(t))\rangle + \frac{H\sigma_{max}^2}{2N}\mathbb{E}\left[\|\boldsymbol{w}(t+1) - \boldsymbol{w}(t)\|^2\,|\mathcal{F}_t\right]$$

$$\overset{(\star)}{=} \mathcal{L}(\boldsymbol{w}(t)) + \frac{1}{(1-\beta)\eta}\langle \mathbb{E}\left[\boldsymbol{w}(t+1) - \boldsymbol{w}(t)\,|\mathcal{F}_t\right], \beta\left(\boldsymbol{w}(t) - \boldsymbol{w}(t-1)\right) - \mathbb{E}\left[\boldsymbol{w}(t+1) - \boldsymbol{w}(t)\,|\mathcal{F}_t\right]\rangle$$

$$+ \frac{H\sigma_{max}^2}{2N}\mathbb{E}\left[\|\boldsymbol{w}(t+1) - \boldsymbol{w}(t)\|^2\,|\mathcal{F}_t\right]$$

$$= \mathcal{L}(\boldsymbol{w}(t)) + \frac{\beta}{(1-\beta)\eta}\langle\left(\boldsymbol{w}(t) - \boldsymbol{w}(t-1)\right), \mathbb{E}\left[\boldsymbol{w}(t+1) - \boldsymbol{w}(t)\,|\mathcal{F}_t\right]\rangle$$

$$+ \frac{H\sigma_{max}^2}{2N}\mathbb{E}\left[\|\boldsymbol{w}(t+1) - \boldsymbol{w}(t)\|^2\,|\mathcal{F}_t\right] - \frac{1}{(1-\beta)\eta}\|\mathbb{E}\left[\boldsymbol{w}(t+1) - \boldsymbol{w}(t)\,|\mathcal{F}_t\right]\|^2$$

$$\leq \mathcal{L}(\boldsymbol{w}(t)) + \frac{\beta}{2(1-\beta)\eta}\|\boldsymbol{w}(t) - \boldsymbol{w}(t-1)\|^2 + \frac{\beta}{2(1-\beta)\eta}\|\mathbb{E}\left[\boldsymbol{w}(t+1) - \boldsymbol{w}(t)\,|\mathcal{F}_t\right]\|^2$$

$$+ \frac{H\sigma_{max}^2}{2N}\mathbb{E}\left[\|\boldsymbol{w}(t+1) - \boldsymbol{w}(t)\|^2\,|\mathcal{F}_t\right] - \frac{1}{(1-\beta)\eta}\|\mathbb{E}\left[\boldsymbol{w}(t+1) - \boldsymbol{w}(t)\,|\mathcal{F}_t\right]\|^2,$$

where Eq. $(\star)$ is becasue $\mathbb{E}\left[\boldsymbol{w}(t+1) - \boldsymbol{w}(t)\,|\mathcal{F}_t\right] = -(1-\beta)\eta\nabla\mathcal{L}(\boldsymbol{w}(t)) + \beta\left(\boldsymbol{w}(t) - \boldsymbol{w}(t-1)\right)$ due to the definition of SGDM (Eq. (2)). Rearranging the above inequality and taking expectations of both sides with respect to $\{\boldsymbol{w}(s)\}_{s=1}^t$ leads to

$$\mathbb{E}\left[\mathcal{L}(\boldsymbol{w}(t+1))\right] + \frac{2-\beta}{2(1-\beta)\eta}\mathbb{E}\|\mathbb{E}\left[\boldsymbol{w}(t+1) - \boldsymbol{w}(t)\,|\mathcal{F}_t\right]\|^2$$

$$- \frac{H\sigma_{max}^2}{2N}\mathbb{E}\left[\|\boldsymbol{w}(t+1) - \boldsymbol{w}(t)\|^2\right]$$

$$\leq \mathbb{E}\mathcal{L}(\boldsymbol{w}(t)) + \frac{\beta}{2(1-\beta)\eta}\mathbb{E}\|\boldsymbol{w}(t) - \boldsymbol{w}(t-1)\|^2. \tag{42}$$

On the other hand, we wish to obtain some positive constant $\alpha$ from Eq. (42), such that (at least),

$$\mathbb{E}\left[\mathcal{L}(\boldsymbol{w}(t+1))\right] + \alpha\mathbb{E}\|\boldsymbol{w}(t+1) - \boldsymbol{w}(t)\|^2.$$

$$\leq \mathbb{E}\mathcal{L}(\boldsymbol{w}(t)) + \alpha\mathbb{E}\|\boldsymbol{w}(t) - \boldsymbol{w}(t-1)\|^2, \tag{43}$$

which requires to lower bound $\mathbb{E}\|\mathbb{E}\left[\boldsymbol{w}(t+1) - \boldsymbol{w}(t)\,|\mathcal{F}_t\right]\|^2$ by $\mathbb{E}\|\boldsymbol{w}(t+1) - \boldsymbol{w}(t)\|^2$. However, in general cases, $\mathbb{E}\|\mathbb{E}\left[\boldsymbol{w}(t+1) - \boldsymbol{w}(t)\,|\mathcal{F}_t\right]\|^2$ is only upper bounded by $\mathbb{E}\|\boldsymbol{w}(t+1) - \boldsymbol{w}(t)\|^2$ (Holder's Inequality), although in our case, $\|\mathbb{E}\left[\boldsymbol{w}(t+1) - \boldsymbol{w}(t)\,|\mathcal{F}_t\right]\|^2$ can be bounded as

$$\|\mathbb{E}\left[\boldsymbol{w}(t+1) - \boldsymbol{w}(t)\,|\mathcal{F}_t\right]\|^2$$

$$= \|-(1-\beta)\eta\nabla\mathcal{L}(\boldsymbol{w}(t)) + \beta\left(\boldsymbol{w}(t) - \boldsymbol{w}(t-1)\right)\|^2$$

$$= \|-(1-\beta)\eta\nabla\mathcal{L}(\boldsymbol{w}(t))\|^2 + \|\beta\left(\boldsymbol{w}(t) - \boldsymbol{w}(t-1)\right)\|^2 + 2\beta(1-\beta)\eta\langle\boldsymbol{w}(t) - \boldsymbol{w}(t-1), -\nabla\mathcal{L}(\boldsymbol{w}(t))\rangle,$$

while by the separability of the dataset and that the loss is non-increasing, $\mathbb{E}\left[\|\boldsymbol{w}(t+1) - \boldsymbol{w}(t)\|^2 \,|\mathcal{F}_t\right]$ can be bounded as

$$
\mathbb{E}\left[\|\boldsymbol{w}(t+1) - \boldsymbol{w}(t)\|^2 \,|\mathcal{F}_t\right]
$$
$$
\leq \frac{N\sigma_{max}^2}{b\gamma^2} \|\mathbb{E}\left[\boldsymbol{w}(t+1) - \boldsymbol{w}(t) \,|\mathcal{F}_t\right]\|^2 . \tag{44}
$$

By Eqs. (42) and (44), we have that to ensure Eq. (43), it is required that

$$
\frac{2-\beta}{2(1-\beta)\eta} \frac{b\gamma^2}{N\sigma_{max}^2} - \frac{H\sigma_{max}^2}{2N} \geq \frac{\beta}{2(1-\beta)\eta},
$$

which puts additional constraint on $\beta$ as

$$
\beta \leq \frac{2b\gamma^2 - H\eta\sigma_{max}^4}{b\gamma^2 + N\sigma_{max}^2 - H\sigma_{max}^4\eta}.
$$

Specifically, the upper bound becomes close to 0 when $N$ becomes large, and constrains $\beta$ in a small range.

## D Implicit regularization of deterministic Adam

This section collects the proof of the convergent direction of Adam, i.e., Theorem 5. The methodology of this section bears great similarity with GDM, although the preconditioner of Adam requires specific treatment for analysis. The proof is still divided into two stages: (1). we first prove the sum of squared gradients along the trajectory is finite. Additionally, we prove the convergent rate of loss is $\mathcal{O}(\frac{1}{t})$; (2). we prove $\boldsymbol{w}(t) - \ln(t)\hat{\boldsymbol{w}}$ has bounded norm. Before we present these two stages of proof, we will first give the required range of $\eta$ for which Theorem 3 holds. The analyses of this section hold for almost every dataset, and the "almost every" constraint is further moved in Section F.1.

### D.1 Choice of learning rate

Let $H_{s_0}$ be the smooth parameter over $[s_0, \infty)$ given by Assumption 3. (D). Let $\beta_2 = (c\beta_1)^4$ ($c > 1$). The "sufficiently small learning rate" in Theorem 3 means

$$
\eta \leq \frac{\sqrt{\varepsilon}\inf_{t\geq 2}\left(\frac{1-\beta_1^t}{1-\beta_1} - \frac{1-\beta_1^{t-1}}{c(1-\beta_1)}\frac{1-(c\beta_1)^t}{1-(c\beta_1)^{t-1}}\right)}{H_{\ell^{-1}((1-c\beta_1)^{-1}N\mathcal{L}(\boldsymbol{w}(1)))}}.
$$

To ensure $\eta$ is well-defined, we need to prove

$$
\inf_{t\geq 2}\left(\frac{1-\beta_1^t}{1-\beta_1} - \frac{1-\beta_1^{t-1}}{c(1-\beta_1)}\frac{1-(c\beta_1)^t}{1-(c\beta_1)^{t-1}}\right) > 0,
$$

and we introduce the following technical lemma:

**Lemma 16.** *Define* $f_t(x) = \frac{1-x^t}{x(1-x^{t-1})}$, $\forall t \in \mathbb{Z}, t \geq 2$. *We have* $f_t(x)$ *is decreasing with respect to* $x$. *Furthermore, for any* $x \in [0, 1)$, *we have*

$$
f(x) \geq \sqrt[4]{f(x^4)}. \tag{45}
$$

*Proof.* First of all, by definition,

$$
f(x) = \frac{1-x^t}{x-x^t} = 1 + \frac{1-x}{x-x^t} = 1 + \frac{1-x}{x(1-x^{t-1})} = 1 + \frac{1}{x(1+x+\cdots+x^{t-2})}
$$

is monotonously decreasing as $0 \leq x < 1$. Secondly, Eq. (45) is equivalent to

$$
\frac{(1-x^t)^4}{\beta_1^4(1-x^{t-1})^4} \geq \frac{(1-x^{4t})}{x^4(1-x^{4(t-1)})}
$$
$$
\Longleftrightarrow \frac{(1-x^t)^3}{(1-x^{t-1})^3} \geq \frac{(1+x^t)(1+x^{2t})}{(1+x^{t-1})(1+x^{2(t-1)})}.
$$

The left side of the above inequality is no smaller than 1, while the right side is no larger than 1, which completes the proof. □

We are now ready to prove $\eta$ is well-defined. First of all, for every $t$, we have

$$\frac{1-\beta_1^t}{1-\beta_1} - \frac{1-\beta_1^{t-1}}{c(1-\beta_1)}\frac{1-(c\beta_1)^t}{1-(c\beta_1)^{t-1}}$$

$$=\frac{\beta_1(1-\beta_1^{t-1})}{1-\beta_1}\left(\frac{1-\beta_1^t}{\beta_1(1-\beta_1^{t-1})} - \frac{1-(c\beta_1)^t}{(c\beta_1)(1-(c\beta_1)^{t-1})}\right)$$

$$\overset{(\star)}{=}\frac{\beta_1(1-\beta_1^{t-1})}{1-\beta_1}\left(f_t(\beta_1) - f_t(c\beta_1)\right) > 0, \tag{46}$$

where Eq. $(\star)$ is by Lemma 16 and $c\beta_1 = \sqrt[4]{\beta_2} < 1$.

On the other hand, we have

$$\lim_{t\to\infty}\left(\frac{1-\beta_1^t}{1-\beta_1} - \frac{1-\beta_1^{t-1}}{c(1-\beta_1)}\frac{1-(c\beta_1)^t}{1-(c\beta_1)^{t-1}}\right) = \left(1 - \frac{1}{c}\right)\frac{1}{1-\beta_1}. \tag{47}$$

By Eq. (46) and Eq. (47), we obtain $\frac{1-\beta_1^t}{1-\beta_1} - \frac{1-\beta_1^{t-1}}{c(1-\beta_1)}\frac{1-(c\beta_1)^t}{1-(c\beta_1)^{t-1}}$ is lower bounded by some positive constant across $t$, and $\eta$ is well defined.

## D.2 Sum of gradients along the trajectory is bounded

We start with the following lemma, which indicates $\mathcal{L}(\boldsymbol{w}(t)) + \|\sqrt[4]{\varepsilon\mathbb{1}_d + \hat{\boldsymbol{\nu}}(t)}\odot(\boldsymbol{w}(t)-\boldsymbol{w}(t-1))\|^2$ is a proper Lyapunov function for Adam.

**Lemma 17.** *Let all conditions in Theorem 5 hold. Then, for any $t \geq 1$,*

$$\mathcal{L}(t+1) + \frac{1}{2}\frac{1-\beta_1^t}{\eta(1-\beta_1)}\left\|\sqrt[4]{\varepsilon\mathbb{1}_d + \hat{\boldsymbol{\nu}}(t)}\odot(\boldsymbol{w}(t+1)-\boldsymbol{w}(t))\right\|^2$$

$$\leq\mathcal{L}(\boldsymbol{w}(t)) + \left\|\sqrt[4]{\varepsilon\mathbb{1}_d + \hat{\boldsymbol{\nu}}(t-1)}\odot(\boldsymbol{w}(t)-\boldsymbol{w}(t-1))\right\|^2\frac{1-\beta_1^{t-1}}{2c\eta(1-\beta_1)}\frac{1-(c\beta_1)^t}{1-(c\beta_1)^{t-1}}. \tag{48}$$

*Proof.* We start with the case $t = 1$. To begin with, we have $\mathcal{L}$ is $H_{\ell^{-1}(N\mathcal{L}(\boldsymbol{w}(1)))}$ smooth around $\boldsymbol{w}(1)$. By definition $H_x$ is non-increasing with respect to $x$, and since $\ell^{-1}$ is also non-increasing, we have

$$H_{\ell^{-1}(N\mathcal{L}(\boldsymbol{w}(1)))} \leq H_{\ell^{-1}(\frac{1}{1-c\beta_1}N\mathcal{L}(\boldsymbol{w}(1)))},$$

which further indicates when $\alpha$ is small enough,

$$\mathcal{L}(\boldsymbol{w}(1+\alpha)) \overset{(\star)}{\leq}\mathcal{L}(\boldsymbol{w}(1)) + \alpha\langle\nabla\mathcal{L}(\boldsymbol{w}(1)), \boldsymbol{w}(2)-\boldsymbol{w}(1)\rangle + \frac{L}{2}\alpha^2\|\boldsymbol{w}(2)-\boldsymbol{w}(1)\|$$

$$=\mathcal{L}(\boldsymbol{w}(1)) - \alpha\left\langle\nabla\mathcal{L}(\boldsymbol{w}(1)), \eta\frac{1}{\sqrt{\varepsilon\mathbb{1}_d + \hat{\boldsymbol{\nu}}(1)}}\odot\nabla\mathcal{L}(\boldsymbol{w}(1))\right\rangle + \boldsymbol{o}(\alpha^2)$$

$$\leq\mathcal{L}(\boldsymbol{w}(1)) - \frac{1}{2\eta}\alpha^2\left\|\sqrt[4]{\varepsilon\mathbb{1}_d + \hat{\boldsymbol{\nu}}(1)}\odot(\boldsymbol{w}(2)-\boldsymbol{w}(t))\right\|^2,$$

where in Eq. $(\star)$ we denote $L \overset{\triangle}{=} H_{\ell^{-1}(\frac{1}{1-c\beta_1}N\mathcal{L}(\boldsymbol{w}(1)))}$, and the last inequality is due to $\frac{1}{2\eta}\alpha^2$ $\left\|\sqrt[4]{\varepsilon\mathbb{1}_d + \hat{\boldsymbol{\nu}}(1)}\odot(\boldsymbol{w}(2)-\boldsymbol{w}(t))\right\|^2 = \boldsymbol{o}(\alpha^2)$, and $\left\langle\nabla\mathcal{L}(\boldsymbol{w}(1)), \eta\frac{1}{\sqrt{\varepsilon\mathbb{1}_d+\hat{\boldsymbol{\nu}}(1)}}\odot\nabla\mathcal{L}(\boldsymbol{w}(1))\right\rangle$ is positive.

Now if there exists an $\alpha \in (0,1)$, such that Eq. (48) fails, we denote $\alpha^* = \inf\{\alpha : Eq.(48)\ fails\ for\ 1+\alpha\}$. We have $\alpha^* > 0$, and the equality in Eq. (48) holds for $1+\alpha^*$. Therefore, we have for any $\alpha \in (0,\alpha^*)$,

$$\mathcal{L}(\boldsymbol{w}(1+\alpha)) \leq \mathcal{L}(\boldsymbol{w}(1+\alpha)) + \frac{1}{2\eta}\alpha^2\left\|\sqrt[4]{\varepsilon\mathbb{1}_d + \hat{\boldsymbol{\nu}}(1)}\odot(\boldsymbol{w}(2)-\boldsymbol{w}(t))\right\|^2 \leq \mathcal{L}(\boldsymbol{w}(1)),$$

which by Lemma 6 leads to $\mathcal{L}$ is $H_{\ell^{-1}(N\mathcal{L}(\boldsymbol{w}(1)))}$ smooth (thus $L$ smooth) over the set $\{\boldsymbol{w}(1+\alpha) : \alpha \in [0, \alpha^*]\}$, and

$$\mathcal{L}(\boldsymbol{w}(1+\alpha^*))$$

$$\leq \mathcal{L}(\boldsymbol{w}(1)) + \alpha^* \langle \nabla \mathcal{L}(\boldsymbol{w}(1)), \boldsymbol{w}(2) - \boldsymbol{w}(1) \rangle + \frac{L}{2}(\alpha^*)^2 \|\boldsymbol{w}(2) - \boldsymbol{w}(1)\|^2$$

$$= \mathcal{L}(\boldsymbol{w}(1)) - \alpha^* \left\langle \frac{1}{\eta} \sqrt{\varepsilon \mathbb{1}_d + \hat{\boldsymbol{\nu}}(1)} \odot (\boldsymbol{w}(2) - \boldsymbol{w}(1)), \boldsymbol{w}(2) - \boldsymbol{w}(1) \right\rangle + \frac{L}{2}(\alpha^*)^2 \|\boldsymbol{w}(2) - \boldsymbol{w}(1)\|^2$$

$$= \mathcal{L}(\boldsymbol{w}(1)) - \alpha^* \frac{1}{\eta} \left\| \sqrt[4]{\varepsilon \mathbb{1}_d + \hat{\boldsymbol{\nu}}(1)} \odot (\boldsymbol{w}(2) - \boldsymbol{w}(1)) \right\|^2 + \frac{L}{2}(\alpha^*)^2 \left\| \frac{1}{\sqrt[4]{\varepsilon \mathbb{1}_d + \hat{\boldsymbol{\nu}}(1)}} \odot \sqrt[4]{\varepsilon \mathbb{1}_d + \hat{\boldsymbol{\nu}}(1)} \odot (\boldsymbol{w}(2) - \boldsymbol{w}(1)) \right\|^2$$

$$\leq \mathcal{L}(\boldsymbol{w}(1)) - \alpha^* \frac{1}{\eta} \left\| \sqrt[4]{\varepsilon \mathbb{1}_d + \hat{\boldsymbol{\nu}}(1)} \odot (\boldsymbol{w}(2) - \boldsymbol{w}(1)) \right\|^2 + \frac{L}{2\sqrt{\varepsilon}}(\alpha^*)^2 \left\| \sqrt[4]{\varepsilon \mathbb{1}_d + \hat{\boldsymbol{\nu}}(1)} \odot (\boldsymbol{w}(2) - \boldsymbol{w}(1)) \right\|^2$$

$$< \mathcal{L}(\boldsymbol{w}(1)) - (\alpha^*)^2 \frac{1}{\eta} \left\| \sqrt[4]{\varepsilon \mathbb{1}_d + \hat{\boldsymbol{\nu}}(1)} \odot (\boldsymbol{w}(2) - \boldsymbol{w}(1)) \right\|^2 + \frac{L}{2\sqrt{\varepsilon}}(\alpha^*)^2 \left\| \sqrt[4]{\varepsilon \mathbb{1}_d + \hat{\boldsymbol{\nu}}(1)} \odot (\boldsymbol{w}(2) - \boldsymbol{w}(1)) \right\|^2$$

$$\leq \mathcal{L}(\boldsymbol{w}(1)) - (\alpha^*)^2 \frac{1}{2\eta} \left\| \sqrt[4]{\varepsilon \mathbb{1}_d + \hat{\boldsymbol{\nu}}(1)} \odot (\boldsymbol{w}(2) - \boldsymbol{w}(1)) \right\|^2, \tag{49}$$

where the second-to-last inequality is due to $\|\boldsymbol{w}(2) - \boldsymbol{w}(1)\| > 0$ (by Lemma 9) and $\alpha^* > (\alpha^*)^2$, while the last inequality is due to

$$\eta \leq \frac{\sqrt{\varepsilon} \inf_{t \geq 2} \left( \frac{1-\beta_1^t}{1-\beta_1} - \frac{1-\beta_1^{t-1}}{c(1-\beta_1)} \frac{1-(c\beta_1)^t}{1-(c\beta_1)^{t-1}} \right)}{L} \leq \frac{\sqrt{\varepsilon} \left( \frac{1-\beta_1^2}{1-\beta_1} - \frac{1-\beta_1}{c(1-\beta_1)} \frac{1-(c\beta_1)^2}{1-(c\beta_1)} \right)}{L}$$

$$= \frac{\sqrt{\varepsilon} \left( 1 + \beta_1 - \frac{1+(c\beta_1)}{c} \right)}{L} = \frac{\sqrt{\varepsilon} \left( 1 - \frac{1}{c} \right)}{L} < \frac{\sqrt{\varepsilon}}{L}.$$

Eq. (49) contradicts the fact that the equality in Eq. (48) holds for $1 + \alpha^*$, which completes the proof of $t = 1$.

If $t \geq 2$, following the similar routine as $t = 1$, we also prove Eq. (48) by reduction to absurdity. If there exist $t$ and $\alpha$ such that Eq. (48) fails. Denote $t^*$ as the smallest time such that there exists an $\alpha \in [0,1)$ such that Eq. (48) fails for $t^*$ and $\alpha$. By Lemma 9, $\left\| \sqrt[4]{\varepsilon \mathbb{1}_d + \hat{\boldsymbol{\nu}}(t^* - 1)} \odot (\boldsymbol{w}(t^*) - \boldsymbol{w}(t^* - 1)) \right\|^2$ is positive, and strict inequality in Eq. (48) holds for $t$ and $\alpha = 0$, which by continuity leads to

$$1 > \alpha^* \triangleq \inf\{\alpha \in [0,1] : Eq.(48) \; fails \; for \; 1 + \alpha\} > 0.$$

Then, for any $\alpha \in [0, \alpha^*]$, we have

$$\mathcal{L}(\boldsymbol{w}(t^* + \alpha))$$

$$\leq \mathcal{L}(\boldsymbol{w}(t^* + \alpha)) + \frac{1}{2}\alpha^2 \frac{1 - \beta_1^{t^*}}{\eta(1 - \beta_1)} \left\| \sqrt[4]{\varepsilon \mathbb{1}_d + \hat{\boldsymbol{\nu}}(t^*)} \odot (\boldsymbol{w}(t^* + 1) - \boldsymbol{w}(t^*)) \right\|^2$$

$$\leq \mathcal{L}(\boldsymbol{w}(t^*)) + \frac{1 - \beta_1^{t^*-1}}{2c\eta(1 - \beta_1)} \frac{1 - (c\beta_1)^{t^*}}{1 - (c\beta_1)^{t^*-1}} \left\| \sqrt[4]{\varepsilon \mathbb{1}_d + \hat{\boldsymbol{\nu}}(t^* - 1)} \odot (\boldsymbol{w}(t^*) - \boldsymbol{w}(t^* - 1)) \right\|^2.$$

On the other hand, for any time $2 \leq s \leq t^* - 1$, we have

$$\mathcal{L}(\boldsymbol{w}(s+1)) + \frac{1}{2} \frac{1 - \beta_1^s}{\eta(1 - \beta_1)} \left\| \sqrt[4]{\varepsilon \mathbb{1}_d + \hat{\boldsymbol{\nu}}(s)} \odot (\boldsymbol{w}(s+1) - \boldsymbol{w}(s)) \right\|^2$$

$$\leq \mathcal{L}(\boldsymbol{w}(s)) + \frac{\beta_1(1 - \beta_1^{s-1})}{2c\eta(1 - \beta_1)} \frac{1 - (c\beta_1)^s}{1 - (c\beta_1)^{s-1}} \left\| \sqrt[4]{\varepsilon \mathbb{1}_d + \hat{\boldsymbol{\nu}}(s-1)} \odot (\boldsymbol{w}(s) - \boldsymbol{w}(s-1)) \right\|^2. \tag{50}$$

By Eq. (47), we have

$$\frac{1 - \beta_1^s}{\eta(1 - \beta_1)} > \frac{1 - \beta_1^s}{c\eta(1 - \beta_1)} = \frac{1 - \beta_1^s}{c\eta(1 - \beta_1)} \frac{1 - (c\beta_1)^{s+1}}{1 - (c\beta_1)^s} \frac{1 - (c\beta_1)^s}{1 - (c\beta_1)^{s+1}},$$

which by $\frac{(1-\beta_1^{s-1})}{(1-\beta_1^s)}$ further leads to

$$
\begin{aligned}
&\mathcal{L}(\boldsymbol{w}(s)) + \frac{1-\beta_1^{s-1}}{2c\eta(1-\beta_1)} \frac{1-(c\beta_1)^s}{1-(c\beta_1)^{s-1}} \left\| \sqrt[4]{\varepsilon\mathbb{1}_d + \hat{\boldsymbol{\nu}}(s-1)} \odot (\boldsymbol{w}(s) - \boldsymbol{w}(s-1)) \right\|^2 \\
\geq& \mathcal{L}(\boldsymbol{w}(s+1)) + \frac{1}{2} \frac{1-\beta_1^s}{\eta(1-\beta_1)} \left\| \sqrt[4]{\varepsilon\mathbb{1}_d + \hat{\boldsymbol{\nu}}(s)} \odot (\boldsymbol{w}(s+1) - \boldsymbol{w}(s)) \right\|^2 \\
>& \mathcal{L}(\boldsymbol{w}(s+1)) + \frac{1-\beta_1^s}{2c\eta(1-\beta_1)} \frac{1-(c\beta_1)^{s+1}}{1-(c\beta_1)^s} \frac{1-(c\beta_1)^s}{1-(c\beta_1)^{s+1}} \left\| \sqrt[4]{\varepsilon\mathbb{1}_d + \hat{\boldsymbol{\nu}}(s)} \odot (\boldsymbol{w}(s+1) - \boldsymbol{w}(s)) \right\|^2 \\
>& \frac{1-(c\beta_1)^s}{1-(c\beta_1)^{s+1}} \left( \mathcal{L}(\boldsymbol{w}(s+1)) + \frac{1-\beta_1^s}{2c\eta(1-\beta_1)} \frac{1-(c\beta_1)^{s+1}}{1-(c\beta_1)^s} \left\| \sqrt[4]{\varepsilon\mathbb{1}_d + \hat{\boldsymbol{\nu}}(s)} \odot (\boldsymbol{w}(s+1) - \boldsymbol{w}(s)) \right\|^2 \right).
\end{aligned}
\tag{51}
$$

On the other hand, for $s = 1$, we have

$$
\begin{aligned}
\mathcal{L}(\boldsymbol{w}(1)) \geq& \mathcal{L}(\boldsymbol{w}(2)) + \frac{1}{2} \frac{1-\beta_1}{\eta(1-\beta_1)} \left\| \sqrt[4]{\varepsilon\mathbb{1}_d + \hat{\boldsymbol{\nu}}(1)} \odot (\boldsymbol{w}(2) - \boldsymbol{w}(1)) \right\|^2 \\
\geq& \frac{1-(c\beta_1)}{1-(c\beta_1)^2} \left( \mathcal{L}(\boldsymbol{w}(2)) + \frac{1-\beta_1}{2c\eta(1-\beta_1)} \frac{1-(c\beta_1)^2}{1-(c\beta_1)} \left\| \sqrt[4]{\varepsilon\mathbb{1}_d + \hat{\boldsymbol{\nu}}(1)} \odot (\boldsymbol{w}(2) - \boldsymbol{w}(1)) \right\|^2 \right).
\end{aligned}
\tag{52}
$$

Combining Eqs. (50), (51), and (52), we have

$$
\begin{aligned}
&\mathcal{L}(\boldsymbol{w}(t^* + \alpha)) \\
\leq& \mathcal{L}(\boldsymbol{w}(t^*)) + \frac{1-\beta_1^{t^*-1}}{2c\eta(1-\beta_1)} \frac{1-(c\beta_1)^{t^*}}{1-(c\beta_1)^{t^*-1}} \left\| \sqrt[4]{\varepsilon\mathbb{1}_d + \hat{\boldsymbol{\nu}}(t^*-1)} \odot (\boldsymbol{w}(t^*) - \boldsymbol{w}(t^*-1)) \right\|^2 \\
<& \frac{1-(c\beta_1)^{t^*}}{1-(c\beta_2)^{t^*-1}} \left( \mathcal{L}(\boldsymbol{w}(t^*-1)) + \frac{1-\beta_1^{t^*-2}}{2c\eta(1-\beta_1)} \frac{1-(c\beta_1)^{t^*-1}}{1-(c\beta_1)^{t^*-2}} \left\| \sqrt[4]{\varepsilon\mathbb{1}_d + \hat{\boldsymbol{\nu}}(t^*-2)} \odot (\boldsymbol{w}(t^*-1) - \boldsymbol{w}(t^*-2)) \right\|^2 \right) \\
<& \cdots \\
<& \frac{1-(c\beta_1)^{t^*}}{1-(c\beta_1)^2} \left( \mathcal{L}(\boldsymbol{w}(2)) + \frac{1-\beta_1}{2c\eta(1-\beta_1)} \frac{1-(c\beta_1)^2}{1-(c\beta_1)} \left\| \sqrt[4]{\varepsilon\mathbb{1}_d + \hat{\boldsymbol{\nu}}(1)} \odot (\boldsymbol{w}(2) - \boldsymbol{w}(1)) \right\|^2 \right) \\
\leq& \frac{1-(c\beta_1)^{t^*}}{1-c\beta_1} \mathcal{L}(\boldsymbol{w}(1)) < \frac{1}{1-c\beta_1} \mathcal{L}(\boldsymbol{w}(1)).
\end{aligned}
$$

Therefore, by Lemma 6, $\mathcal{L}$ is $H_{\ell^{-1}(\frac{1}{1-c\beta_1}N\mathcal{L}(\boldsymbol{w}(1)))}$ smooth (thus $L$ smooth) over the set $\{\boldsymbol{w}(t^* + \alpha) : \alpha \in [0, \alpha^*]\}$, which further leads to

$$\mathcal{L}(\boldsymbol{w}(t^* + \alpha^*))$$

$$\leq \mathcal{L}(\boldsymbol{w}(t^*)) + \alpha^* \langle \nabla \mathcal{L}(\boldsymbol{w}(t^*)), \boldsymbol{w}(t^* + 1) - \boldsymbol{w}(t^*) \rangle + \frac{L}{2}(\alpha^*)^2 \|\boldsymbol{w}(t^* + 1) - \boldsymbol{w}(t^*)\|^2$$

$$\overset{(\bullet)}{=} -\frac{\alpha^*}{\eta(1 - \beta_1)} \Big\langle \boldsymbol{w}(t^* + 1) - \boldsymbol{w}(t^*), (1 - \beta_1^{t^*})\sqrt{\varepsilon\mathbb{1}_d + \hat{\boldsymbol{\nu}}(t^*)} \odot (\boldsymbol{w}(t^* + 1) - \boldsymbol{w}(t^*))$$

$$- \beta_1(1 - \beta_1^{t^*-1})\sqrt{\varepsilon\mathbb{1}_d + \hat{\boldsymbol{\nu}}(t^*)} \odot (\boldsymbol{w}(t^*) - \boldsymbol{w}(t^* - 1)) \Big\rangle$$

$$+ \mathcal{L}(\boldsymbol{w}(t^*)) + \frac{L}{2}(\alpha^*)^2 \|\boldsymbol{w}(t^* + 1) - \boldsymbol{w}(t^*)\|^2$$

$$= \mathcal{L}(\boldsymbol{w}(t^*)) + \frac{L}{2}(\alpha^*)^2 \|\boldsymbol{w}(t^* + 1) - \boldsymbol{w}(t^*)\|^2 - \frac{\alpha^*(1 - \beta_1^{t^*})}{\eta(1 - \beta_1)} \left\| \sqrt[4]{\varepsilon\mathbb{1}_d + \hat{\boldsymbol{\nu}}(t^*)} \odot (\boldsymbol{w}(t^* + 1) - \boldsymbol{w}(t^*)) \right\|^2$$

$$+ \beta_1 \frac{\alpha^*(1 - \beta_1^{t^*-1})}{\eta(1 - \beta_1)} \Big\langle \boldsymbol{w}(t^* + 1) - \boldsymbol{w}(t^*), \sqrt{\varepsilon\mathbb{1}_d + \hat{\boldsymbol{\nu}}(t^*)} \odot (\boldsymbol{w}(t^*) - \boldsymbol{w}(t^* - 1)) \Big\rangle$$

$$= \mathcal{L}(\boldsymbol{w}(t^*)) + \frac{L}{2}(\alpha^*)^2 \|\boldsymbol{w}(t^* + 1) - \boldsymbol{w}(t^*)\|^2 - \frac{\alpha^*(1 - \beta_1^{t^*})}{\eta(1 - \beta_1)} \left\| \sqrt[4]{\varepsilon\mathbb{1}_d + \hat{\boldsymbol{\nu}}(t^*)} \odot (\boldsymbol{w}(t^* + 1) - \boldsymbol{w}(t^*)) \right\|^2$$

$$+ \beta_1 \frac{\alpha^*(1 - \beta_1^{t^*-1})}{\eta(1 - \beta_1)} \Big\langle \frac{\sqrt[8]{\varepsilon\mathbb{1}_d + \hat{\boldsymbol{\nu}}(t^* - 1)}}{\sqrt[8]{\varepsilon\mathbb{1}_d + \hat{\boldsymbol{\nu}}(t^*)}} \odot \sqrt[4]{\varepsilon\mathbb{1}_d + \hat{\boldsymbol{\nu}}(t^*)} \odot (\boldsymbol{w}(t^* + 1) - \boldsymbol{w}(t^*)),$$

$$\frac{\sqrt[8]{\varepsilon\mathbb{1}_d + \hat{\boldsymbol{\nu}}(t^* - 1)}}{\sqrt[8]{\varepsilon\mathbb{1}_d + \hat{\boldsymbol{\nu}}(t^*)}} \odot \sqrt[4]{\varepsilon\mathbb{1}_d + \hat{\boldsymbol{\nu}}(t^* - 1)} \odot (\boldsymbol{w}(t^*) - \boldsymbol{w}(t^* - 1)) \Big\rangle$$

$$\leq \mathcal{L}(\boldsymbol{w}(t^*)) + \frac{L}{2}(\alpha^*)^2 \|\boldsymbol{w}(t^* + 1) - \boldsymbol{w}(t^*)\|^2 - \frac{\alpha^*(1 - \beta_1^{t^*})}{\eta(1 - \beta_1)} \left\| \sqrt[4]{\varepsilon\mathbb{1}_d + \hat{\boldsymbol{\nu}}(t^*)} \odot (\boldsymbol{w}(t^* + 1) - \boldsymbol{w}(t^*)) \right\|^2$$

$$+ \beta_1 \frac{(\alpha^*)^2(1 - \beta_1^{t^*-1})}{2\eta(1 - \beta_1)} \left\| \frac{\sqrt[8]{\varepsilon\mathbb{1}_d + \hat{\boldsymbol{\nu}}(t^* - 1)}}{\sqrt[8]{\varepsilon\mathbb{1}_d + \hat{\boldsymbol{\nu}}(t^*)}} \odot \sqrt[4]{\varepsilon\mathbb{1}_d + \hat{\boldsymbol{\nu}}(t^*)} \odot (\boldsymbol{w}(t^* + 1) - \boldsymbol{w}(t^*)) \right\|^2$$

$$+ \beta_1 \frac{(1 - \beta_1^{t^*-1})}{2\eta(1 - \beta_1)} \left\| \frac{\sqrt[8]{\varepsilon\mathbb{1}_d + \hat{\boldsymbol{\nu}}(t^* - 1)}}{\sqrt[8]{\varepsilon\mathbb{1}_d + \hat{\boldsymbol{\nu}}(t^*)}} \odot \sqrt[4]{\varepsilon\mathbb{1}_d + \hat{\boldsymbol{\nu}}(t^* - 1)} \odot (\boldsymbol{w}(t^*) - \boldsymbol{w}(t^* - 1)) \right\|^2$$

$$\overset{(\diamond)}{\leq} \mathcal{L}(\boldsymbol{w}(t^*)) + \frac{L}{2}(\alpha^*)^2 \|\boldsymbol{w}(t^* + 1) - \boldsymbol{w}(t^*)\|^2 - \frac{\alpha^*(1 - \beta_1^{t^*})}{\eta(1 - \beta_1)} \left\| \sqrt[4]{\varepsilon\mathbb{1}_d + \hat{\boldsymbol{\nu}}(t^*)} \odot (\boldsymbol{w}(t^* + 1) - \boldsymbol{w}(t^*)) \right\|^2$$

$$+ \beta_1 \frac{(\alpha^*)^2(1 - \beta_1^{t^*-1})}{2\eta(1 - \beta_1)} \frac{1 - (c\beta_1)^{t^*}}{c\beta_1(1 - (c\beta_1)^{t^*-1})} \left\| \sqrt[4]{\varepsilon\mathbb{1}_d + \hat{\boldsymbol{\nu}}(t^*)} \odot (\boldsymbol{w}(t^* + 1) - \boldsymbol{w}(t^*)) \right\|^2$$

$$+ \beta_1 \frac{(1 - \beta_1^{t^*-1})}{2\eta(1 - \beta_1)} \frac{1 - (c\beta_1)^{t^*}}{c\beta_1(1 - (c\beta_1)^{t^*-1})} \left\| \sqrt[4]{\varepsilon\mathbb{1}_d + \hat{\boldsymbol{\nu}}(t^* - 1)} \odot (\boldsymbol{w}(t^*) - \boldsymbol{w}(t^* - 1)) \right\|^2$$

$$\leq \mathcal{L}(\boldsymbol{w}(t^*)) + \frac{L}{2\sqrt{\varepsilon}}(\alpha^*)^2 \| \sqrt[4]{\varepsilon\mathbb{1}_d + \hat{\boldsymbol{\nu}}(t^*)} \odot (\boldsymbol{w}(t^* + 1) - \boldsymbol{w}(t^*)) \|^2$$

$$- \frac{\alpha^*(1 - \beta_1^{t^*})}{\eta(1 - \beta_1)} \left\| \sqrt[4]{\varepsilon\mathbb{1}_d + \hat{\boldsymbol{\nu}}(t^*)} \odot (\boldsymbol{w}(t^* + 1) - \boldsymbol{w}(t^*)) \right\|^2$$

$$+ \beta_1 \frac{(\alpha^*)^2(1 - \beta_1^{t^*})}{2\eta(1 - \beta_1)} \frac{1 - (c\beta_1)^{t^*}}{c\beta_1(1 - (c\beta_1)^{t^*-1})} \left\| \sqrt[4]{\varepsilon\mathbb{1}_d + \hat{\boldsymbol{\nu}}(t^*)} \odot (\boldsymbol{w}(t^* + 1) - \boldsymbol{w}(t^*)) \right\|^2$$

$$+ \beta_1 \frac{(1 - \beta_1^{t^*})}{2\eta(1 - \beta_1)} \frac{1 - (c\beta_1)^{t^*}}{c\beta_1(1 - (c\beta_1)^{t^*-1})} \left\| \sqrt[4]{\varepsilon\mathbb{1}_d + \hat{\boldsymbol{\nu}}(t^* - 1)} \odot (\boldsymbol{w}(t^*) - \boldsymbol{w}(t^* - 1)) \right\|^2$$

$$\overset{(\square)}{<} \mathcal{L}(\boldsymbol{w}(t^*)) - \frac{(\alpha^*)^2(1 - \beta_1^{t^*})}{2\eta(1 - \beta_1)} \left\| \sqrt[4]{\varepsilon\mathbb{1}_d + \hat{\boldsymbol{\nu}}(t^*)} \odot (\boldsymbol{w}(t^* + 1) - \boldsymbol{w}(t^*)) \right\|^2$$

$$+\frac{(1-\beta_1^{t^*-1})}{2\eta(1-\beta_1)}\frac{1-(c\beta_1)^{t^*}}{c(1-(c\beta_1)^{t^*-1})}\left\|\sqrt[4]{\varepsilon\mathbb{1}_d+\hat{\boldsymbol{\nu}}(t^*-1)}\odot(\boldsymbol{w}(t^*)-\boldsymbol{w}(t^*-1))\right\|^2,$$

where Eq. ($\bullet$) is due to an alternative form of the Adam's update rule:

$$(1-\beta_1^{t^*})\sqrt{\varepsilon\mathbb{1}_d+\hat{\boldsymbol{\nu}}(t^*)}\odot(\boldsymbol{w}(t^*+1)-\boldsymbol{w}(t^*))-\beta_1(1-\beta_1^{t^*-1})\sqrt{\varepsilon\mathbb{1}_d+\hat{\boldsymbol{\nu}}(t^*-1)}\odot(\boldsymbol{w}(t^*)-\boldsymbol{w}(t^*-1))$$
$$=-\eta(1-\beta_1)\nabla\mathcal{L}(\boldsymbol{w}(t^*)), \tag{53}$$

Inequality ($\diamond$) is due to

$$\frac{\sqrt[4]{\varepsilon\mathbb{1}_d+\hat{\boldsymbol{\nu}}(t^*-1)}}{\sqrt[4]{\varepsilon\mathbb{1}_d+\hat{\boldsymbol{\nu}}(t^*)}}=\sqrt[4]{\frac{\varepsilon\mathbb{1}_d+\hat{\boldsymbol{\nu}}(t^*-1)}{\varepsilon\mathbb{1}_d+\hat{\boldsymbol{\nu}}(t^*)}}=\sqrt[4]{\frac{\varepsilon\mathbb{1}_d+\hat{\boldsymbol{\nu}}(t^*-1)}{\varepsilon\mathbb{1}_d+\frac{\beta_2\boldsymbol{\nu}(t^*-1)+(1-\beta_2)\nabla\mathcal{L}(\boldsymbol{w}(t^*))^2}{1-\beta_2^{t^*}}}}$$

$$\leq\sqrt[4]{\frac{\varepsilon\mathbb{1}_d+\hat{\boldsymbol{\nu}}(t^*-1)}{\varepsilon\mathbb{1}_d+\frac{\beta_2\boldsymbol{\nu}(t^*-1)}{1-\beta_2^{t^*}}}}=\sqrt[4]{\frac{\varepsilon\mathbb{1}_d+\hat{\boldsymbol{\nu}}(t^*-1)}{\varepsilon\mathbb{1}_d+\frac{\beta_2(1-\beta_2^{t^*-1})\hat{\boldsymbol{\nu}}(t^*-1)}{1-\beta_2^{t^*}}}}\leq\sqrt[4]{\frac{\varepsilon\mathbb{1}_d+\hat{\boldsymbol{\nu}}(t^*-1)}{\frac{\beta_2(1-\beta_2^{t^*-1})\hat{\boldsymbol{\nu}}(t^*-1)}{1-\beta_2^{t^*}}\varepsilon\mathbb{1}_d+\frac{\beta_2(1-\beta_2^{t^*-1})\hat{\boldsymbol{\nu}}(t^*-1)}{1-\beta_2^{t^*}}}}$$

$$=\sqrt[4]{\frac{1-\beta_2^{t^*}}{\beta_2(1-\beta_2^{t^*-1})}}\mathbb{1}_d\ (\textit{all the computings are component-wisely}),$$

and $f(c\beta_1)\geq\sqrt[4]{f((c\beta_1)^4)}$, and Inequality ($\square$) is due to

$$\frac{L}{2\sqrt{\varepsilon}}\leq\frac{\inf_{t\geq2}\left(\frac{1-\beta_1^t}{1-\beta_1}-\frac{1-\beta_1^{t-1}}{c(1-\beta_1)}\frac{1-(c\beta_1)^t}{1-(c\beta_1)^{t-1}}\right)}{2\eta}\leq\frac{\left(\frac{1-\beta_1^{t^*}}{1-\beta_1}-\frac{1-\beta_1^{t^*-1}}{c(1-\beta_1)}\frac{1-(c\beta_1)^{t^*}}{1-(c\beta_1)^{t^*-1}}\right)}{2\eta},$$

$\alpha^*>(\alpha^*)^2$, and $\left\|\sqrt[4]{\varepsilon\mathbb{1}_d+\hat{\boldsymbol{\nu}}(t^*)}\odot(\boldsymbol{w}(t^*+1)-\boldsymbol{w}(t^*))\right\|^2>0$.

This contradicts to that the equality in Eq. (48) holds for $t^*+\alpha^*$.

The proof is completed. $\qquad\square$

As $\lim_{t\to\infty}\beta_1^t=0$ and $\lim_{t\to\infty}(c\beta_1)^t=0$, we have the following corollary based on Lemma 1.

**Corollary 7.** *Let all assumptions in Theorem 5 hold. Then, for large enough t, we have*

$$\mathcal{L}(\boldsymbol{w}(t+1))+\frac{1}{2\sqrt[4]{c}\eta(1-\beta_1)}\left\|\sqrt[4]{\varepsilon\mathbb{1}_d+\hat{\boldsymbol{\nu}}(t)}\odot(\boldsymbol{w}(t+1)-\boldsymbol{w}(t))\right\|^2$$
$$\leq\mathcal{L}(\boldsymbol{w}(t))+\frac{1}{2\sqrt[2]{c}\eta(1-\beta_1)}\left\|\sqrt[4]{\varepsilon\mathbb{1}_d+\hat{\boldsymbol{\nu}}(t-1)}\odot(\boldsymbol{w}(t)-\boldsymbol{w}(t-1))\right\|^2. \tag{54}$$

*Consequently, we have*

$$\sum_{t=1}^{\infty}\|\nabla\mathcal{L}(\boldsymbol{w}(t))\|^2<\infty. \tag{55}$$

The proof of Corollary 7 relies on the following classical lemma on the equivalence between the convergence of two non-negative sequence. The proof is omitted here and can be found in [43].

**Lemma 18** (c.f. Lemma 27, [43]). *Let $\{a_i\}_{i=1}^{\infty}$ be a series of non-negative reals, and $\varepsilon$ be a positive real. Then, $\sum_{i=1}^{\infty}a_i<\infty$ is equivalent to $\sum_{i=1}^{\infty}\frac{a_i}{\sqrt{\varepsilon+\sum_{s=1}^{i}a_s}}<\infty$.*

*Proof of Corollary 7.* We have

$$\lim_{t\to\infty}\frac{1-\beta_1^{t-1}}{2c\eta(1-\beta_1)}\frac{1-(c\beta_1)^t}{1-(c\beta_1)^{t-1}}=\frac{1}{2c\eta(1-\beta_1)}<\frac{1}{2\sqrt[2]{c}\eta(1-\beta_1)},$$

$$\lim_{t\to\infty}\frac{1-\beta_1^t}{2\eta(1-\beta_1)}=\frac{1}{2\eta(1-\beta_1)}>\frac{1}{2\sqrt[4]{c}\eta(1-\beta_1)},$$

which completes the proof of Eq. (54). Rearranging Eq. (54) leads to

$$\frac{\sqrt[4]{c}-1}{2\sqrt[2]{c}\eta(1-\beta_1)}\left\|\sqrt[4]{\varepsilon\mathbb{1}_d+\hat{\boldsymbol{\nu}}(t)}\odot(\boldsymbol{w}(t+1)-\boldsymbol{w}(t))\right\|^2 \le \frac{1}{2\sqrt[4]{c}\eta(1-\beta_1)}\left\|\sqrt[4]{\varepsilon\mathbb{1}_d+\hat{\boldsymbol{\nu}}(t-1)}\odot(\boldsymbol{w}(t)-\boldsymbol{w}(t-1))\right\|^2$$

$$+\mathcal{L}(\boldsymbol{w}(t))-\left(\mathcal{L}(\boldsymbol{w}(t+1))+\frac{1}{2\sqrt[4]{c}\eta(1-\beta_1)}\left\|\sqrt[4]{\varepsilon\mathbb{1}_d+\hat{\boldsymbol{\nu}}(t)}\odot(\boldsymbol{w}(t+1)-\boldsymbol{w}(t))\right\|^2\right),$$

which by iteration further leads to that for a large enough time $T_1$

$$\sum_{t=T_1}^{T_2}\frac{\sqrt[4]{c}-1}{2\sqrt[2]{c}\eta(1-\beta_1)}\left\|\sqrt[4]{\varepsilon\mathbb{1}_d+\hat{\boldsymbol{\nu}}(t)}\odot(\boldsymbol{w}(t+1)-\boldsymbol{w}(t))\right\|^2$$

$$\le\mathcal{L}(\boldsymbol{w}(T_1))+\frac{1}{2\sqrt[4]{c}\eta(1-\beta_1)}\left\|\sqrt[4]{\varepsilon\mathbb{1}_d+\hat{\boldsymbol{\nu}}(T_1-1)}\odot(\boldsymbol{w}(T_1)-\boldsymbol{w}(T_1-1))\right\|^2$$

$$-\mathcal{L}(\boldsymbol{w}(T_2+1))+\frac{1}{2\sqrt[4]{c}\eta(1-\beta_1)}\left\|\sqrt[4]{\varepsilon\mathbb{1}_d+\hat{\boldsymbol{\nu}}(T_2)}\odot(\boldsymbol{w}(T_2+1)-\boldsymbol{w}(T_2+1))\right\|^2$$

$$<\mathcal{L}(\boldsymbol{w}(T_1))+\frac{1}{2\sqrt[4]{c}\eta(1-\beta_1)}\left\|\sqrt[4]{\varepsilon\mathbb{1}_d+\hat{\boldsymbol{\nu}}(T_1-1)}\odot(\boldsymbol{w}(T_1)-\boldsymbol{w}(T_1-1))\right\|^2.$$

Consequently, we obtain

$$\sum_{t=1}^{\infty}\frac{\sqrt[4]{c}-1}{2\sqrt[2]{c}\eta(1-\beta_1)}\left\|\sqrt[4]{\varepsilon\mathbb{1}_d+\hat{\boldsymbol{\nu}}(t)}\odot(\boldsymbol{w}(t+1)-\boldsymbol{w}(t))\right\|^2 < \infty. \tag{56}$$

On the other hand, for any $t$, we have

$$\left\|\sqrt[4]{\varepsilon\mathbb{1}_d+\hat{\boldsymbol{\nu}}(t)}\odot(\boldsymbol{w}(t+1)-\boldsymbol{w}(t))\right\|\left\|\sqrt[4]{\varepsilon\mathbb{1}_d+\hat{\boldsymbol{\nu}}(t)}\odot\hat{\boldsymbol{w}}\right\|$$

$$\ge\left\langle\sqrt[4]{\varepsilon\mathbb{1}_d+\hat{\boldsymbol{\nu}}(t)}\odot(\boldsymbol{w}(t+1)-\boldsymbol{w}(t)),\sqrt[4]{\varepsilon\mathbb{1}_d+\hat{\boldsymbol{\nu}}(t)}\odot\hat{\boldsymbol{w}}\right\rangle=\left\langle\sqrt[2]{\varepsilon\mathbb{1}_d+\hat{\boldsymbol{\nu}}(t)}\odot(\boldsymbol{w}(t+1)-\boldsymbol{w}(t)),\hat{\boldsymbol{w}}\right\rangle$$

$$=\langle-\eta\hat{\boldsymbol{m}}(t),\hat{\boldsymbol{w}}\rangle=-\frac{\eta(1-\beta_1)}{1-\beta_1^t}\left\langle\sum_{s=1}^{t}\beta_1^{t-s}\nabla\mathcal{L}(\boldsymbol{w}(s)),\hat{\boldsymbol{w}}\right\rangle$$

$$=-\frac{\eta(1-\beta_1)}{1-\beta_1^t}\frac{1}{N}\left\langle\sum_{s=1}^{t}\beta_1^{t-s}\sum_{\boldsymbol{x}_i\in S}\ell'(\langle\boldsymbol{x}_i,\boldsymbol{w}(s)\rangle)\boldsymbol{x}_i,\hat{\boldsymbol{w}}\right\rangle\ge-\frac{\eta(1-\beta_1)}{1-\beta_1^t}\frac{1}{N}\sum_{s=1}^{t}\beta_1^{t-s}\sum_{\boldsymbol{x}_i\in S}\ell'(\langle\boldsymbol{x}_i,\boldsymbol{w}(s)\rangle)$$

$$\ge-\frac{\eta(1-\beta_1)}{1-\beta_1^t}\frac{1}{N}\sum_{\boldsymbol{x}_i\in S}\ell'(\langle\boldsymbol{x}_i,\boldsymbol{w}(t)\rangle)\ge\frac{\eta(1-\beta_1)}{1-\beta_1^t}\|\nabla\mathcal{L}(\boldsymbol{w}(t))\|,$$

which by Eq. (56) indicates

$$\sum_{t=1}^{\infty}\left(\frac{\eta(1-\beta_1)}{1-\beta_1^t}\right)^2\frac{\|\nabla\mathcal{L}(\boldsymbol{w}(t))\|^2}{\left\|\sqrt[4]{\varepsilon\mathbb{1}_d+\hat{\boldsymbol{\nu}}(t)}\odot\hat{\boldsymbol{w}}\right\|^2}<\infty.$$

As $\lim_{t\to\infty}\left(\frac{\eta(1-\beta_1)}{1-\beta_1^t}\right)^2=\eta^2(1-\beta_1)^2$, we then obtain

$$\sum_{t=1}^{\infty}\frac{\|\nabla\mathcal{L}(\boldsymbol{w}(t))\|^2}{\sqrt[2]{\varepsilon+\sum_{s=1}^{t}\|\nabla\mathcal{L}(\boldsymbol{w}(t))\|^2}}\le\sum_{t=1}^{\infty}\frac{\|\nabla\mathcal{L}(\boldsymbol{w}(t))\|^2}{\sqrt[2]{\varepsilon+\sum_{s=1}^{t}(1-\beta)\beta^{t-s}\|\nabla\mathcal{L}(\boldsymbol{w}(t))\|^2}}$$

$$\le\sqrt{\frac{1}{1-\beta}}\sum_{t=1}^{\infty}\frac{\|\nabla\mathcal{L}(\boldsymbol{w}(t))\|^2}{\sqrt[2]{\varepsilon+\frac{\sum_{s=1}^{t}(1-\beta)\beta^{t-s}\|\nabla\mathcal{L}(\boldsymbol{w}(t))\|^2}{1-\beta^t}}}\le d\sqrt{\frac{1}{1-\beta}}\sum_{t=1}^{\infty}\frac{\|\nabla\mathcal{L}(\boldsymbol{w}(t))\|^2}{\left\|\sqrt[4]{\varepsilon\mathbb{1}_d+\frac{\sum_{s=1}^{t}(1-\beta)\beta^{t-s}\nabla\mathcal{L}(\boldsymbol{w}(t))^2}{1-\beta^t}}\right\|^2}$$

$$=d\sqrt{\frac{1}{1-\beta}}\sum_{t=1}^{\infty}\frac{\|\nabla\mathcal{L}(\boldsymbol{w}(t))\|^2}{\left\|\sqrt[4]{\varepsilon\mathbb{1}_d+\hat{\boldsymbol{\nu}}(t)}\right\|^2}\le d\|\hat{\boldsymbol{w}}\|_\infty^2\sqrt{\frac{1}{1-\beta}}\sum_{t=1}^{\infty}\frac{\|\nabla\mathcal{L}(\boldsymbol{w}(t))\|^2}{\left\|\sqrt[4]{\varepsilon\mathbb{1}_d+\hat{\boldsymbol{\nu}}(t)}\odot\hat{\boldsymbol{w}}\right\|^2}<\infty,$$

which by Lemma 18 completes the proof. $\qquad\square$

Based on Corollary 7, we can further prove Lemma 19, characterizing the convergent rate of loss $\mathcal{L}$ directly.

**Lemma 19.** *Let all conditions in Theorem 5 hold. Then, $\mathcal{L}(\boldsymbol{w}(t)) = \Theta\left(t^{-1}\right)$, $\|\boldsymbol{w}(t)\| = \Theta(\ln(t))$, and $\|\boldsymbol{w}(t) - \boldsymbol{w}(t-1)\| = \Theta(t^{-1})$.*

*Proof of Lemma 19.* To begin with, Eq. (53) indicates

$$\|\eta(1-\beta_1)\nabla\mathcal{L}(\boldsymbol{w}(t))\|^2$$

$$= \left\|(1-\beta_1^t)\sqrt{\varepsilon\mathbb{1}_d + \hat{\boldsymbol{\nu}}(t)} \odot (\boldsymbol{w}(t+1) - \boldsymbol{w}(t)) - \beta_1(1-\beta_1^{t-1})\sqrt{\varepsilon\mathbb{1}_d + \hat{\boldsymbol{\nu}}(t-1)} \odot (\boldsymbol{w}(t) - \boldsymbol{w}(t-1))\right\|^2$$

$$\leq \left\|(1-\beta_1^t)\sqrt{\varepsilon\mathbb{1}_d + \hat{\boldsymbol{\nu}}(t)} \odot (\boldsymbol{w}(t+1) - \boldsymbol{w}(t))\right\| + \left\|\beta_1(1-\beta_1^{t-1})\sqrt{\varepsilon\mathbb{1}_d + \hat{\boldsymbol{\nu}}(t-1)} \odot (\boldsymbol{w}(t) - \boldsymbol{w}(t-1))\right\|^2$$

$$\leq \left(\left\|\sqrt{\varepsilon\mathbb{1}_d + \hat{\boldsymbol{\nu}}(t)} \odot (\boldsymbol{w}(t+1) - \boldsymbol{w}(t))\right\| + \left\|\sqrt{\varepsilon\mathbb{1}_d + \hat{\boldsymbol{\nu}}(t-1)} \odot (\boldsymbol{w}(t) - \boldsymbol{w}(t-1))\right\|\right)^2$$

$$\leq 2\left(\left\|\sqrt{\varepsilon\mathbb{1}_d + \hat{\boldsymbol{\nu}}(t)} \odot (\boldsymbol{w}(t+1) - \boldsymbol{w}(t))\right\|^2 + \left\|\sqrt{\varepsilon\mathbb{1}_d + \hat{\boldsymbol{\nu}}(t-1)} \odot (\boldsymbol{w}(t) - \boldsymbol{w}(t-1))\right\|^2\right) \qquad (57)$$

On the other hand, by Corollary 7,

$$\sum_{s=1}^{\infty}\|\nabla\mathcal{L}(\boldsymbol{w}(s))\|^2 < \infty,$$

which following the same routine as Corollary 6 leads to

$$\langle \boldsymbol{w}(t), \boldsymbol{x} \rangle \to \infty, \forall \boldsymbol{x} \in \tilde{\boldsymbol{S}}.$$

Therefore, by Lemma 7, there exists a large enough time $T_1$, such that $\forall t \geq T_1$,

$$\frac{1}{K}\ell(\langle \boldsymbol{w}(t), \boldsymbol{x} \rangle) \leq -\ell'(\langle \boldsymbol{w}(t), \boldsymbol{x} \rangle) \leq K\ell(\langle \boldsymbol{w}(t), \boldsymbol{x} \rangle), \forall \boldsymbol{x} \in \boldsymbol{S},$$

which by the separable assumption further leads to

$$\frac{\gamma}{K}\mathcal{L}(\boldsymbol{w}(t)) \leq -\frac{\gamma}{N}\sum_{\boldsymbol{x}\in\boldsymbol{S}}\ell'(\langle \boldsymbol{w}(t), \boldsymbol{x} \rangle) \leq \frac{1}{N}\left\langle -\sum_{\boldsymbol{x}\in\boldsymbol{S}}\ell'(\langle \boldsymbol{w}(t), \boldsymbol{x} \rangle)\boldsymbol{x}, \gamma\hat{\boldsymbol{w}} \right\rangle$$

$$\leq \frac{1}{N}\left\|\sum_{\boldsymbol{x}\in\boldsymbol{S}}\ell'(\langle \boldsymbol{w}(t), \boldsymbol{x} \rangle)\boldsymbol{x}\right\|\|\gamma\hat{\boldsymbol{w}}\| = \|\nabla\mathcal{L}(\boldsymbol{w}(t))\|$$

$$\leq -\frac{1}{N}\sum_{\boldsymbol{x}\in\boldsymbol{S}}\ell'(\langle \boldsymbol{w}(t), \boldsymbol{x} \rangle) \leq K\mathcal{L}(\boldsymbol{w}(t)). \qquad (58)$$

Combining Eq. (26) and the above inequality, we have

$$\left(\frac{\eta(1-\beta_1)\gamma}{K}\right)^2 \mathcal{L}(\boldsymbol{w}(t))^2 \leq 2\left(\left\|\sqrt{\varepsilon\mathbb{1}_d + \hat{\boldsymbol{\nu}}(t)} \odot (\boldsymbol{w}(t+1) - \boldsymbol{w}(t))\right\|^2 \right.$$

$$\left. + \left\|\sqrt{\varepsilon\mathbb{1}_d + \hat{\boldsymbol{\nu}}(t-1)} \odot (\boldsymbol{w}(t) - \boldsymbol{w}(t-1))\right\|^2\right). \qquad (59)$$

On the other hand, by Eq. (56), we have

$$\sum_{t=1}^{\infty}\left\|\sqrt[4]{\varepsilon\mathbb{1}_d + \hat{\boldsymbol{\nu}}(t)} \odot (\boldsymbol{w}(t+1) - \boldsymbol{w}(t))\right\|^2 < \infty.$$

Therefore, there exists large enough time $T_2$, such that $\forall t > T_2$,

$$\left\|\sqrt[4]{\varepsilon\mathbb{1}_d + \hat{\boldsymbol{\nu}}(t)} \odot (\boldsymbol{w}(t+1) - \boldsymbol{w}(t))\right\|^2 < 1,$$

and thus,

$$\left\|\sqrt[4]{\varepsilon\mathbb{1}_d + \hat{\boldsymbol{\nu}}(t)} \odot (\boldsymbol{w}(t+1) - \boldsymbol{w}(t))\right\|^4 < \left\|\sqrt[4]{\varepsilon\mathbb{1}_d + \hat{\boldsymbol{\nu}}(t)} \odot (\boldsymbol{w}(t+1) - \boldsymbol{w}(t))\right\|^2. \qquad (60)$$

Combining Eq. (59) and Eq. (60), there exists a positive real constant $C$, such that

$$\mathcal{L}(\boldsymbol{w}(t))^2 \leq C \left( \left\| \sqrt{\varepsilon \mathbb{1}_d + \hat{\boldsymbol{\nu}}(t)} \odot (\boldsymbol{w}(t+1) - \boldsymbol{w}(t)) \right\|^2 \right.$$
$$\left. + \left\| \sqrt{\varepsilon \mathbb{1}_d + \hat{\boldsymbol{\nu}}(t-1)} \odot (\boldsymbol{w}(t) - \boldsymbol{w}(t-1)) \right\|^2 \right),$$

$$\left\| \sqrt[4]{\varepsilon \mathbb{1}_d + \hat{\boldsymbol{\nu}}(t)} \odot (\boldsymbol{w}(t+1) - \boldsymbol{w}(t)) \right\|^4 \leq C \left( \left\| \sqrt{\varepsilon \mathbb{1}_d + \hat{\boldsymbol{\nu}}(t)} \odot (\boldsymbol{w}(t+1) - \boldsymbol{w}(t)) \right\|^2 \right.$$
$$\left. + \left\| \sqrt{\varepsilon \mathbb{1}_d + \hat{\boldsymbol{\nu}}(t-1)} \odot (\boldsymbol{w}(t) - \boldsymbol{w}(t-1)) \right\|^2 \right).$$

Rearranging Eq. (54) leads to

$$\frac{\sqrt[4]{c} - 1}{4\sqrt[2]{c}\eta(1 - \beta_1)} \left( \left\| \sqrt[4]{\varepsilon \mathbb{1}_d + \hat{\boldsymbol{\nu}}(t-1)} \odot (\boldsymbol{w}(t) - \boldsymbol{w}(t-1)) \right\|^2 \right.$$
$$\left. + \left\| \sqrt[4]{\varepsilon \mathbb{1}_d + \hat{\boldsymbol{\nu}}(t-1)} \odot (\boldsymbol{w}(t) - \boldsymbol{w}(t-1)) \right\|^2 \right)$$
$$\leq \mathcal{L}(\boldsymbol{w}(t)) + \frac{\sqrt[4]{c} + 1}{4\sqrt[2]{c}\eta(1 - \beta_1)} \left\| \sqrt[4]{\varepsilon \mathbb{1}_d + \hat{\boldsymbol{\nu}}(t-1)} \odot (\boldsymbol{w}(t) - \boldsymbol{w}(t-1)) \right\|^2$$
$$- \left( \mathcal{L}(\boldsymbol{w}(t+1)) + \frac{\sqrt[4]{c} + 1}{4\sqrt[2]{c}\eta(1 - \beta_1)} \left\| \sqrt[4]{\varepsilon \mathbb{1}_d + \hat{\boldsymbol{\nu}}(t)} \odot (\boldsymbol{w}(t+1) - \boldsymbol{w}(t)) \right\|^2 \right),$$

which further indicates

$$\left( \mathcal{L}(\boldsymbol{w}(t)) + \frac{\sqrt[4]{c} + 1}{4\sqrt[2]{c}\eta(1 - \beta_1)} \left\| \sqrt[4]{\varepsilon \mathbb{1}_d + \hat{\boldsymbol{\nu}}(t-1)} \odot (\boldsymbol{w}(t) - \boldsymbol{w}(t-1)) \right\|^2 \right)^2$$
$$\leq 2 \left( \mathcal{L}(\boldsymbol{w}(t))^2 + \frac{\sqrt[4]{c} + 1}{4\sqrt[2]{c}\eta(1 - \beta_1)} \left\| \sqrt[4]{\varepsilon \mathbb{1}_d + \hat{\boldsymbol{\nu}}(t-1)} \odot (\boldsymbol{w}(t) - \boldsymbol{w}(t-1)) \right\|^4 \right)$$
$$\leq 2C \left( 1 + \frac{\sqrt[4]{c} + 1}{4\sqrt[2]{c}\eta(1 - \beta_1)} \right) \left( \left\| \sqrt{\varepsilon \mathbb{1}_d + \hat{\boldsymbol{\nu}}(t)} \odot (\boldsymbol{w}(t+1) - \boldsymbol{w}(t)) \right\|^2 \right.$$
$$\left. + \left\| \sqrt{\varepsilon \mathbb{1}_d + \hat{\boldsymbol{\nu}}(t-1)} \odot (\boldsymbol{w}(t) - \boldsymbol{w}(t-1)) \right\|^2 \right)$$
$$\leq 2C \left( 1 + \frac{\sqrt[4]{c} + 1}{4\sqrt[2]{c}\eta(1 - \beta_1)} \right) \frac{4\sqrt[2]{c}\eta(1 - \beta_1)}{\sqrt[4]{c} - 1} \left( \mathcal{L}(\boldsymbol{w}(t)) + \frac{\sqrt[4]{c} + 1}{4\sqrt[2]{c}\eta(1 - \beta_1)} \right.$$
$$\cdot \left\| \sqrt[4]{\varepsilon \mathbb{1}_d + \hat{\boldsymbol{\nu}}(t-1)} \odot (\boldsymbol{w}(t) - \boldsymbol{w}(t-1)) \right\|^2 - (\mathcal{L}(\boldsymbol{w}(t+1)))$$
$$\left. + \frac{\sqrt[4]{c} + 1}{4\sqrt[2]{c}\eta(1 - \beta_1)} \left\| \sqrt[4]{\varepsilon \mathbb{1}_d + \hat{\boldsymbol{\nu}}(t)} \odot (\boldsymbol{w}(t+1) - \boldsymbol{w}(t)) \right\|^2 \right) \right).$$

Denote $\xi(t)$ as

$$\xi(t) \triangleq \mathcal{L}(\boldsymbol{w}(t)) + \frac{\sqrt[4]{c} + 1}{4\sqrt[2]{c}\eta(1 - \beta_1)} \left\| \sqrt[4]{\varepsilon \mathbb{1}_d + \hat{\boldsymbol{\nu}}(t-1)} \odot (\boldsymbol{w}(t) - \boldsymbol{w}(t-1)) \right\|^2.$$

We then have

$$\xi(t)^2 \leq 2C \left( 1 + \frac{\sqrt[4]{c} + 1}{4\sqrt[2]{c}\eta(1 - \beta_1)} \right) \frac{4\sqrt[2]{c}\eta(1 - \beta_1)}{\sqrt[4]{c} - 1} (\xi(t) - \xi(t+1)),$$

which leads to

$$\xi(t) = \mathcal{O}\left( \frac{1}{t} \right), \ i.e., \ \mathcal{L}(\boldsymbol{w}(t)) = \mathcal{O}\left( \frac{1}{t} \right),$$
$$and \ \left\| \sqrt[4]{\varepsilon \mathbb{1}_d + \hat{\boldsymbol{\nu}}(t-1)} \odot (\boldsymbol{w}(t) - \boldsymbol{w}(t-1)) \right\|^2 = \mathcal{O}\left( \frac{1}{t} \right).$$

Due to Eq. (58), we further have $\|\nabla\mathcal{L}(\boldsymbol{w}(t))\| = \mathcal{O}(t^{-1})$, which indicates

$$
\begin{aligned}
\|\boldsymbol{w}(t)\| \leq & \|\boldsymbol{w}(1)\| + \sum_{s=1}^{t} \|\boldsymbol{w}(s+1) - \boldsymbol{w}(s)\| = \|\boldsymbol{w}(1)\| + \eta \sum_{s=1}^{t} \left\| \frac{\hat{\boldsymbol{m}}(s)}{\sqrt{\hat{\boldsymbol{\nu}}(s) + \varepsilon \mathbb{1}_d}} \right\| \\
\leq & \|\boldsymbol{w}(1)\| + \frac{\eta}{\sqrt{\varepsilon}} \sum_{s=1}^{t} \|\hat{\boldsymbol{m}}(s)\| = \|\boldsymbol{w}(1)\| + \frac{\eta}{\sqrt{\varepsilon}} \sum_{s=1}^{t} \frac{1}{(1-\beta^s)} \left\| \sum_{i=1}^{s} \beta^{s-i} \nabla\mathcal{L}(\boldsymbol{w}(i)) \right\| \\
\leq & \|\boldsymbol{w}(1)\| + \frac{\eta}{\sqrt{\varepsilon}(1-\beta)} \sum_{s=1}^{t} \sum_{i=1}^{s} \beta^{s-i} \|\nabla\mathcal{L}(\boldsymbol{w}(i))\| \\
\leq & \|\boldsymbol{w}(1)\| + \frac{\eta}{\sqrt{\varepsilon}(1-\beta)^2} \sum_{s=1}^{t} \|\nabla\mathcal{L}(\boldsymbol{w}(s))\| = \mathcal{O}(\ln(t)).
\end{aligned}
$$

Therefore, for any $\boldsymbol{x} \in \boldsymbol{S}$, we have $\langle \boldsymbol{w}(t), \boldsymbol{x} \rangle = \mathcal{O}(\ln(t))$, which by $\ell$ is exponential-tailed leads to $\ell(\langle \boldsymbol{w}(t), \boldsymbol{x} \rangle) = \Omega(t^{-1})$, and thus $\mathcal{L}(\boldsymbol{w}(t)) = \Theta(t^{-1})$. Also, since $\mathcal{L}(\boldsymbol{w}(t)) = \mathcal{O}(t^{-1})$, we have $\langle \boldsymbol{w}(t), \boldsymbol{x} \rangle = \Omega(\ln(t))$, which further leads to $\|\boldsymbol{w}(t)\| = \Omega(\ln(t))$, and thus $\|\boldsymbol{w}(t)\| = \Theta(\ln(t))$.

Finally, we have

$$
\begin{aligned}
\gamma \sum_{s=1}^{t} \beta^{t-s} \|\nabla\mathcal{L}(\boldsymbol{w}(s))\| = & \frac{\gamma}{N} \sum_{s=1}^{t} \beta^{t-s} \left\| \sum_{\boldsymbol{x} \in \boldsymbol{S}} \ell'(\langle \boldsymbol{w}(s), \boldsymbol{x} \rangle) \boldsymbol{x} \right\| \\
\leq & -\frac{\gamma}{N} \sum_{s=1}^{t} \beta^{t-s} \sum_{\boldsymbol{x} \in \boldsymbol{S}} \ell'(\langle \boldsymbol{w}(s), \boldsymbol{x} \rangle) \leq -\frac{1}{N} \sum_{s=1}^{t} \beta^{t-s} \left\langle \sum_{\boldsymbol{x} \in \boldsymbol{S}} \ell'(\langle \boldsymbol{w}(s), \boldsymbol{x} \rangle) \boldsymbol{x}, \gamma \hat{\boldsymbol{w}} \right\rangle \\
\leq & \left\| \sum_{s=1}^{t} \beta^{t-s} \nabla\mathcal{L}(\boldsymbol{w}(s)) \right\| \|\gamma \hat{\boldsymbol{w}}\| = \|\boldsymbol{m}(t)\| \leq \sum_{s=1}^{t} \beta^{t-s} \|\nabla\mathcal{L}(\boldsymbol{w}(s))\|,
\end{aligned}
$$

which leads to $\|\boldsymbol{m}(t)\| = \Theta(t^{-1})$. Similarly, we have $\boldsymbol{\nu}(t) = \mathcal{O}(t^{-2})$, component-wisely. As $\lim_{t \to \infty} \beta_1^t = 0$ and $\lim_{t \to \infty} \beta_2^t = 0$, we have

$$
\|\boldsymbol{w}(t) - \boldsymbol{w}(t-1)\| = \left\| \frac{\hat{\boldsymbol{m}}(t)}{\sqrt{\varepsilon \mathbb{1}_d + \hat{\boldsymbol{\nu}}(t)}} \right\| = \Theta(t^{-1}).
$$

The proof is completed. $\qquad \square$

## D.3 Parameter dynamics

By Lemma 4, there exists a solution $\tilde{\boldsymbol{w}}$ as the solution of Eq. (10) with $C_3 = \frac{\eta}{(1-\beta)\sqrt{\varepsilon}}$. Define $\boldsymbol{r}(t)$ as

$$
\boldsymbol{r}(t) \stackrel{\triangle}{=} \boldsymbol{w}(t) - \ln(t)\hat{\boldsymbol{w}} - \tilde{\boldsymbol{w}}, \tag{61}
$$

and we only need to prove $\|\boldsymbol{r}(t)\|$ is bounded over time. We then prove $\boldsymbol{r}(t)$ has bounded norm. Specifically, we will prove the following lemma:

**Lemma 20.** *Let all conditions in Theorem 5 hold. Then, $\|\boldsymbol{r}(t)\|$ is bounded if and only if $g(t)$ is upper bounded, where $g(t)$ is defined as follows.*

$$
\begin{aligned}
g(t) \stackrel{\triangle}{=} & \left\langle \boldsymbol{r}(t), (1 - \beta_1^{t-1})\sqrt{\varepsilon \mathbb{1}_d + \hat{\boldsymbol{\nu}}(t-1)} \odot (\boldsymbol{w}(t) - \boldsymbol{w}(t-1)) \right\rangle \\
& \cdot \frac{\beta_1}{1-\beta_1} + \frac{\sqrt{\varepsilon}}{2} \|\boldsymbol{r}(t)\|^2 - \frac{\beta_1}{1-\beta_1} \sum_{\tau=2}^{t} \langle \boldsymbol{r}(\tau) - \boldsymbol{r}(\tau-1), \\
& (1 - \beta_1^{\tau-1})\sqrt{\varepsilon \mathbb{1}_d + \hat{\boldsymbol{\nu}}(\tau-1)} \odot (\boldsymbol{w}(\tau) - \boldsymbol{w}(\tau-1)) \rangle.
\end{aligned}
$$

*Furthermore, we have $\sum_{s=1}^{\infty} (g(t+1) - g(t))$ is upper bounded.*

Similar to GDM, the proof of Lemma 20 is divided into two parts, each focus on one claim of it. We start with the first claim.

**Lemma 21.** *Let all conditions in Theorem 5 hold. Then, $\|\boldsymbol{r}(t)\|$ is bounded if and only if $g(t)$ is upper bounded.*

*Proof.* Following the same routine as Lemma 11 and Lemma 14, we only need to prove

$$\lim_{t\to\infty}\left\|(1-\beta_1^{t-1})\sqrt{\varepsilon\mathbb{1}_d+\hat{\boldsymbol{\nu}}(t-1)}\odot(\boldsymbol{w}(t)-\boldsymbol{w}(t-1))\right\|=0, \tag{62}$$

and

$$\sum_{\tau=2}^{\infty}\left|\langle\boldsymbol{r}(\tau)-\boldsymbol{r}(\tau-1),(1-\beta_1^{\tau-1})\sqrt{\varepsilon\mathbb{1}_d+\hat{\boldsymbol{\nu}}(\tau-1)}\odot(\boldsymbol{w}(\tau)-\boldsymbol{w}(\tau-1))\rangle\right|<\infty. \tag{63}$$

As for Eq. (62), by Lemma 19, we have

$$\left\|(1-\beta_1^{t-1})\sqrt{\varepsilon\mathbb{1}_d+\hat{\boldsymbol{\nu}}(t-1)}\odot(\boldsymbol{w}(t)-\boldsymbol{w}(t-1))\right\|$$
$$=\mathcal{O}(t^{-1})=\boldsymbol{o}(1).$$

As for Eq. (62), we have

$$\left|\left\langle\boldsymbol{r}(\tau)-\boldsymbol{r}(\tau-1),(1-\beta_1^{\tau-1})\sqrt{\varepsilon\mathbb{1}_d+\hat{\boldsymbol{\nu}}(\tau-1)}\odot(\boldsymbol{w}(\tau)-\boldsymbol{w}(\tau-1))\right\rangle\right|$$
$$=\left|\left\langle\boldsymbol{w}(\tau)-\boldsymbol{w}(\tau-1)-\ln\frac{\tau}{\tau-1}\hat{\boldsymbol{w}},(1-\beta_1^{\tau-1})\sqrt{\varepsilon\mathbb{1}_d+\hat{\boldsymbol{\nu}}(\tau-1)}\odot(\boldsymbol{w}(\tau)-\boldsymbol{w}(\tau-1))\right\rangle\right|$$
$$\leq\left|\left\langle\ln\frac{\tau}{\tau-1}\hat{\boldsymbol{w}},(1-\beta_1^{\tau-1})\sqrt{\varepsilon\mathbb{1}_d+\hat{\boldsymbol{\nu}}(\tau-1)}\odot(\boldsymbol{w}(\tau)-\boldsymbol{w}(\tau-1))\right\rangle\right|$$
$$+(1-\beta_1^{\tau-1})\left\|\sqrt[4]{\varepsilon\mathbb{1}_d+\hat{\boldsymbol{\nu}}(\tau-1)}\odot(\boldsymbol{w}(\tau)-\boldsymbol{w}(\tau-1))\right\|^2$$
$$\overset{(\star)}{=}\mathcal{O}(\tau^{-2}),$$

where Eq. $(\star)$ is due to Lemma 19 and $\ln(\frac{\tau}{\tau-1})=\Theta(\tau^{-1})$.

The proof is completed. $\qquad\square$

We conclude the proof of Theorem 5 by showing $g(t)$ is upper bounded.

**Lemma 22.** *Let all conditions in Theorem 5 hold. Then, $g(t)$ is upper bounded.*

*Proof.* $g(t)$ is upper bounded is equivalent to $\sum_{t=1}^{\infty}g(t+1)-g(t)<\infty$. We then prove this lemma by calculating $g(t+1)-g(t)$ directly.

$$g(t+1)-g(t)$$
$$=\frac{\sqrt{\varepsilon}}{2}\|\boldsymbol{r}(t+1)\|^2+\frac{\beta_1}{1-\beta_1}\left\langle\boldsymbol{r}(t+1),(1-\beta_1^t)\sqrt{\varepsilon\mathbb{1}_d+\hat{\boldsymbol{\nu}}(t)}\odot(\boldsymbol{w}(t+1)-\boldsymbol{w}(t))\right\rangle$$
$$-\left(\frac{\sqrt{\varepsilon}}{2}\|\boldsymbol{r}(t)\|^2+\frac{\beta_1}{1-\beta_1}\left\langle\boldsymbol{r}(t),(1-\beta_1^{t-1})\sqrt{\varepsilon\mathbb{1}_d+\hat{\boldsymbol{\nu}}(t-1)}\odot(\boldsymbol{w}(t)-\boldsymbol{w}(t-1))\right\rangle\right)$$
$$-\frac{\beta_1}{1-\beta_1}\langle\boldsymbol{r}(t+1)-\boldsymbol{r}(t),(1-\beta_1^t)\sqrt{\varepsilon\mathbb{1}_d+\hat{\boldsymbol{\nu}}(t)}\odot(\boldsymbol{w}(t+1)-\boldsymbol{w}(t))\rangle$$
$$=\frac{\sqrt{\varepsilon}}{2}\|\boldsymbol{r}(t+1)-\boldsymbol{r}(t)\|^2+\frac{\beta_1}{1-\beta_1}\left\langle\boldsymbol{r}(t),(1-\beta_1^t)\sqrt{\varepsilon\mathbb{1}_d+\hat{\boldsymbol{\nu}}(t)}\odot(\boldsymbol{w}(t+1)-\boldsymbol{w}(t))\right.$$
$$\left.-(1-\beta_1^{t-1})\sqrt{\varepsilon\mathbb{1}_d+\hat{\boldsymbol{\nu}}(t-1)}\odot(\boldsymbol{w}(t)-\boldsymbol{w}(t-1))\right\rangle+\sqrt{\varepsilon}\langle\boldsymbol{r}(t+1)-\boldsymbol{r}(t),\boldsymbol{r}(t)\rangle$$
$$\overset{(\star)}{=}\left\langle\boldsymbol{r}(t),-(1-\beta_1^t)\sqrt{\varepsilon\mathbb{1}_d+\hat{\boldsymbol{\nu}}(t)}\odot(\boldsymbol{w}(t+1)-\boldsymbol{w}(t))-\frac{\eta}{1-\beta}\nabla\mathcal{L}(\boldsymbol{w}(t))\right\rangle$$
$$+\frac{\sqrt{\varepsilon}}{2}\|\boldsymbol{r}(t+1)-\boldsymbol{r}(t)\|^2+\sqrt{\varepsilon}\langle\boldsymbol{r}(t+1)-\boldsymbol{r}(t),\boldsymbol{r}(t)\rangle,$$

where Eq. $(\star)$ is due to a simple rearranging of the update rule of Adam, i.e.,

$$\frac{\beta_1}{1-\beta_1}\left((1-\beta_1^t)\sqrt{\varepsilon\mathbb{1}_d+\hat{\boldsymbol{\nu}}(t)}\odot(\boldsymbol{w}(t+1)-\boldsymbol{w}(t))-(1-\beta_1^{t-1})\sqrt{\varepsilon\mathbb{1}_d+\hat{\boldsymbol{\nu}}(t-1)}\odot(\boldsymbol{w}(t)-\boldsymbol{w}(t-1))\right)$$

$$=-\frac{\eta}{1-\beta_1}\nabla\mathcal{L}(\boldsymbol{w}(t))-(1-\beta_1^t)\sqrt{\varepsilon\mathbb{1}_d+\hat{\boldsymbol{\nu}}(t)}\odot(\boldsymbol{w}(t+1)-\boldsymbol{w}(t)).$$

On the one hand, as $\|\boldsymbol{r}(t+1)-\boldsymbol{r}(t)\|=\|\boldsymbol{w}(t+1)-\boldsymbol{w}(t)-\ln\frac{t+1}{t}\hat{\boldsymbol{w}}\|=\mathcal{O}(t^{-1})$,

$$\sum_{t=1}^{\infty}\frac{\sqrt{\varepsilon}}{2}\|\boldsymbol{r}(t+1)-\boldsymbol{r}(t)\|^2<\infty.$$

On the other hand,

$$\left\langle\boldsymbol{r}(t),-(1-\beta_1^t)\sqrt{\varepsilon\mathbb{1}_d+\hat{\boldsymbol{\nu}}(t)}\odot(\boldsymbol{w}(t+1)-\boldsymbol{w}(t))-\frac{\eta}{1-\beta}\nabla\mathcal{L}(\boldsymbol{w}(t))\right\rangle$$

$$+\sqrt{\varepsilon}\langle\boldsymbol{r}(t+1)-\boldsymbol{r}(t),\boldsymbol{r}(t)\rangle$$

$$=\left\langle\boldsymbol{r}(t),-(1-\beta_1^t)\sqrt{\varepsilon\mathbb{1}_d+\hat{\boldsymbol{\nu}}(t)}\odot(\boldsymbol{w}(t+1)-\boldsymbol{w}(t))-\frac{\eta}{1-\beta}\nabla\mathcal{L}(\boldsymbol{w}(t))\right\rangle$$

$$+\sqrt{\varepsilon}\left\langle\boldsymbol{w}(t+1)-\boldsymbol{w}(t)-\ln\left(\frac{t+1}{t}\right)\hat{\boldsymbol{w}},\boldsymbol{r}(t)\right\rangle$$

$$=\left\langle\boldsymbol{r}(t),-(1-\beta_1^t)\sqrt{\varepsilon\mathbb{1}_d+\hat{\boldsymbol{\nu}}(t)}\odot(\boldsymbol{w}(t+1)-\boldsymbol{w}(t))+\sqrt{\varepsilon}(\boldsymbol{w}(t+1)-\boldsymbol{w}(t))\right\rangle$$

$$+\left\langle\boldsymbol{r}(t),-\sqrt{\varepsilon}\ln\left(\frac{t+1}{t}\right)\hat{\boldsymbol{w}}-\frac{\eta}{1-\beta}\nabla\mathcal{L}(\boldsymbol{w}(t))\right\rangle$$

$$\overset{(\bullet)}{=}\mathcal{O}(\beta_1^t+t^{-2})+\left\langle\boldsymbol{r}(t),-\sqrt{\varepsilon}\ln\left(\frac{t+1}{t}\right)\hat{\boldsymbol{w}}-\frac{\eta}{1-\beta}\nabla\mathcal{L}(\boldsymbol{w}(t))\right\rangle,$$

where Eq. $(\bullet)$ is due to $\hat{\boldsymbol{\nu}}(t)=\mathcal{O}(t^{-2})$.

Furthermore, following exactly the same routine as Lemma 13, we have

$$\sum_{t=1}^{\infty}\left\langle\boldsymbol{r}(t),-\sqrt{\varepsilon}\ln\left(\frac{t+1}{t}\right)\hat{\boldsymbol{w}}-\frac{\eta}{1-\beta}\nabla\mathcal{L}(\boldsymbol{w}(t))\right\rangle<\infty.$$

The proof is completed. $\qquad\square$

# E   Implicit regularization of RMSProp (w/o. r) with decaying learning rate

This section collects the proof of Theorem 6. To begin with, we formally define RMSProp (w/o. r) as follows to facilitate latter analysis: for each $t\in\{0,1,2\cdots\}$, divide the sample set $S$ into $K$ subsets $\{\boldsymbol{B}(Kt+1),\cdots,\boldsymbol{B}(K(t+1))\}$ uniformly and i.i.d., and let

$$\boldsymbol{\nu}(0)=0,\boldsymbol{\nu}(\tau)=\beta_2\boldsymbol{\nu}(\tau-1)+(1-\beta_2)\left(\nabla\mathcal{L}_{\boldsymbol{B}(\tau)}(\boldsymbol{w}(\tau-1))\right)^2,$$

$$(\text{RMSProp (w/o. r))}:\ \boldsymbol{w}(\tau)=\boldsymbol{w}(\tau-1)-\eta_\tau\frac{\nabla\mathcal{L}_{\boldsymbol{B}(\tau)}(\boldsymbol{w}(\tau-1))}{\sqrt{\boldsymbol{\nu}(\tau)+\varepsilon\mathbb{1}_d}}. \tag{64}$$

Here the $\mathcal{L}_{\boldsymbol{B}(\tau)}$ is the individual loss average over $\boldsymbol{B}(\tau)$, i.e., $\mathcal{L}_{\boldsymbol{B}(\tau)}(\boldsymbol{w})=\frac{\sum_{(\boldsymbol{x},\boldsymbol{y})\in\boldsymbol{B}(\tau)}\ell(-\boldsymbol{y}\langle\boldsymbol{w},\boldsymbol{x}\rangle)}{b}$, where $b=\frac{N}{K}$ is the batch size. With Eq. (64), we restate the loss convergence result in [33] as the following proposition.

**Proposition 1** (Corollary 4.1 in [33], restated). *Suppose $\ell$ is non-negative and L-smooth. Furthermore, assume that there exists a constant D, s.t., $\forall t\geq 0$, $\forall\boldsymbol{w}\in\mathbb{R}^d$*

$$\sum_{\tau=Kt+1}^{K(t+1)}\|\nabla\mathcal{L}_{\boldsymbol{B}(\tau)}(\boldsymbol{w})\|^2\leq D\|N\nabla\mathcal{L}(\boldsymbol{w})\|^2, \tag{65}$$

*and*

$$T_2\left(\beta_2\right) \triangleq \sqrt{\frac{10dK}{\beta_2^K}}dKD\left((1-\beta_2)\frac{\left(\frac{4K^2}{\beta_2^K}-1\right)}{2}+\left(\frac{1}{\sqrt{\beta_2^K}}-1\right)\right) \leq \frac{\sqrt{2}-1}{2\sqrt{2}}.$$

*Then, RMSProp (w/o. r) with decaying learning rate $\eta_\tau = \frac{\eta_1}{\sqrt{\tau}}$ satisfies*

$$\sum_{t=0}^{T}\frac{1}{\sqrt{t+1}}\|\nabla\mathcal{L}(\boldsymbol{w}(Kt))\| = \mathcal{O}(\ln(T)).$$

Combining Assumptions 1 and 2, we immediately get the following corollary:

**Corollary 8.** *Let Assumptions 1, 2, and 3. (S) hold. Let*

$$T_2\left(\beta_2\right) \triangleq \sqrt{\frac{10dK}{\beta_2^K}}\frac{dK}{b^2\gamma^2}\left((1-\beta_2)\frac{\left(\frac{4K^2}{\beta_2^K}-1\right)}{2}+\left(\frac{1}{\sqrt{\beta_2^K}}-1\right)\right) \leq \frac{\sqrt{2}-1}{2\sqrt{2}}.$$

*Then, there exists a positive constant $C_1$ and $C_3$ independent of random sampling, s.t.,*

$$\sum_{t=0}^{T}\frac{1}{\sqrt{t+1}}\|\nabla\mathcal{L}(\boldsymbol{w}(Kt))\| \leq C_1\ln(T), \forall T \geq 0,$$

*and*

$$\mathcal{L}(\boldsymbol{w}(K(t+1))) \leq \mathcal{L}(\boldsymbol{w}(Kt)) - \frac{C_3}{\sqrt{t+1}}\|\nabla\mathcal{L}(\boldsymbol{w}(Kt))\| + \frac{C_1C_3}{t+1}.$$

Applying relationship between $\ell$ and $\ell'$ of the exponentially-tailed loss, we can further obtain the loss convergent rate.

**Lemma 23.** *Let all the conditions in Theorem 6 hold. Then, we have*

$$\mathcal{L}(\boldsymbol{w}(Kt)) = \mathcal{O}\left(\frac{1}{\sqrt{t}}\right).$$

*Proof.* To begin with, we show that there exists an increasing positive integer sequence $\{t_i\}_{i=1}^{\infty}$, and $\|\nabla\mathcal{L}(\boldsymbol{w}(Kt_i))\| \leq 2C_1\frac{1}{\sqrt{t_i+1}}$ by reduction to absurdity. Otherwise, suppose there exists a positive integer $T_1$, such that $\|\nabla\mathcal{L}(\boldsymbol{w}(Kt))\| > 2C_1\frac{1}{\sqrt{t+1}}, \forall t \geq T_1$. Therefore, for $T \geq T_1$, we have

$$\sum_{t=0}^{T}\frac{1}{\sqrt{t+1}}\|\nabla\mathcal{L}(\boldsymbol{w}(Kt))\| \geq \sum_{t=T_1}^{T}\frac{1}{\sqrt{t+1}}\|\nabla\mathcal{L}(\boldsymbol{w}(Kt))\|$$

$$\geq 2C_1\sum_{t=T_1}^{T}\frac{1}{t+1} \geq 2C_1\ln\frac{T+2}{T_1+1}.$$

Let $T$ be large enough, we have $2C_1\ln\frac{T+2}{T_1+1} > C_1\ln T$, which contradicts Corollary 8.

Denote $\boldsymbol{T} = \{t > 0 : \|\nabla\mathcal{L}(\boldsymbol{w}(Kt))\| \leq 2C_1\frac{1}{\sqrt{t+1}}\}$. By the above discussion, $\boldsymbol{T}$ contains an increasing positive integer sequence. We then prove that if $s \in \boldsymbol{T}$ and $s > s^{\boldsymbol{T}}$, where $s^{\boldsymbol{T}}$ is defined as

$$s^{\boldsymbol{T}} \triangleq \max\left\{\left(\frac{2C_1}{C_g}\right)^2 - 1, (C_3\gamma)^2 - 1, \left(\frac{9C_1}{\gamma C_l}\right)^2 - 1, 4, \frac{4}{C_3\gamma(\sqrt{\frac{3}{2}}-1)}\ln\frac{18\sqrt{\frac{3}{2}}}{\gamma} - 2\right\}$$

($C_l$ and $C_g$ is defined in Corollary 2), there exists $s < r \leq 2s$, and $r \in \boldsymbol{T}$. We slightly abuse the notation and let $r = \inf\{t : t \in \boldsymbol{T}, t > s\}$. If $r = s+1$, this claim trivially holds. Otherwise, as $s \in \boldsymbol{T}$, we have

$$\|\nabla\mathcal{L}(\boldsymbol{w}(Ks))\| \leq 2C_1\frac{1}{\sqrt{s+1}} \leq C_g,$$

which by Corollary 2 further leads to

$$\mathcal{L}(\boldsymbol{w}(Ks)) \leq \frac{4}{\gamma}\|\nabla\mathcal{L}(\boldsymbol{w}(Ks))\|.$$

Therefore, by Corollary 8, we have

$$\mathcal{L}(\boldsymbol{w}(K(s+1))) \leq \mathcal{L}(\boldsymbol{w}(Ks)) + \frac{C_1 C_3}{s+1} \leq \frac{8C_1}{\gamma}\frac{1}{\sqrt{s+1}} + \frac{C_1 C_3}{s+1} \leq \frac{9C_1}{\gamma}\frac{1}{\sqrt{s+1}},$$

which by $s > s^{\boldsymbol{T}}$ further leads to

$$\mathcal{L}(\boldsymbol{w}(K(s+1))) \leq C_l,$$

and thus $\mathcal{L}(\boldsymbol{w}(K(s+1))) \leq \frac{4}{\gamma}\|\nabla\mathcal{L}(\boldsymbol{w}(K(s+1)))\|$. As $(s+1) \notin \boldsymbol{T}$, we have

$$\|\nabla\mathcal{L}(\boldsymbol{w}(K(s+1)))\| > 2C_1\frac{1}{\sqrt{s+2}},$$

which leads to

$$\mathcal{L}(\boldsymbol{w}(K(s+2))) \leq \mathcal{L}(\boldsymbol{w}(K(s+1))) - \frac{C_3}{2\sqrt{s+2}}\|\nabla\mathcal{L}(\boldsymbol{w}(K(s+1)))\| \leq \left(1 - \frac{C_3\gamma}{8\sqrt{s+2}}\right)\mathcal{L}(\boldsymbol{w}(K(s+1))),$$

and thus $\mathcal{L}(\boldsymbol{w}(K(s+2))) \leq C_l$. By the inductive method, we have for any $j \in \{s+2, \cdots, r\}$,

$$\mathcal{L}(\boldsymbol{w}(K(j))) \leq \Pi_{i=s+2}^{j}\left(1 - \frac{C_3\gamma}{8\sqrt{i}}\right)\mathcal{L}(\boldsymbol{w}(K(s+1))) \leq e^{-\sum_{i=s+2}^{j}\frac{C_3\gamma}{8\sqrt{i}}}\mathcal{L}(\boldsymbol{w}(K(s+1)))$$

$$\leq e^{-\frac{C_3\gamma(\sqrt{j+1}-\sqrt{s+2})}{4}}\mathcal{L}(\boldsymbol{w}(K(s+1))) \leq e^{-\frac{C_3\gamma(\sqrt{j+1}-\sqrt{s+2})}{4}}\frac{9C_1}{\gamma\sqrt{s+1}}.$$

If $r > 2s$, applying $j = 2s$ into the above equation leads to

$$\mathcal{L}(\boldsymbol{w}(K(2s))) \leq e^{-\frac{C_3\gamma(\sqrt{2s+1}-\sqrt{s+2})}{4}}\frac{9C_1}{\gamma\sqrt{s+1}} \leq C_l,$$

and

$$\|\nabla\mathcal{L}(\boldsymbol{w}(K(2s)))\| \leq e^{-\frac{C_3\gamma(\sqrt{2s+1}-\sqrt{s+2})}{4}}\frac{36C_1}{\gamma\sqrt{s+1}} \leq \frac{2C_1}{\sqrt{2s+1}},$$

which leads to $2s \in \boldsymbol{T}$, and contradicts the definition of $r$. Therefore, we have $r \leq 2s$.

As $\boldsymbol{T}$ contains an increasing integer sequence, there exists an $s_0$, s.t., $s_0 \in \boldsymbol{T}$ and $s_0 > s^{\boldsymbol{T}}$. Let $t$ be any positive integer larger than $s_0$ and let $t'$ be the largest integer smaller than $t$ and belongs to $\boldsymbol{T}$. We have $t' \geq s_0$, and $t \leq 2t'$ by the above discussion. Therefore, we have

$$\mathcal{L}(Kt) \leq \frac{9C_1}{\gamma}\frac{1}{\sqrt{t'+1}} \leq \frac{9\sqrt{2}C_1}{\gamma}\frac{1}{\sqrt{t+2}}.$$

The proof is completed. $\qquad\qquad\square$

As a corollary, we can obtain an asymptotic estimation of $\nabla\mathcal{L}_{\boldsymbol{B}(\tau)}(\boldsymbol{w}(\tau))$.

**Corollary 9.** *Let all the conditions in Theorem 6 hold. Then, we have*

$$\|\nabla\mathcal{L}_{\boldsymbol{B}(\tau)}(\boldsymbol{w}(\tau))\| = \mathcal{O}\left(\frac{1}{\sqrt{\tau}}\right).$$

*Proof.* Let $\tau > K(s^{\boldsymbol{T}}+1)$, where $s^{\boldsymbol{T}}$ is defined as Lemma 23. Let $t = \lceil\frac{\tau}{K}\rceil > s^{\boldsymbol{T}}$ and $s = \tau - Kt$. Then, we have

$$\|\nabla\mathcal{L}(Kt)\| \leq C_l,$$

and

$$\|\nabla\mathcal{L}(\boldsymbol{w}(Kt))\| \leq 4\mathcal{L}(\boldsymbol{w}(Kt)) \leq \frac{36\sqrt{2}C_1}{\gamma}\frac{1}{\sqrt{t+2}}.$$

On the other hand, we have

$$\|\nabla\mathcal{L}_{\boldsymbol{B}(\tau)}(\boldsymbol{w}(Kt))\| \leq \frac{1}{\gamma}\frac{N}{b}\|\nabla\mathcal{L}(\boldsymbol{w}(Kt))\| \leq \frac{N}{b\gamma}\frac{36\sqrt{2}C_1}{\gamma}\frac{1}{\sqrt{t+2}}.$$

As $\mathcal{L}_{\boldsymbol{B}(\tau)}$ is $H$ smooth, we further have

$$\|\nabla\mathcal{L}_{\boldsymbol{B}(\tau)}(\boldsymbol{w}(Kt)) - \nabla\mathcal{L}_{\boldsymbol{B}(\tau)}(\boldsymbol{w}(\tau))\| \leq H\left\|\sum_{i=0}^{s-1}\eta_{Kt+i+1}\frac{\nabla\mathcal{L}_{\boldsymbol{B}(Kt+i+1)}(\boldsymbol{w}(Kt+i))}{\sqrt{\hat{\boldsymbol{\nu}}(Kt+i+1)} + \varepsilon\mathbb{1}_d}\right\|$$
$$\leq \frac{KH}{\sqrt{1-\beta_2}\sqrt{Kt+1}}.$$

Combining the above two equations, we have

$$\|\nabla\mathcal{L}_{\boldsymbol{B}(\tau)}(\boldsymbol{w}(\tau))\| \leq \frac{KH}{\sqrt{1-\beta_2}\sqrt{Kt+1}} + \frac{N}{b\gamma}\frac{36\sqrt{2}C_1}{\gamma}\frac{1}{\sqrt{t+2}}$$
$$\leq \frac{KH}{\sqrt{1-\beta_2}\sqrt{\tau-K}} + \frac{N}{b\gamma}\frac{36\sqrt{2}C_1}{\gamma}\frac{\sqrt{K}}{\sqrt{\tau+K}}.$$

The proof is completed. $\qquad\square$

Using Corollary 9, we are now able to prove Theorem 6.

*Proof of Theorem 6 (for almost every dataset).* To begin with, define $\boldsymbol{r}(t) \triangleq \boldsymbol{w}(t) - \frac{1}{2}\ln\left(\frac{\eta}{K}t\right)\hat{\boldsymbol{w}} - \tilde{\boldsymbol{w}} - \frac{1}{2}\boldsymbol{n}(t)$, where $\tilde{\boldsymbol{w}}$ is the solution of Eq. (10) with $C_3 = \frac{2}{N\sqrt{\varepsilon}}$, and $\boldsymbol{n}(t)$ is given by Lemma 5. As in the case of GDM, SGDM, and Adam (w/s), $\boldsymbol{w}(t) - \frac{1}{2}\ln\left(\frac{\eta}{K}t\right)$ has bounded norm if and only if $\boldsymbol{r}(t)$ has bounded norm. Also, it is a sufficient condition to ensure $\|\boldsymbol{r}(t)\|$ is bounded that both $A \triangleq \sum_{t=1}^{\infty}\|\boldsymbol{r}(t+1)-\boldsymbol{r}(t)\|^2 < \infty$ and $B \triangleq \sum_{t=1}^{\infty}\langle\boldsymbol{r}(t+1)-\boldsymbol{r}(t),\boldsymbol{r}(t)\rangle < \infty$.
As for $A$, we have

$$A = \sum_{t=1}^{\infty}\|\boldsymbol{r}(t)-\boldsymbol{r}(t-1)\|^2$$
$$= \sum_{t=1}^{\infty}\left\|\boldsymbol{w}(t+1)-\boldsymbol{w}(t)-\frac{1}{2}\ln\left(\frac{t+1}{t}\right)\hat{\boldsymbol{w}}-\frac{1}{2}\boldsymbol{n}(t+1)+\frac{1}{2}\boldsymbol{n}(t)\right\|^2$$
$$\leq 3\left(\sum_{t=1}^{\infty}\|\boldsymbol{w}(t+1)-\boldsymbol{w}(t)\|^2 + \frac{1}{2}\sum_{t=1}^{\infty}\ln\left(\frac{t+1}{t}\right)\|\hat{\boldsymbol{w}}\|^2 + \sum_{t=1}^{\infty}\frac{1}{4}\|-\boldsymbol{n}(t+1)+\boldsymbol{n}(t)\|^2\right)$$
$$\overset{(*)}{<}\infty,$$

where Inequality $(*)$ is due to

$$\|\boldsymbol{w}(t+1)-\boldsymbol{w}(t)\| = \frac{\eta_1}{\sqrt{t+1}}\left\|\frac{\nabla\mathcal{L}_{\boldsymbol{B}(t)}(\boldsymbol{w}(t))}{\sqrt{\varepsilon\mathbb{1}_d + \hat{\boldsymbol{\nu}}(t+1)}}\right\| \leq \frac{\eta_1}{\sqrt{t+1}\sqrt{\varepsilon}}\|\nabla\mathcal{L}_{\boldsymbol{B}(t)}(\boldsymbol{w}(t))\| = \mathcal{O}\left(\frac{1}{t}\right),$$

$\ln\left(\frac{t+1}{t}\right) = \mathcal{O}\left(\frac{1}{t}\right)$, and $\|-\boldsymbol{n}(t+1)+\boldsymbol{n}(t)\| = \mathcal{O}\left(\frac{1}{t}\right)$.

As for $B$, we have

$$B = \sum_{t=1}^{\infty} \left\langle \boldsymbol{w}(t+1) - \boldsymbol{w}(t) - \frac{1}{2}\ln\left(\frac{t+1}{t}\right)\hat{\boldsymbol{w}} - \frac{1}{2}\boldsymbol{n}(t+1) + \frac{1}{2}\boldsymbol{n}(t), \boldsymbol{r}(t) \right\rangle$$

$$= \sum_{t=1}^{\infty} \left\langle \boldsymbol{w}(t+1) - \boldsymbol{w}(t) - \frac{N}{2bt}\sum_{i:\boldsymbol{x}_i \in \boldsymbol{B}(t) \cap \boldsymbol{S}_s} v_i \boldsymbol{x}_i, \boldsymbol{r}(t) \right\rangle$$

$$= \sum_{t=1}^{\infty} \left\langle -\frac{1}{\sqrt{t+1}}\frac{\nabla\mathcal{L}_{\boldsymbol{B}(t)}(\boldsymbol{w}(t))}{\sqrt{\varepsilon\mathbb{1}_d + \hat{\boldsymbol{\nu}}(t+1)}} - \frac{N}{2bt}\sum_{i:\boldsymbol{x}_i \in \boldsymbol{B}(t) \cap \boldsymbol{S}_s} v_i \boldsymbol{x}_i, \boldsymbol{r}(t) \right\rangle$$

$$= \sum_{t=1}^{\infty} \left\langle -\frac{1}{\sqrt{t+1}}\frac{\nabla\mathcal{L}_{\boldsymbol{B}(t)}(\boldsymbol{w}(t))}{\sqrt{\varepsilon\mathbb{1}_d + \hat{\boldsymbol{\nu}}(t+1)}} + \frac{1}{\sqrt{t}}\frac{\nabla\mathcal{L}(\boldsymbol{w}(t))}{\sqrt{\varepsilon\mathbb{1}_d + \hat{\boldsymbol{\nu}}(t+1)}}, \boldsymbol{r}(t) \right\rangle$$

$$+ \sum_{t=1}^{\infty} \left\langle -\frac{1}{\sqrt{t}}\frac{\nabla\mathcal{L}_{\boldsymbol{B}(t)}(\boldsymbol{w}(t))}{\sqrt{\varepsilon\mathbb{1}_d + \hat{\boldsymbol{\nu}}(t+1)}} + \frac{1}{\sqrt{t}\sqrt{\varepsilon}}\nabla\mathcal{L}_{\boldsymbol{B}(t)}(\boldsymbol{w}(t)), \boldsymbol{r}(t) \right\rangle$$

$$+ \sum_{t=1}^{\infty} \left\langle -\frac{1}{\sqrt{t}\sqrt{\varepsilon}}\nabla\mathcal{L}_{\boldsymbol{B}(t)}(\boldsymbol{w}(t)) - \frac{N}{2bt}\sum_{i:\boldsymbol{x}_i \in \boldsymbol{B}(t) \cap \boldsymbol{S}_s} v_i \boldsymbol{x}_i, \boldsymbol{r}(t) \right\rangle. \tag{66}$$

On the other hand, as

$$\|\boldsymbol{w}(t)\| \le \|\boldsymbol{w}(1)\| + \sum_{s=1}^{t-1}\frac{1}{\sqrt{s+1}}\left\|\frac{\nabla\mathcal{L}_{\boldsymbol{B}(t)}(\boldsymbol{w}(s))}{\sqrt{\varepsilon\mathbb{1}_d + \hat{\boldsymbol{\nu}}(s+1)}}\right\| = \mathcal{O}(\ln(t)),$$

we have $\|\boldsymbol{r}(t)\| = \mathcal{O}(\ln(t))$. Also, we have $\|\nabla\mathcal{L}_{\boldsymbol{B}(t)}(\boldsymbol{w}(t))^2\| = \mathcal{O}(\frac{1}{t})$, and thus $\left\|\frac{1}{\sqrt{\varepsilon\mathbb{1}_d + \hat{\boldsymbol{\nu}}(t)}} - \frac{1}{\sqrt{\varepsilon\mathbb{1}_d}}\right\| = \mathcal{O}(\frac{1}{t})$. Combining these estimations, we have

$$\left\langle -\frac{1}{\sqrt{t+1}}\frac{\nabla\mathcal{L}_{\boldsymbol{B}(t)}(\boldsymbol{w}(t))}{\sqrt{\varepsilon\mathbb{1}_d + \hat{\boldsymbol{\nu}}(t+1)}} + \frac{1}{\sqrt{t}}\frac{\nabla\mathcal{L}_{\boldsymbol{B}(t)}(\boldsymbol{w}(t))}{\sqrt{\varepsilon\mathbb{1}_d + \hat{\boldsymbol{\nu}}(t+1)}}, \boldsymbol{r}(t) \right\rangle = \mathcal{O}\left(\frac{\ln t}{t^{\frac{5}{2}}}\right).$$

and

$$\left\langle -\frac{1}{\sqrt{t}}\frac{\nabla\mathcal{L}_{\boldsymbol{B}(t)}(\boldsymbol{w}(t))}{\sqrt{\varepsilon\mathbb{1}_d + \hat{\boldsymbol{\nu}}(t+1)}} + \frac{1}{\sqrt{t}\sqrt{\varepsilon}}\nabla\mathcal{L}_{\boldsymbol{B}(t)}(\boldsymbol{w}(t)), \boldsymbol{r}(t) \right\rangle = \mathcal{O}\left(\frac{\ln t}{t^2}\right),$$

Therefore,

$$\sum_{t=1}^{\infty} \left\langle -\frac{1}{\sqrt{t+1}}\frac{\nabla\mathcal{L}_{\boldsymbol{B}(t)}(\boldsymbol{w}(t))}{\sqrt{\varepsilon\mathbb{1}_d + \hat{\boldsymbol{\nu}}(t+1)}} + \frac{1}{\sqrt{t}}\frac{\nabla\mathcal{L}_{\boldsymbol{B}(t)}(\boldsymbol{w}(t))}{\sqrt{\varepsilon\mathbb{1}_d + \hat{\boldsymbol{\nu}}(t+1)}}, \boldsymbol{r}(t) \right\rangle < \infty,$$

$$\sum_{t=1}^{\infty} \left\langle -\frac{1}{\sqrt{t}}\frac{\nabla\mathcal{L}_{\boldsymbol{B}(t)}(\boldsymbol{w}(t))}{\sqrt{\varepsilon\mathbb{1}_d + \hat{\boldsymbol{\nu}}(t+1)}} + \frac{1}{\sqrt{t}\sqrt{\varepsilon}}\nabla\mathcal{L}_{\boldsymbol{B}(t)}(\boldsymbol{w}(t)), \boldsymbol{r}(t) \right\rangle < \infty.$$

As for the last term in Eq. (66), we have

$$\left\langle -\frac{1}{\sqrt{t}\sqrt{\varepsilon}}\nabla\mathcal{L}_{\boldsymbol{B}(t)}(\boldsymbol{w}(t)) - \frac{N}{2bt}\sum_{i:\boldsymbol{x}_i \in \boldsymbol{B}(t) \cap \boldsymbol{S}_s} v_i \boldsymbol{x}_i, \boldsymbol{r}(t) \right\rangle$$

$$= \frac{1}{b\sqrt{t}\sqrt{\varepsilon}}\sum_{i:\boldsymbol{x}_i \in \boldsymbol{B}(s) \cap \boldsymbol{S}_s}\left(-\ell'(\langle\boldsymbol{w}(t),\boldsymbol{x}_i\rangle) - \frac{1}{\sqrt{t}}e^{-\langle\tilde{\boldsymbol{w}},\boldsymbol{x}_i\rangle}\right)\langle\boldsymbol{r}(t),\boldsymbol{x}_i\rangle$$

$$+ \frac{1}{b\sqrt{t}\sqrt{\varepsilon}}\sum_{i:\boldsymbol{x}_i \in \boldsymbol{B}(s) \cap \boldsymbol{S}_s^c}\langle\boldsymbol{r}(t), -\ell'(\langle\boldsymbol{w}(t),\boldsymbol{x}_i\rangle)\boldsymbol{x}_i\rangle.$$

Then, following the same routine as Lemma 15, we have

$$\sum_{t=1}^{\infty} \left\langle -\frac{1}{\sqrt{t}\sqrt{\varepsilon}}\nabla\mathcal{L}_{\boldsymbol{B}(t)}(\boldsymbol{w}(t)) - \frac{N}{2bt}\sum_{i:\boldsymbol{x}_i \in \boldsymbol{B}(t) \cap \boldsymbol{S}_s} v_i \boldsymbol{x}_i, \boldsymbol{r}(t) \right\rangle < \infty.$$

The proof is completed. □

# F  Applications & Extensions

## F.1  Deriving the conclusion for every dataset

In Sections 4, 5, and 6, we only derive the implicit regularization for almost every dataset, but not all the separable datasets. In this section, we show that the "almost every" condition can be removed as the following theorem.

**Theorem 7.** *We have the following conclusions:*

- *For GDM, let all the conditions in Theorem 2 hold. Then, GDM converges to the $L^2$ max-margin solution;*

- *For SGDM sampling with replacement, let all the conditions in Theorem 3 hold. Then, SGDM (w/. r) converges to the $L^2$ max-margin solution (the same for SGDM (w/o. r), except a different learning rate upper bound);*

- *For deterministic Adam, let all the conditions in Theorem 5 hold. Then, Adam (w/s) converges to the $L^2$ max-margin solution;*

- *For RMSProp (w/o. r), let all the conditions in Theorem 6 hold. Then, RMSProp (w/o. r) converges to the $L^2$ max-margin solution.*

It can be easily observe that to prove Theorem 7, the analysis of Stage I of every optimizer still works. Therefore, we only need to change Stage II. As the analyses of Stage II are highly overlapped for different optimizers, we only provide a proof of GDM.

To begin with, we present some notations and results on the structure of the separable dataset from [34].

Let $\bar{\boldsymbol{S}}_0 = \{1, \cdots, N\}$, and $\bar{\mathbf{P}}_0 = \mathbb{I}_{d \times d}$. We then recursively define the index sets $\boldsymbol{S}_m^+$, $\boldsymbol{S}_m^=$, $\boldsymbol{S}_m$, and $\bar{\boldsymbol{S}}_m$:

$$\boldsymbol{S}_m^+ = \left\{ i \in \bar{\boldsymbol{S}}_{m-1} \mid \langle \hat{\boldsymbol{w}}_m, \bar{\mathbf{P}}_{m-1} \boldsymbol{x}_i \rangle > 1 \right\},$$

$$\boldsymbol{S}_m^= = \left\{ i \in \bar{\boldsymbol{S}}_{m-1} \mid \langle \hat{\boldsymbol{w}}_m, \bar{\mathbf{P}}_{m-1} \boldsymbol{x}_i \rangle = 1 \right\} = \bar{\boldsymbol{S}}_{m-1}/\boldsymbol{S}_m^+,$$

$$\boldsymbol{S}_m = \left\{ i \in \boldsymbol{S}_m^= \mid \exists \boldsymbol{\alpha} \in \mathbb{R}_{\geq 0}^N : \hat{\boldsymbol{w}}_m = \sum_{k=1}^N \alpha_k \bar{\mathbf{P}}_{m-1} \boldsymbol{x}_k, \alpha_i > 0, \forall j \notin \mathcal{S}_m^= : \alpha_j = 0 \right\},$$

$$\bar{\boldsymbol{S}}_m = \boldsymbol{S}_m^= / \boldsymbol{S}_m.$$

where $\bar{\mathbf{P}}_m = \bar{\mathbf{P}}_{m-1} \left( \mathbf{I}_d - \boldsymbol{S}_{\boldsymbol{S}_m} \boldsymbol{S}_{\boldsymbol{S}_m}^\dagger \right)$ (we also denote $\mathbf{P}_m = \mathbb{I}_{d \times d} - \bar{\mathbf{P}}_m$), and $\hat{\boldsymbol{w}}_m$ is defined as the max-margin solution of dataset $\bar{\mathbf{P}}_{m-1} \boldsymbol{S}_{\bar{\boldsymbol{S}}_{m-1}}$ (that is, the transferred data $x_i$ with index in $\bar{\boldsymbol{S}}_{m-1}$ projected through matrix $\bar{\mathbf{P}}_{m-1}$):

$$\hat{\boldsymbol{w}}_m = \operatorname*{argmin}_{\boldsymbol{w} \in \mathbb{R}^d} \|\boldsymbol{w}\|^2, \text{ s.t. } \langle \boldsymbol{w}, \bar{\mathbf{P}}_{m-1} \boldsymbol{x}_i \rangle \geq 1, \forall i \in \bar{\boldsymbol{S}}_{m-1}. \tag{67}$$

The existence of the $\boldsymbol{\alpha}$ is guaranteed by the KKT condition of Eq. (67). The above procedure will produce a sequence $\hat{\boldsymbol{w}}_1$, $\hat{\boldsymbol{w}}_2$, $\cdots$, and will stop at $\hat{\boldsymbol{w}}_M$ if $\bar{\boldsymbol{S}}_M$ is empty (if the sequence is infinite, we let $M = \infty$). For every $i \leq M$, we have $\hat{\boldsymbol{w}}_i$ is non-zero, and $\boldsymbol{S}_i$ is non-empty, which leads to $|\bar{\boldsymbol{S}}_i| < |\bar{\boldsymbol{S}}_{i+1}|$, and $M \leq N$.

The following lemma characterize the structure of the dataset.

**Lemma 24** (Lemma 17, [34]). $\forall \boldsymbol{\beta} \in \mathbb{R}_{>0}^{|\boldsymbol{S}_1|}$, *we can find a unique $\tilde{\boldsymbol{w}}_1$, such that*

$$\sum_{i \in \boldsymbol{S}_1} \boldsymbol{x}_i \beta_i \exp\left(-\langle \boldsymbol{x}_i, \tilde{\boldsymbol{w}}_1 \rangle\right) = \hat{\boldsymbol{w}}_1,$$

*and $\tilde{\boldsymbol{w}}_1 \in \text{Col}(\boldsymbol{S}_{\boldsymbol{S}_1})$.*

With Lemma 24, we then define $\tilde{\boldsymbol{w}}_m$ as the solution of

$$\frac{\eta}{1-\beta} \sum_{i \in \mathcal{S}_m} \exp\left(-\sum_{k=1}^{m}\langle \tilde{\boldsymbol{w}}_k, \boldsymbol{x}_i\rangle\right) \bar{\mathbf{P}}_{m-1}\boldsymbol{x}_i = \hat{\boldsymbol{w}}_m,$$

with $\mathbf{P}_{m-1}\tilde{\boldsymbol{w}}_m = 0$ and $\bar{\mathbf{P}}_m\tilde{\boldsymbol{w}}_m = 0$. We also define $\tilde{\boldsymbol{w}} = \sum_{m=1}^{M} \tilde{\boldsymbol{w}}_m$.

We then have the following lemma:

**Lemma 25** (Lemma 18, [34]). $\forall m > k \geq 1$, the equations

$$\sum_{i \in \boldsymbol{S}_m} \exp\left(-\langle \tilde{\boldsymbol{w}}, \boldsymbol{x}_i\rangle\right) \mathbf{P}_{m-1}\boldsymbol{x}_i = \sum_{k=1}^{m-1} \left[\sum_{i \in \mathcal{S}_k} \exp\left(-\langle \tilde{\boldsymbol{w}}, \boldsymbol{x}_i\rangle\right) \boldsymbol{x}_i \boldsymbol{x}_i^{\top}\right] \check{\boldsymbol{w}}_{k,m}$$

under the constraints $\mathbf{P}_{k-1}\check{\boldsymbol{w}}_{k,m} = 0$ and $\bar{\mathbf{P}}_k\check{\boldsymbol{w}}_{k,m} = 0$ have the unique solution $\check{\boldsymbol{w}}_{k,m}$.

We then denote

$$\boldsymbol{r}(t) = \boldsymbol{w}(t) - \hat{\boldsymbol{w}}_1 \log(t) - \tilde{\boldsymbol{w}} - \tau(t),$$

where $\tau(t) = \left(\sum_{m=2}^{M} \hat{\boldsymbol{w}}_m \log^{\circ m}(t) + \sum_{m=1}^{M}\sum_{k=1}^{m-1} \frac{\check{\boldsymbol{w}}_{k,m}}{\prod_{r=k}^{m-1}\log^{\circ r}(t)}\right)$

Similar to Eq. 28, we define

$$g(t) \triangleq \frac{1}{2}\|\boldsymbol{r}(t)\|^2 + \frac{\beta}{1-\beta}\langle \boldsymbol{r}(t), \boldsymbol{w}(t) - \boldsymbol{w}(t-1)\rangle - \frac{\beta}{1-\beta}\sum_{\tau=2}^{t}\langle \boldsymbol{r}(\tau) - \boldsymbol{r}(\tau-1), \boldsymbol{w}(\tau) - \boldsymbol{w}(\tau-1)\rangle.$$

We then have the following lemma parallel to Lemma 11:

**Lemma 26.** Let all conditions in Theorem 2 hold. We have $\sup_t g(t)$ is finite. Furthermore, $\sup_t \|\boldsymbol{r}(t)\|$ is finite if and only if $\sup_t g(t)$ is finite, and consequently $\sup_t \|\boldsymbol{r}(t)\|$ is finite.

*Proof.* The proof of the second argument follows the same routine as the proof of the first argument in Lemma 11, and we omit it here.

As for the first argument, we have

$$\sum_{t=1}^{\infty}(g(t+1) - g(t)) = \frac{1}{2}\sum_{t=1}^{\infty}\|\boldsymbol{r}(t+1) - \boldsymbol{r}(t)\|^2$$
$$+ \sum_{t=1}^{\infty}\left\langle \boldsymbol{r}(t), -\frac{\eta}{1-\beta}\nabla\mathcal{L}(\boldsymbol{w}(t)) - \ln\frac{t+1}{t}\hat{\boldsymbol{w}}_1 - (\tau(t+1) - \tau(t))\right\rangle,$$

where the first term can be shown to be finite similar to Lemma 11, while the second term is finite by Lemma 14 in [34].

The proof is completed. □

## F.2  Implicit regularization of SGDM (w/o. r)

This section provides formal description of the implicit regularization of SGDM (w/o. r) and its corresponding proof. To begin with, we would like to provide a formal definition of SGDM (w/o. r). SGDM (w/o. r) differs from SGDM by applying sampling without replacement to obtain $\boldsymbol{B}(t)$ in Eq. (2). Specifically, let $K = \frac{N}{b}$. For any $T \geq 0$, we call time series $\{KT + 1, \cdots, KT + K\}$ the $(T+1)$-th epoch, and during the $T+1$-th epoch, the dataset $\boldsymbol{S}$ is randomly uniformly divided into $K$ parts $\{\boldsymbol{B}(KT + 1), \cdots, \boldsymbol{B}(KT + K)\}$, with $\bigcup_{t=KT+1}^{KT+K} \boldsymbol{B}(t) = \boldsymbol{S}$. The implicit regularization of SGDM (w/o. r) is then stated as the following theorem:

**Theorem 8.** Let Assumptions 1, 2, and 3. (S) hold. Let learning rate $\eta$ be small enough, and $\beta \in [0, 1)$. Then, for almost every dataset $\boldsymbol{S}$, SGDM (w/o. r) satisfies $\boldsymbol{w}(t) - \ln(t)\hat{\boldsymbol{w}}$ is bounded as $t \to \infty$, and $\lim_{t\to\infty}\frac{\boldsymbol{w}(t)}{\|\boldsymbol{w}(t)\|} = \frac{\hat{\boldsymbol{w}}}{\|\hat{\boldsymbol{w}}\|}$.

The without-replacement sampling method leads to the direction of every trajectory of mini-SGDM converge to the max-margin solution, compared to the same conclusion holds for SGDM a.s.. We prove the theorem following the same framework of GDM, by proceeding with two stages.

**Stage I.** The following lemma proves $\mathcal{L}(\boldsymbol{u}(t))$ is an Lyapunov function for SGDM (w/o. r) and without the a.s. condition.

**Lemma 27.** *Let all conditions in Theorem 8 hold. Then, we have*

$$\mathcal{L}(\boldsymbol{u}(t+1)) \leq \mathcal{L}(\boldsymbol{u}(1)) - \Omega(\eta) \sum_{s=1}^{t} \|\nabla\mathcal{L}(\boldsymbol{w}(s))\|^2.$$

*Proof.* By the Taylor Expansion of $\mathcal{L}(\boldsymbol{u}(t+1))$ at $\boldsymbol{u}(t)$, we have

$$\mathcal{L}(\boldsymbol{u}(KT+T+1))$$
$$\leq \mathcal{L}(\boldsymbol{u}(KT+1)) - \eta\left\langle \nabla\mathcal{L}(\boldsymbol{u}(KT+1)), \sum_{t=1}^{K}\nabla\mathcal{L}_{\boldsymbol{B}(t+KT)}(\boldsymbol{w}(t+KT))\right\rangle$$
$$+ \frac{H\eta^2}{2}\left\|\sum_{t=1}^{K}\mathcal{L}_{\boldsymbol{B}(t+KT)}(\boldsymbol{w}(t+KT))\right\|^2. \tag{68}$$

On the other hand, for any $t \in \{2, \cdots, K\}$, we have

$$\boldsymbol{w}(KT+t) - \boldsymbol{w}(KT+1) = \eta\sum_{s=1}^{t}\left(\sum_{\ell=1}^{KT+s}\beta^{KT+s-\ell}\nabla\mathcal{L}_{\boldsymbol{B}(\ell)}(\boldsymbol{w}(\ell))\right)$$

$$= \eta\sum_{s=1}^{t}\left(\sum_{\ell=KT+1}^{KT+s}\beta^{KT+s-\ell}\nabla\mathcal{L}_{\boldsymbol{B}(\ell)}(\boldsymbol{w}(\ell))\right) + \eta\sum_{s=1}^{t}\left(\sum_{\ell=1}^{KT}\beta^{KT+s-\ell}\nabla\mathcal{L}_{\boldsymbol{B}(\ell)}(\boldsymbol{w}(\ell))\right)$$

$$= \eta\sum_{\ell=1}^{t}\frac{1-\beta^{t-\ell+1}}{1-\beta}\nabla\mathcal{L}_{\boldsymbol{B}(KT+\ell)}(\boldsymbol{w}(KT+\ell)) + \eta\frac{\beta(1-\beta^t)}{1-\beta}\sum_{\ell=1}^{KT}\beta^{KT-\ell}\nabla\mathcal{L}_{\boldsymbol{B}(\ell)}(\boldsymbol{w}(\ell))$$

$$= \eta\sum_{\ell=1}^{t}\frac{1-\beta^{t-\ell+1}}{1-\beta}\nabla\mathcal{L}_{\boldsymbol{B}(KT+\ell)}(\boldsymbol{w}(KT+\ell)) - \eta\sum_{\ell=1}^{t}\frac{1-\beta^{t-\ell+1}}{1-\beta}\nabla\mathcal{L}_{\boldsymbol{B}(KT+\ell)}(\boldsymbol{w}(KT+1))$$

$$+ \eta\frac{\beta(1-\beta^t)}{1-\beta}\sum_{\ell=1}^{KT}\beta^{KT-\ell}\nabla\mathcal{L}_{\boldsymbol{B}(\ell)}(\boldsymbol{w}(\ell)) + \eta\sum_{\ell=1}^{t}\frac{1-\beta^{t-\ell+1}}{1-\beta}\nabla\mathcal{L}_{\boldsymbol{B}(KT+\ell)}(\boldsymbol{w}(KT+1)),$$

which by $\eta$ is small enough further indicates

$$\|\boldsymbol{w}(KT+t) - \boldsymbol{w}(KT+1)\|$$
$$\leq \eta\left\|\sum_{\ell=1}^{t}\frac{1-\beta^{t-\ell+1}}{1-\beta}\nabla\mathcal{L}_{\boldsymbol{B}(KT+\ell)}(\boldsymbol{w}(KT+\ell)) - \sum_{\ell=1}^{t}\frac{1-\beta^{t-\ell+1}}{1-\beta}\nabla\mathcal{L}_{\boldsymbol{B}(KT+\ell)}(\boldsymbol{w}(KT+1))\right\|$$

$$+ \eta\left\|\frac{\beta(1-\beta^t)}{1-\beta}\sum_{\ell=1}^{KT}\beta^{KT-\ell}\nabla\mathcal{L}_{\boldsymbol{B}(\ell)}(\boldsymbol{w}(\ell))\right\| + \eta\left\|\sum_{\ell=1}^{t}\frac{1-\beta^{t-\ell+1}}{1-\beta}\nabla\mathcal{L}_{\boldsymbol{B}(KT+\ell)}(\boldsymbol{w}(KT+1))\right\|$$

$$= \mathcal{O}(\eta)\sum_{\ell=2}^{t}\|\boldsymbol{w}(KT+\ell) - \boldsymbol{w}(KT+1)\| + \mathcal{O}(\eta)\left(\sum_{\ell=1}^{KT}\beta^{KT-\ell}\|\nabla\mathcal{L}_{\boldsymbol{B}(\ell)}(\boldsymbol{w}(\ell))\|\right)$$
$$+ \mathcal{O}(\eta)\|\nabla\mathcal{L}(\boldsymbol{w}(KT+1))\|.$$

Applying the same analysis to $\|\boldsymbol{w}(KT+t-1) - \boldsymbol{w}(KT+1)\|$ recursively, we finally obtain

$$\|\boldsymbol{w}(KT+t) - \boldsymbol{w}(KT+1)\|$$
$$\leq \mathcal{O}(\eta)\left(\sum_{\ell=1}^{KT}\beta^{KT-\ell}\|\nabla\mathcal{L}_{\boldsymbol{B}(\ell)}(\boldsymbol{w}(\ell))\|\right) + \mathcal{O}(\eta)\|\nabla\mathcal{L}(\boldsymbol{w}(KT+1))\|. \tag{69}$$

Applying Eq. (69) to the $\left\|\nabla\mathcal{L}_{\boldsymbol{B}(\ell)}(\boldsymbol{w}(\ell))\right\|$ in Eq. (69) ($\forall \ell \in [1, KT]$) iterative and choosing $\eta$ to be small enough, we further have

$$\|\boldsymbol{w}(KT + t) - \boldsymbol{w}(KT + 1)\|$$

$$\leq\mathcal{O}(\eta)\left(\sum_{\ell=0}^{T-1}\sqrt{\beta^{K(T-\ell)}}\left\|\nabla\mathcal{L}_{\boldsymbol{B}(K\ell+1)}(\boldsymbol{w}(K\ell+1))\right\|\right) + \mathcal{O}(\eta)\left\|\nabla\mathcal{L}(\boldsymbol{w}(KT+1))\right\|$$

$$=\mathcal{O}(\eta)\left(\sum_{\ell=0}^{T}\sqrt{\beta^{K(T-\ell)}}\left\|\nabla\mathcal{L}_{\boldsymbol{B}(K\ell+1)}(\boldsymbol{w}(K\ell+1))\right\|\right).$$

Therefore,

$$\sum_{t=1}^{K}\nabla\mathcal{L}_{\boldsymbol{B}(t+KT)}(\boldsymbol{w}(t+KT))$$

$$=\sum_{t=1}^{K}\nabla\mathcal{L}_{\boldsymbol{B}(t+KT)}(\boldsymbol{w}(t)) + \mathcal{O}\left(\eta\left(\sum_{\ell=0}^{T}\sqrt{\beta^{K(T-\ell)}}\left\|\nabla\mathcal{L}_{\boldsymbol{B}(K\ell+1)}(\boldsymbol{w}(K\ell+1))\right\|\right)\right)$$

$$=K\nabla\mathcal{L}(\boldsymbol{w}(t)) + \mathcal{O}\left(\eta\left(\sum_{\ell=0}^{T}\sqrt{\beta^{K(T-\ell)}}\left\|\nabla\mathcal{L}_{\boldsymbol{B}(K\ell+1)}(\boldsymbol{w}(K\ell+1))\right\|\right)\right). \qquad (70)$$

Similarly, one can obtain

$$\nabla\mathcal{L}(\boldsymbol{u}(KT+1))$$

$$=\nabla\mathcal{L}(\boldsymbol{w}(KT+1)) + \mathcal{O}\left(\|\boldsymbol{w}(KT+1) - \boldsymbol{w}(KT)\|\right)$$

$$=\nabla\mathcal{L}(\boldsymbol{w}(KT+1)) + \mathcal{O}\left(\eta\left(\sum_{\ell=0}^{T}\sqrt{\beta^{K(T-\ell)}}\left\|\nabla\mathcal{L}_{\boldsymbol{B}(K\ell+1)}(\boldsymbol{w}(K\ell+1))\right\|\right)\right). \qquad (71)$$

Applying Eq. (70) and Eq. (71) back to the Taylor Expansion (Eq. (68)), we have

$$\mathcal{L}(\boldsymbol{u}(KT+T+1))$$

$$\leq\mathcal{L}(\boldsymbol{u}(KT+1)) - \Omega(\eta)\left\langle\nabla\mathcal{L}(\boldsymbol{w}(KT+1)), \nabla\mathcal{L}(\boldsymbol{w}(KT+1))\right\rangle$$

$$+ \mathcal{O}\left(\eta^2\left(\sum_{\ell=0}^{T}\sqrt{\beta^{K(T-\ell)}}\left\|\nabla\mathcal{L}_{\boldsymbol{B}(K\ell+1)}(\boldsymbol{w}(K\ell+1))\right\|\right)^2\right)$$

$$\leq\mathcal{L}(\boldsymbol{u}(KT+1)) - \Omega(\eta)\left\langle\nabla\mathcal{L}(\boldsymbol{w}(KT+1)), \nabla\mathcal{L}(\boldsymbol{w}(KT+1))\right\rangle$$

$$+ \mathcal{O}\left(\eta^2\left(\sum_{\ell=0}^{T}\sqrt{\beta^{K(T-\ell)}}\left\|\nabla\mathcal{L}_{\boldsymbol{B}(K\ell+1)}(\boldsymbol{w}(K\ell+1))\right\|^2\right)\right).$$

Summing the above inequality over $T$ and setting $\eta$ small enough leads to the conclusion.

The proof is completed. $\qquad\square$

### F.3 Extension to the multi-class classification problem

As mentioned in Section 3, despite all the previous analyses are aimed at the binary classification problem, they can be naturally extended to the analyses multi-class classification problem. Specifically, in the linear multi-class classification problem, for any $(\boldsymbol{x}, \boldsymbol{y}) \in \mathbb{R}^{d_X} \times \{1, \cdots, C\}$ in the sample set $\boldsymbol{S}$, the (individual) logistic loss with parameter $\boldsymbol{W} \in \mathbb{R}^{C \times d_X}$ is denoted as

$$\ell(\boldsymbol{y}, \boldsymbol{W}\boldsymbol{x}) = \ln\frac{e^{\boldsymbol{W}_{\boldsymbol{y}}, \boldsymbol{x}}}{\sum_{i=1}^{C} e^{\boldsymbol{W}_{i}, \boldsymbol{x}}}.$$

Correspondingly, dataset $\boldsymbol{S}$ is separable if there exists a parameter $\boldsymbol{W}$, such that $\forall(\boldsymbol{x}, \boldsymbol{y}) \in \boldsymbol{S}$, we have $\boldsymbol{W}_{\boldsymbol{y}}, \boldsymbol{x} > \boldsymbol{W}_{i}, \boldsymbol{x}, \forall i \neq \boldsymbol{y}$. The multi-class $L^2$ max-margin problem is then defined as

$$\min\|\boldsymbol{W}\|_F, \ subject \ to: \ \boldsymbol{W}_{\boldsymbol{y}}, \boldsymbol{x} \geq \boldsymbol{W}_{i}, \boldsymbol{x} + 1, \forall(\boldsymbol{x}, \boldsymbol{y}) \in \boldsymbol{S}, i \neq \boldsymbol{y},$$

where $\|\cdot\|_F$ denotes the Frobenius norm. Denote $\hat{\boldsymbol{W}}$ as the $L^2$ max-margin solution, we have SGDM and Adam (w/s) still converges to the direction of $\hat{\boldsymbol{W}}$.

**Theorem 9.** *For linear multi-class classification problem using logistic loss and almost every separable data, with a small enough learning rate, and $1 > \beta_2 > \beta_1^4 \geq 0$ (for Adam (w/s)), SGDM and Adam (w/s) converge to the multi-class $L^2$ max-margin solution (a.s. for SGDM SGDM (w/. r)).*

Here we use several notations and lemmas from [34]. We define $\boldsymbol{w} = \text{vec}(\boldsymbol{W})$, $\hat{\boldsymbol{w}} = \text{vec}(\hat{\boldsymbol{W}})$, $\boldsymbol{e}_i \in \boldsymbol{R}^C$ ($i \in \{1, \cdots, C\}$) satisfying $(\boldsymbol{e}_i)_j = \delta_{ij}$, and $\boldsymbol{A}_i = \boldsymbol{e}_i \otimes \boldsymbol{I}_{d_X}$, where $\boldsymbol{I}_{d_X}$ is the identity matrix with dimension $d_X$. We still consider the normalized data, i.e., $\|\boldsymbol{x}\| \leq 1, \forall(\boldsymbol{x}, \boldsymbol{y}) \in \boldsymbol{S}$. Then, the individual loss of sample $(\boldsymbol{x}, \boldsymbol{y})$ can be then represented as

$$\ell(\boldsymbol{y}, \boldsymbol{W}\boldsymbol{x}) = \ln \frac{e^{\langle \boldsymbol{w}, \boldsymbol{A}_y \boldsymbol{x}\rangle}}{\sum_{i=1}^C e^{\langle \boldsymbol{w}, \boldsymbol{A}_i \boldsymbol{x}\rangle}}.$$

Furthermore, the gradient of training error at $\boldsymbol{W}$ has the form

$$\nabla\mathcal{L}(\boldsymbol{w}) = \frac{1}{N} \sum_{(\boldsymbol{x},\boldsymbol{y})\in\boldsymbol{S}} \sum_{i=1}^C \frac{1}{\sum_{j=1}^C e^{\langle \boldsymbol{w},(\boldsymbol{A}_j - \boldsymbol{A}_i)\boldsymbol{x}\rangle}} (\boldsymbol{A}_i - \boldsymbol{A}_y)\boldsymbol{x}.$$

and the Hessian matrix of $\mathcal{L}$ can be represented as

$$\mathcal{H}\mathcal{L}(\boldsymbol{w}) = \frac{1}{N} \sum_{(\boldsymbol{x},\boldsymbol{y})\in\boldsymbol{S}} \sum_{i=1}^C \frac{\sum_{j=1}^C e^{\langle \boldsymbol{w},(\boldsymbol{A}_j - \boldsymbol{A}_i)\boldsymbol{x}\rangle}}{\left(\sum_{j=1}^C e^{\langle \boldsymbol{w},(\boldsymbol{A}_j - \boldsymbol{A}_i)\boldsymbol{x}\rangle}\right)^2} (\boldsymbol{A}_i - \boldsymbol{A}_y)\boldsymbol{x}((\boldsymbol{A}_j - \boldsymbol{A}_i)\boldsymbol{x})^\top,$$

one can then easily verify all absolute value of the eigenvalues of $\mathcal{H}\mathcal{L}(\boldsymbol{w})$ is no larger than 2, which indicates $\mathcal{L}$ is 2-globally smooth.

On the other hand, the separable assumption leads to $\langle \hat{\boldsymbol{w}}, (\boldsymbol{A}_y - \boldsymbol{A}_i)\boldsymbol{x}\rangle > 0, \forall \boldsymbol{y} \neq i$, which further indicates

$$\langle \nabla\mathcal{L}(\boldsymbol{w}), \hat{\boldsymbol{w}}\rangle > 0.$$

Let $\gamma = \frac{1}{\|\hat{\boldsymbol{w}}\|}$, following the similar routine as the binary case, we have for a random subset of $\boldsymbol{S}$ sampled uniformly without replacement with size $b$, we have

$$\|\nabla\mathcal{L}(\boldsymbol{w})\|^2 \leq \mathbb{E}_{\boldsymbol{B}(t)}\|\nabla\mathcal{L}_{\boldsymbol{B}(t)}(\boldsymbol{w})\|^2 \leq \frac{2N}{\gamma b^2}\|\nabla\mathcal{L}(\boldsymbol{w})\|^2. \tag{72}$$

Similarly, we have for any positive real series $\{a_t\}_{t=t_1}^{t_2}$,

$$\gamma \sum_{t=t_1}^{t_2} a(t)\|\nabla\mathcal{L}(\boldsymbol{w}(t))\| \leq \left\|\sum_{t=t_1}^{t_2} a(t)\nabla\mathcal{L}(\boldsymbol{w}(t))\right\| \leq \sum_{t=t_1}^{t_2} a(t)\|\nabla\mathcal{L}(\boldsymbol{w}(t))\|. \tag{73}$$

The proofs of Stage I can then be obtained with Lyapunov functions unchanged and by replacing the corresponding lemmas using Eq. (72) and Eq. (73).

As for the proofs of Stage II, the Lyapunov functions are still the same, while we only need to prove the sum of $\langle \boldsymbol{r}(t), -\frac{\eta}{1-\beta}\nabla\mathcal{L}(\boldsymbol{w}(t)) - \ln\frac{t+1}{t}\hat{\boldsymbol{w}}\rangle$ (for GDM, $\langle \boldsymbol{r}(t), -\frac{\eta}{1-\beta}\nabla\mathcal{L}_{\boldsymbol{B}(t)}(\boldsymbol{w}(t)) - \frac{N}{bt}\sum_{i:\boldsymbol{x}_i\in\boldsymbol{B}(t)\cap\boldsymbol{S}_s} v_i\boldsymbol{x}_i\rangle$ for SGDM, $\langle \boldsymbol{r}(t), -\sqrt{\varepsilon}\ln\left(\frac{t+1}{t}\right)\hat{\boldsymbol{w}} - \frac{\eta}{1-\beta}\nabla\mathcal{L}(\boldsymbol{w}(t))\rangle$ for Adam). For the multi-class case using GDM, We present the following lemma from [34], while the other two cases can be proved similarly:

**Lemma 28** (Part of the proof of Lemma 20, [34]). *If $\langle \boldsymbol{w}(t), (\boldsymbol{A}_y - \boldsymbol{A}_i)\boldsymbol{x}\rangle \to \infty$ as $t \to \infty$, $\forall(\boldsymbol{x}, \boldsymbol{y}) \in \boldsymbol{S}$ and $\forall i \neq \boldsymbol{y}$, we have the sum of $\langle \boldsymbol{r}(t), -\frac{\eta}{1-\beta}\nabla\mathcal{L}(\boldsymbol{w}(t)) - \ln\frac{t+1}{t}\hat{\boldsymbol{w}}\rangle$ is upper bounded.*

The proof of Theorem 9 is then completed.

### F.4 Precisely characterize the convergence rate

Theorem 2 and Theorem 3 can be further extended to precisely characterize the asymptotic convergence rate for (S)GDM as the following theorem.

**Theorem 10.** *Let all the conditions in Theorem 2 (Theorem 3) hold. Assume the linear span of support vectors contains the whole dataset. Then, we have*

$$\lim_{t \to \infty} t\mathcal{L}(\boldsymbol{w}(t)) = C\frac{1}{\eta},$$

*where $C$ is a constant independent of learning rate $\eta$, momentum hyperparameter $\beta$, and mini-batch size $b$.*

The proof follows exactly the same routine as (Corollary 1, [23]) and we omit it here.

## G    Experiments

This section collects several experiments supporting our theoretical results.

### G.1    Experiments on linear model

In the following sections, we conduct experiments on the well=posed and ill-posed datasets in (Figures 1 and 3, [34]). Note that while we try our best to recover the result in [34] by using the same hyperparameters and the same dataset, there is still some difference between the result of [34] and us, as the code has not been open source and we are unable to know the initialization method and the random seed.

#### G.1.1    Comparing the training behavior of (S)GD, (S)GDM, deterministic Adam and RMSProp

The experiments in this section is designed to verify our theoretical results, i.e., Theorems 2, 3, 5, and 6. We use exact the same setting as (Figure 1, [34]) (including the same separable synthetic dataset and the same learning rate $1/\sigma_{max}^2$), and run (S)GD, (S)GDM, deterministic Adam and stochastic RMSProp over it different random seeds (for random samples despite the support sets $\{((1.5, 0.5), 1), ((0.5, 1.5), 1), ((-1.5, -0.5), -1), ((-1.5, -0.5), -1)\}$), and random mini-batches. We also report the results with a smaller learning rate $0.1/\sigma_{max}^2$. Both the angle between the output parameter and max-margin solution, the gap between current margin and max margin, parameter norm and the training loss are plotted in Figure 2. The observations can be summarized as follows:

- With proper learning rates, all of (S)GD, (S)GDM, deterministic Adam and stochastic RMSProp converge to the max-margin solution, which supports our theoretical results;

- (Similarity between (S)GD and (S)GDM). The asymptotic training behaviors of GD, SGD, GDM and SGDM are highly similar, which supports our Theorem 10.

- (Different behavior of Adaptive Optimizers). While deterministic Adam and stochastic RMSProp both converge to the max-margin solution, their training behaviors are different from those of (S)GD(M). Specifically, their angle gap from the max margin solution first increases then decreases. This phenomenon, however, can be explained by the proof of Theorems 5 and 6, the key insight of which is that when the training time is large enough, the gradient is small and the adaptive learning rate $1/\sqrt{\boldsymbol{\nu}(t) + \varepsilon \mathbb{1}_d}$ is dominated by $1/\varepsilon \mathbb{1}_d$, and thus Adam behaves like GDM and converges to the max margin solution.

#### G.1.2    Adam on ill-posed dataset

In (Figure 3, [34]), an ill-posed synthetic dataset is proposed to support the argument "Adam does not converge to max-margin solution". Specifically, such a dataset is derived by multiplying 20 to the second coordinate of all of the data in the dataset adopted in Section G.1.1, and thus both make the dataset (almost) degenerates to an one dimension line and lead to large singular value $\sigma_{max}$. However, their experiment does not contradict our implicit regularization result for Adam (i.e., Theorem 5), as our results hold asymptotically and it can be observed that at the end of their

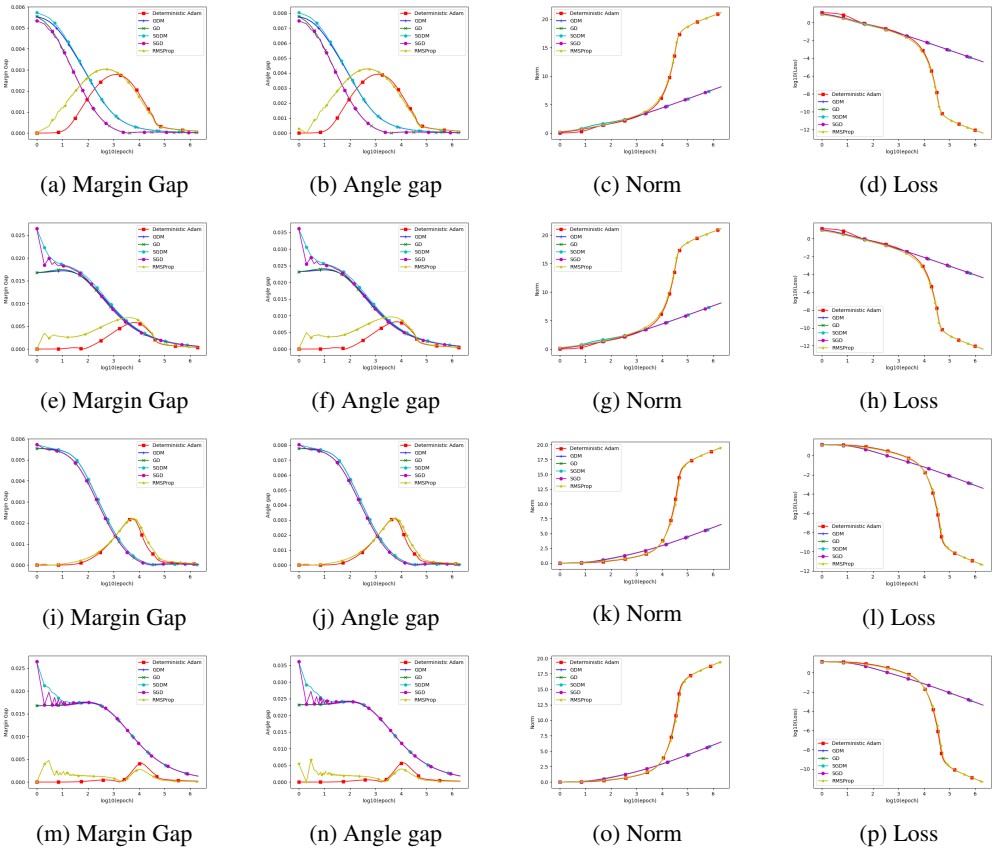

Figure 2: Comparison of (S)GD, (S)GDM, deterministic Adam, and stochastic RMSProp on the synthetic dataset in [34]. The first line use learning rate $1/\sigma_{max}^2$ and random seed 1. The second line use learning rate $1/\sigma_{max}^2$ and random seed 2. The third line use learning rate $0.1/\sigma_{max}^2$ and random seed 1. The forth line use learning rate $0.1/\sigma_{max}^2$ and random seed 2.

experiment, the angle gap is still decreasing. Furthermore, we reconduct the experiment of (Figure 3, [34]) in Figure 3. Specifically, we use the exact setting as [34] (the same ill-posed dataset, the same learning rate $1/\sigma_{max}^2$ both for GD and Adam, and $2 \times 10^6$ training iterations). It can be observed that the convergence of loss is rather slow at the first $10^4$ iteration. Also, the angle gap keeps decreasing at the end of experiment. We increase the training iterations to $2 \times 10^7$ in Figure 3 and find the angle gap is still decreasing. This stands with our result that Adam asymptotically converges to the max-margin solution.

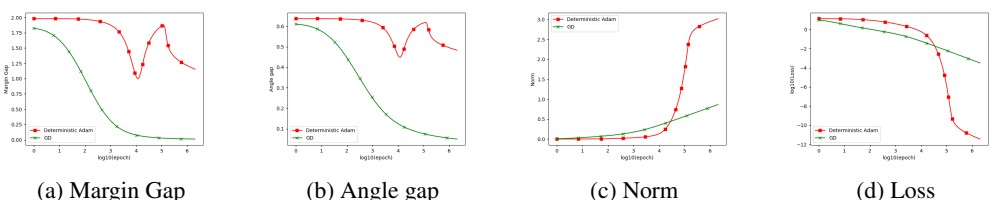

Figure 3: We reproduce the experiment in (Figure 3, [34]). It can be observed that the loss does not vary much until $10^4$ iteration.

## G.2  Evidence from deep neural networks

We conduct an experiment on the MNIST dataset using the four layer convolutional networks used in [21, 43] (first proposed by [22]) to verify whether SGD and SGDM still behave similarly in

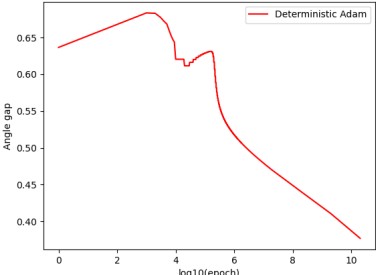

Figure 4: We extend the training time in Figure 2 and find that the angle gap keeps decreasing.

(homogeneous) deep neural networks. The learning rates of the optimizers are all set to be the default in Pytorch. The results can be seen in Figure 5. It can be observed that (1). SGDM achieves similar test accuracy compared to SGD while (2). SGDM converges faster than SGD.

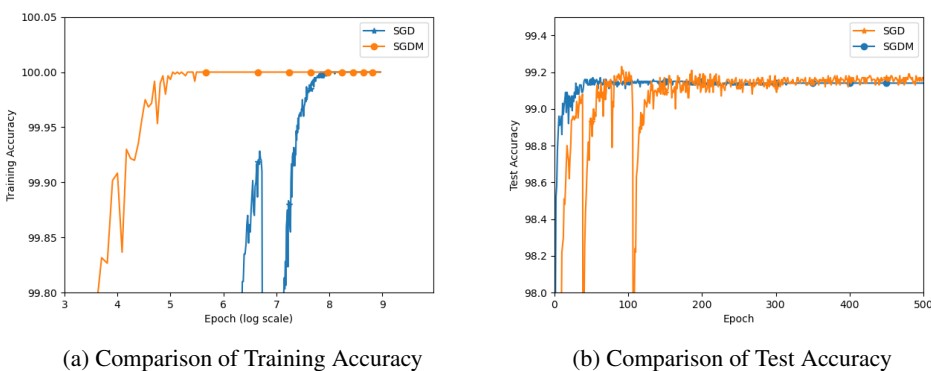

(a) Comparison of Training Accuracy
(b) Comparison of Test Accuracy

Figure 5: Comparison of SGD and SGDM on MNIST with a four-layer CNN.