# OpenReview forum: "Does Momentum Change the Implicit Regularization on Separable Data?"
_NeurIPS.cc/2022/Conference — NeurIPS 2022 Accept_

### Official Review · Reviewer_zpTJ · 2022-07-08

**Rating:** 6
**Confidence:** 4
**Soundness:** 3 good
**Presentation:** 3 good
**Contribution:** 2 fair

**Summary:**

This paper analyzes classification problems with linearly separable data and exponential-tailed loss.  The setting and analysis is inspired from the one in [1]. They start by reminding this setting in Section 3. In section 4, when running GDM on this problem, they show that the solution that is obtained is the L2 max-margin solution. They further detail the steps to get this result. In Section 5,  they show analogous results for SGDM and point out the differences with GDM in the analysis. In Section 6, they show that deterministic Adam and stochastic RMSProp converge to the L2 max margin solution as well. Finally, the authors run experiments on a "well-posed" dataset where they show that all the aforementioned algorithms converge to the l2 max margin solution.





[1] Soudry, Daniel, et al. "The implicit bias of gradient descent on separable data. (2018)

**Questions:**

I would like to raise more technical questions/concerns in this section.

1) I am concerned by the mix between discrete time and continuous time dynamics in the analysis in Section 4. As mentioned in the paper,  the loss for GDM does not decrease contrary to GD. This is why the authors propose Lemma 1. According to me, this Lemma only holds in the discrete dynamics because when eta ->0, the term (1-C1)*||w(t+1)-w(t)||^2/eta blows up. Do the authors agree with this? If this is the case, then I would like a clarification on how the authors can use the result "||\sum_{t=1}|w(t+1)-w(t)||^2 < \infty" in their proof in continuous time in line 212-213.  [1] does not have this issue with going back and forth between discrete time and continuous time dynamics since the loss is decreasing.

2) The experimental section seems a bit light. I think that it would be a good idea to reproduce the exact same plots as Figure 1 in [1] for GD, GDM and deterministic Adam in the case of ill-posed and well-posed datasets.

3) Overall, I think that the main concern I have is that we do not really see the difference between all these algorithms that are very different from each other. They indeed all converge to the L2 max margin solution but maybe we may see the benefit of using a 1st or 2nd order momentum in the convergence rate?



[1] Soudry, Daniel, et al. "The implicit bias of gradient descent on separable data. (2018)


**Limitations:**

The authors mentioned the limitations of their approach in conclusion. In particular, they underlined that the gap between their approach and the deep learning setting.

**Strengths And Weaknesses:**

I summarize as follows the strengths and weaknesses of this paper.

Strengths: I find the paper well written and easy to follow. I appreciated the fact that the proof sketch in Section 4 was easy to understand. I also believe that it is interesting to know that with momentum, the solution obtained is the max margin one.

Weaknesses: I have two major concerns regarding this paper. First, it is not focused on one single algorithm or phenomenon to understand. After reading the paper, the main message that pops in mind is  "several classical algorithms converge to the L2 max margin solution for linearly separable data". I think it would have been more beneficial to isolate a factor such as momentum and characterizing when it is useful.

Another main flaw I find in this paper is that it is incremental. On one hand, the authors do a good job at pointing out the differences between their analysis and the one in [1]. However, I think that the authors should do a better work at i) explaining intuitively with momentum it is not obvious to get the max margin solution ii) point out more clearly the convergence rate to the max margin solution (so that we see the difference with GD). I guess that a theorem like theorem 5 in [1] would be a good idea. Do you have a faster convergence rate when using momentum? The same remark I make above holds for deterministic Adam. I think there would be a lot of benefits to point out the difference between the algorithms.

I would like to raise the authors' attention on [2] that shows a separation between GD and GDM. In their setting, they consider logistic regression with linearly separable data and their learner model is a polynomial activation CNN. They show that GD still converges to the max margin solution but GDM does not converge to the max margin solution. In their setting, the max margin solution does not generalize while the one found by GDM generalizes. Their setting is different from the setting studied in this paper but I think it is good for the authors to be aware of this work.


[1] Soudry, Daniel, et al. "The implicit bias of gradient descent on separable data. (2018)

[2] Jelassi, Samy, and Yuanzhi Li. "Towards understanding how momentum improves generalization in deep learning." (2021).

---

> ### Author Response · Authors · 2022-08-02
> **(Part 1/2) Thanks for positive feedback and constructive comments. The concerns are dually addressed.**
>
> We thank the reviewer for the positive feedback and constructive comments. The concerns are dually addressed below.
>
> **Q1**: First, it is not focused on one single algorithm or phenomenon to understand. I think it would have been more beneficial to isolate a factor such as momentum and characterize when it is useful.
>
> **A1**: We thank the reviewer for the advice. The organization of this paper aims to gradually reveal the effect of momentum, momentum + stochastic sampling, and momentum + stochastic sampling + preconditioning on the implicit bias. The proof techniques for these algorithms are progressive instead of isolated. For example, the proof of GDM inspires the proof of SGDM.
>
> **Q2**: However, I think that the authors should do better work at i) explaining intuitively with the momentum it is not obvious to get the max-margin solution ii) pointing out more clearly the convergence rate to the max-margin solution (so that we see the difference with GD). I guess that a theorem like a theorem 5 in [Soudry et al. 2018] would be a good idea.
>
> **A2**: We thank the reviewer for the suggestion.
>
>  As for i), on the one hand, we will add the following discussion from Line 62 in the introduction section of the updated version to intuitively demonstrate the difficulty to get the implicit regularization of the momentum-based optimizers.
>
>  The technical difficulty of analyzing the implicit regularization of the momentum-based optimizers mainly comes from the historical information of training trajectory contained by the momentum term. Specifically, the foundation of the analysis is to derive the convergence of the loss. It is not trivial because, for momentum-based optimizers, the loss may not monotonously decrease due to the mismatch between the direction of the gradient and the momentum. Furthermore, even if the loss is proved to converge, the converging direction of the parameters is still hard to characterize as it’s difficult to track the training trajectory due to the momentum term.
>
> On the other hand, there are intuitive explanations for the difficulty in the proof sketches, and we invite the reviewer to refer to Line 191-194, Line 227-231, Line 293-300, and Appendix C.2.3 for details. The current available space is quite limited, and we will further include more intuitive explanations for the difficulty in the future version.
>
> As for ii), to begin with, our result “$w(t)-\ln(t)  \hat w$ is bounded” in Theorem 2 (GDM’s implicit regularization) is equivalent to the claim that  “$\Vert \frac{w(t)}{\Vert w(t)\Vert} - \frac{\hat w}{\Vert \hat w\Vert}
> \Vert  =O (\frac{1}{\log t})$” in theorem 5 in [1], and also indicates $L(w(t))=\Theta(\frac{1}{t})$. We will write this explicitly in the revised version. The same result holds for deterministic Adam. Therefore, there is no difference between GD, GDM, and Adam in terms of the convergence rate concerning $t$, while the coefficient hides in $O$ may be different. In Section 7 (Discussion), we further asymptotically investigate the hidden coefficient of GDM in the second paragraph, and we find it independent of the momentum coefficient $\beta$. It means that the asymptotic convergence rate is the same for GD and GDM, which is also observed in the experiment (e.g., Figure 1). We invite the reviewer to see the second paragraph of Section 7 for a detailed justification. The hidden coefficient of Adam is hard to derive due to the dynamic precondition. However, there is intuition from the proof that the convergence to the max-margin solution of Adam may be slower:
>  the proof relies on that when the gradient is small and the adaptive learning rate $\frac{1}{\sqrt{\nu(t)+\varepsilon}}$ is dominated by $\frac{1}{\sqrt{\varepsilon}}$, starting from which  Adam behaves like GDM and converges to the max-margin solution. As it may take lots of time for the gradient to be small, Adam may potentially converge to the max-margin solution slower. This can also be observed from the experiment: in Figure 1, the angle gap of Adam does not decrease until the middle of the training. We include the discussion in Section 7. However, it is hard to provide a precise characterization of Adam due to the dynamic learning rate $\frac{1}{\sqrt{\nu(t)+\varepsilon}}$, and we defer it to future work.

---

> > ### Comment · Reviewer_zpTJ · 2022-08-07
> > **Response post-rebuttal**
> >
> > I thank the reviewers for their response and for adding changes in their updated version of the paper. Even though I am still a bit concerned about the novelty of the contribution, I am satisfied by the response of the reviewers and increase my score by 1 point. My updated score is 6/10.

---

> ### Author Response · Authors · 2022-08-02
> **(Part 2/2) Thanks for positive feedback and constructive comments. The concerns are dually addressed.**
>
> **Q3**: I would like to raise the authors' attention on [Jelassi et al., 2021] which shows a separation between GD and GDM.
>
> **A3**: We thank the reviewer to raise this work and have added the following discussions to the revised version (please refer to Appendix A). [Jelassi et al., 2021] and [Zou et al., 2021] also investigate the generalization behavior of GDM and Adam but focus on a different setting from ours. Specifically, they work on a two-layer convolutional neural network with fixed and untrained second layer and cubic activation, while we work on the linear classifier. Also, except for linear separability, they further pose additional specific requirements for the dataset (e.g., all data shares a same (scaled) patch).  Based on the settings, [Jelassi et al. 2021] show that the output hypothesis by GDM provably generalizes better than GD, while [Zou et al. 2021] prove that the output hypothesis by GD generalizes better than Adam. These results, however, do not contradict our findings due to the difference in the settings. We believe both the works and this paper have their own merits to unveil the mystery of the generalization of momentum-based optimizers.
>
> **Q4**: Why the claim “$||\sum_{t=1}|w(t+1)-w(t)||^2 < \infty$” in the discrete case can be transferred to the continuous case?
>
> **A4**: Good question and thanks for asking. There is a skip in the proof here and we will make it up in the revised version. The property $\int \Vert \frac{dw(t)}{dt}\Vert^2 dt < \infty$ for the continuous case can be derived in the same manner as in the discrete case. Specifically, similar to the discrete case, we choose the potential function as $\xi(t)= L(w(t))+ \frac{\beta}{2(1-\beta)} \Vert  \frac{dw(t)}{dt}\Vert^2$. The derivative of $\xi(t)$ is
> $$\frac{d\xi(t)}{dt}= \langle \nabla L(w(t)), \frac{dw(t)}{dt}\rangle +\frac{\beta}{1-\beta} \langle  \frac{d^2w(t)}{dt^2},  \frac{dw(t)}{dt}\rangle= -\Vert  \frac{dw(t)}{dt}\Vert^2,$$
> where the second equality is due to the definition of the continuous dynamic.
>  Therefore, $\xi(0)-\xi(T)=  \int_0^T \Vert  \frac{dw(t)}{dt}\Vert^2 dt$. Then, taking $T\rightarrow \infty$ yields the conclusion. **On the other hand, the proof based on the continuous dynamics in the main text is for a simple demonstration, while in the proof of the appendix we directly use the discrete updates**. We invite the reviewer to read the proof from Line 717 for details if interested.
>
> **Q5**: The experimental section seems a bit light. I think that it would be a good idea to reproduce the same plots as Figure 1 in [Soudry et al., 2018] for GD, GDM, and deterministic Adam in the case of ill-posed and well-posed datasets.
>
> **A5**: Thanks for the advice. We will add the plot for norm dynamics and loss dynamics in the revised paper just as in Figure 1 and Figure 3 in [Soudry et al., 2018] (ill-posed and well-posed datasets). We try our best to recover the exact results. However, the plots are still a bit different from [Soudry et al., 2018]. This is due to the codes of the experiments on the synthetic datasets in [29] has not been open source, we are unable to know the exact coordinates of the generated random data and adopted initialization, and thus unable to reproduce the results exactly.
>
> **Q6**: Overall, I think that the main concern I have is that we do not really see the difference between all these algorithms that are very different from each other. They indeed all converge to the L2 max margin solution but maybe we may see the benefit of using a 1st or 2nd order momentum in the convergence rate.
>
> **A6**: Please refer to the answer for ii) in **A2**.
>
> **References**
>
> Soudry et al. “The Implicit Bias of Gradient Descent on Separable Data.” (2018).
>
> Jelassi et al. "Towards understanding how momentum improves generalization in deep learning." (2021).
>
> Zou et al. "Understanding the generalization of adam in learning neural networks with proper regularization." (2021).

---

### Official Review · Reviewer_wavj · 2022-07-09

**Rating:** 7
**Confidence:** 3
**Soundness:** 3 good
**Presentation:** 3 good
**Contribution:** 3 good

**Summary:**

The author(s) studied the implicit regularization property of gradient descent method with momentum and its adaptive version. This work is theoretical in essence. It was shown that GDM, SGDM and deterministic Adam all converge to the max-margin solution.

**Questions:**

- The third paragraph of Section 3.4 of [1] states that they observe ADAM does not converges to the max-margin solution. Does the results established in this work contradict with their observation?
- Please see my questions for line 208-213 in the "strengths and weakness" section.

**Limitations:**

This paper has no potential negative societal impact.

**Strengths And Weaknesses:**

Overall, I think this paper makes a descent contribution on understanding the implicit regularization property of GDM and its variants, which is a important research topic in machine learning. I tried to check the proof for Theorem 2 and I think it is correct (though I think the proof sketch stated in the main context should be improved). I recommend acceptance if all other reviewers confirm that the proof is correct.

Some concern:
- Line 208-213 is a little bit confusing. In particular, the first bullet-point states that the analysis of [29] can be directly borrowed. However, the definition of $w$ is different from [29] since momentum is being used. Therefore a more careful explanation should be given. The third bullet-point is also confusing, I understand that $\int \| \frac{d w(s)}{ds} \|^2 ds$ is bounded, but $\int \| \frac{d w(s)}{ds} \| ds$ may not be bounded, so why is $\int \langle \hat{w},\frac{d w(s)}{ds} \rangle ds$ bounded?
- Line 214, it seems that it was shown that $\| r(T) \|^2$ is bounded by $O(1) +o( \| r(T) \| )$ instead of $\| r(T) \|^2 + o(\|r(T)\|)$ bounded by some constant.

Minor:
- $L_2$ and $L^2$ are used interchangeably, the term $L_2$ is used in abstract and the term $L^2$ is used in the main context.
- Line 14, Numerical experiments are conducted and support -> Numerical experiments are conducted to support …
- Line 57, parameter -> parameters
- Line 63, first order -> first-order
- Line 95, Interesting -> interested
- Line 195, better to mention that $\inf \xi(t) > -\infty$

---

> ### Author Response · Authors · 2022-08-02
> **Thanks for positive feedback and constructive comments. The concerns are dually addressed.**
>
> We thank the reviewer for your positive feedback and constructive comments. The typos have been revised accordingly and marked red in the updated manuscript. The concerns are dually addressed below.
>
> **Q1**: Technical concerns:
>
> **Q1.1**: The first bullet-point states that the analysis of [Soudry et al. 2018] can be directly borrowed. However, the trajectory of GDM is different from GD and can be different. Why the analysis can be borrowed?
>
> **A1.1**: The analysis of [Soudry et al. 2018] to bound the term $\int_1^T \langle r(s), -\eta\nabla L (w(s)) -\frac{1}{s} \hat w\rangle ds$ can be generalized, as its proof only requires (1) the loss to be exponentially tailed, (2) the data to be separable, and (3) the gradient to converge to 0. In our analysis of GDM, we also need to check the three requirements. The first two requirements are satisfied by our assumptions (i.e., Assumptions 1 and 2), and the third is ensured by the discussion in Line 196-197 in Stage I. In a word, the aforementioned analysis of [Soudry et al. 2018]  does not rely on the specific training trajectory of $w(t)$ and thus can be borrowed. Also, in the appendix (starting from Line 730), we provide the complete proof of this argument for the sack of completeness.
>
> **Q1.2**: I understand that $\int \Vert \frac{dw(s)}{ds} \Vert^2 ds$ is bounded, but why is $\int \langle \hat w, \frac{dw (s)}{ds}\rangle ds$ bounded?
>
> **A1.2**: There seems to be a misunderstanding. What we try to show is that $\int \frac{1}{s}\langle \hat w, \frac{dw(s)}{ds} \rangle ds$ (an $\frac{1}{s}$ term is missing) is bounded.
>
> **How this term comes**: Originally, we want to bound $\int \langle  \frac{dr(s)}{ds}, \frac{dw(s)}{ds} \rangle ds$.  As $ \frac{dr(s)}{ds} = \frac{dw(s)}{ds}-\frac{1}{s} \hat w$ and  $\int \Vert  \frac{dw(s)}{ds} \Vert^2 ds$ is bounded, we then need to show that $\int \frac{1}{s}\langle \hat w, \frac{dw(s)}{ds} \rangle ds$ is bounded.
>
> **How we handle this term**: As $\int_1^{\infty} \frac{1}{s^2} ds$ is bounded, the argument then directly follows as
> $$\vert \int \langle \frac{1}{s} \hat w,
> \frac{dw(s)}{ds}\rangle ds\vert \le \int \vert \langle \frac{1}{s}\hat w,  \frac{dw(s)}{ds}\rangle\vert ds\le \frac{1}{2} \int  \frac{1}{s^2} \Vert \hat w\Vert^2  ds + \frac{1}{2}\int \Vert \frac{dw(s)}{ds} \Vert^2 ds,$$
> where the second inequality is due to the Mean-value Inequality.
>
> **Q1.3**: In line 214, it seems that it was shown that $\Vert r(T)\Vert^2$  is bounded by $O(1)+ o(\Vert r(T)\Vert)$
> instead of $\Vert r(T)\Vert^2+ o(\Vert r(T)\Vert)$  bounded by some constant.
>
> **A1.3**: These two inequalities are equivalent: the first one is $$\Vert r(T)\Vert^2\le O(1)+ o(\Vert r(T)\Vert).$$ Moving $o(\Vert r(T)\Vert)$ to the left-hand side leads to $$\Vert r(T)\Vert^2+ o(\Vert r(T)\Vert) \le O(1).$$ We guess that the confusion of the reviewer may come from the sign of O and o. In this paper, we use $O$ and $o$ to compare the absolute value of variables. Therefore, $o(\Vert r(T)\Vert) = - o(\Vert r(T)\Vert)$.
>
> **Q2**: The third paragraph of Section 3.4 of [1] states that they observe that ADAM does not converge to the max-margin solution. Do the results established in this work contradict their observation?
>
> **A2**: Thanks for asking. When the reviewer says “[1]”, do you mean the paper “The Implicit Bias of Gradient Descent on Separable Data” by Soudry et al. ? Because [1] in the references of this paper is “The Implicit Regularization of Stochastic Gradient Flow for Least Squares” by Ali et al., where we do not find a counter-example for Adam.
>
> As for the counter-example in [Soudry et al., 2018]: the counter-example is noticed and discussed in Remark 7. We summarize the idea here.
> First of all, this result does not contradict our result for the implicit regularization of Adam, as we study the asymptotic behavior. In the counterexample, it can be observed that the angle gap keeps decreasing after $10^5$ iteration, which stands with our result. In Figure 2 and 3 of our paper, we reproduce the experiment and find that the angle gap keeps decreasing after extending the training time. Secondly, this synthetic dataset is ill-posed, which has large singular values and makes the training rather slow, and it can be observed that the training loss of Adam does not vary much until $10^4$ iteration. On the well-posed dataset adopted by (Figure 1, [Soudry et al., 2018]), we observe that Adam will converge to the max-margin solution rapidly.
>
> **References**
>
> Soudry et al. “The Implicit Bias of Gradient Descent on Separable Data.” (2018).

---

> > ### Comment · Reviewer_wavj · 2022-08-07
> > **Thanks for your response**
> >
> > Thanks for your detailed reponse, all my questions are addressed, I would like to keep my score unchanged.

---

### Official Review · Reviewer_wZ1t · 2022-07-11

**Rating:** 6
**Confidence:** 4
**Soundness:** 3 good
**Presentation:** 3 good
**Contribution:** 3 good

**Summary:**

This paper studies the implicit bias of gradient descent with momentum. The authors show that gradient descent with momentum, stochastic gradient descent with momentum, and deterministic Adam all converge to the L2 max-margin solution.

**Questions:**

There are a number of works studying the generalization error of gradient descent with momentum / Adam:

Jelassi, Samy, and Yuanzhi Li. "Towards understanding how momentum improves generalization in deep learning." (2021).

Zou, Difan, Yuan Cao, Yuanzhi Li, and Quanquan Gu. "Understanding the generalization of adam in learning neural networks with proper regularization." arXiv preprint arXiv:2108.11371 (2021).

It would be very helpful if the authors can discuss the relation between the implicit bias results with these previous works.


**Limitations:**

This paper does not have any potential negative societal impact.

**Strengths And Weaknesses:**

Strengths:

The results of the paper are solid and interesting.

The presentation of the paper is clear and easy to follow.

The authors give a fairly comprehensive analysis of implicit bias in different settings.

Weaknesses:

Although the results in this paper seem valid, they, to some extent, contradict the empirical observation that Adam and SGD usually have different generalization errors, as is discussed in [39]. This is worth some more discussion and investigation.

The experiments are mostly deferred to the appendix, and Figure 1 in the main part of the paper does not seem very convincing.

---

> ### Author Response · Authors · 2022-08-02
> **Thanks for positive feedback and constructive comments. The concerns are dually addressed.**
>
> We thank the reviewer for the positive feedback and constructive comments. The concerns are dually addressed below, and the paper is revised accordingly.
>
> **Q1**: In the existing literature, it is often observed that Adam and SGD usually have different generalization errors, as is discussed in [Wilson et al. 2017]. How to explain this mismatch?
>
> **A1**: Good question.  The mismatch is due to three reasons. Firstly, the gap still exists between assumptions in theory and practice in modern deep learning tasks, e.g., many advanced techniques are applied (e.g. batch normalization, weight decay, dropout, etc.), which are not covered by the existing theory, and these components may change the implicit regularization of optimizers (to varying degrees depending on the optimizer). Understanding the implicit regularization of optimizers with these techniques involved is interesting while surely challenging, and we aim to work on it in future works.
>
> Secondly, implicit regularization can change with the task. Specifically, [Wilson et al. 2017] consider a specific classification problem with least-square loss that has finite-root local minima. While they prove in such a case GD provably generalizes better than adaptive optimizers, their conclusion does not contradict ours, as we consider the classification problem with exponential-tailed loss (e.g., logistic loss).
>
> Thirdly,  although the theoretical results focus on the asymptotic direction of parameters, as shown in Figure 1. (b), the optimization trajectories (e.g., parameter directions) of deterministic Adam and stochastic RMSProp follow a different manner from those of (S)GD(M): the angle gaps do not decrease until the middle of the training, while those of (S)GD(M) keeps decreasing. This can also be explained by the proof of Adam’s implicit regularization, that is,  when the training time is large enough, the gradient is small and the adaptive learning rate $\frac{1}{\sqrt{\nu(t)+\varepsilon}}$ is dominated by $\frac{1}{\sqrt{\varepsilon}}$,  and then  Adam behaves like GDM. Therefore, early stopping may prevent the gradient from being too small, and it makes Adam’s optimization trajectory different from GDM's, and it may bring a worse generalization (in terms of margin). Such a discussion is included in Section 7.
>
>
> **Q2**: The experiments are mostly deferred to the appendix, and Figure 1 in the main part of the paper does not seem very convincing.
>
> **A2**: Thanks for the advice.  Figure 1 is based on the same setting of (Figure 1, [Soudry et al. 2018]), and we provide detailed experiment observations with varying random seeds and hyperparameter settings in Appendix G.  We defer most experiments to the appendix due to the page limit and will put some of them (e.g., Figure 2 and Figure 5) back to the main text in the camera-ready version if accepted.
>
> **Q3**: There are several works studying the generalization error of gradient descent with momentum / Adam. It would be very helpful if the authors can discuss the relation between the implicit bias results with these previous works.
>
> **A3**: We thank the reviewer for raising these two papers, and we have added the following discussions to the revised version (please refer to Appendix A). [Jelassi et al., 2021] and [Zou et al., 2021] also investigate the generalization behavior of GDM and Adam but focus on a different setting from ours. Specifically, they work on a two-layer convolutional neural network with fixed and untrained second layer and cubic activation, while we work on the linear classifier. Also, except for linear separability, they further pose additional specific requirements for the dataset (e.g., all data shares some same (scaled) patch).  Based on the settings, [Jelassi et al. 2021] show that the output hypothesis by GDM provably generalizes better than GD, while [Zou et al. 2021] prove that the output hypothesis by GD generalizes better than Adam. These results, however, do not contradict our findings due to the difference in the settings. We believe both the works and this paper have their own merits to unveil the mystery of the generalization of momentum-based optimizers.
>
> **References**
>
> Wilson et al. "The Marginal Value of Adaptive Gradient Methods in Machine Learning." (2017)
>
>
> Soudry et al. “The Implicit Bias of Gradient Descent on Separable Data.” (2018).
>
> Jelassi et al. "Towards understanding how momentum improves generalization in deep learning." (2021).
>
> Zou et al. "Understanding the generalization of adam in learning neural networks with proper regularization." (2021).

---

### Official Review · Reviewer_HjYT · 2022-07-12

**Rating:** 6
**Confidence:** 4
**Soundness:** 3 good
**Presentation:** 3 good
**Contribution:** 3 good

**Summary:**

This paper studies the implicit regularization of momentum based optimization algorithms on separable data. In particular, this paper proves that gradient descent with momentum, SGD with momentum, and full-batch Adam can both lead to the max-margin solution, which completes the prior results on the implicit regularization of GD and SGD (without momentum).

**Questions:**

1. Although the momentum will not affect the final converging point, how does it affect the optimization trajectory in the early stages? Will it bring some accelerations in terms of convergence, at least for linear classification problems?

2. What happens if we consider early stopping?

3. It would be better to discuss more regarding the nonconvex model. In practice, it is more likely that different optimization algorithms will admit different implicit regularization (or optimization trajectory) in early stages so that they will finally find different local minima, and the quality of the local minima (e.g., sharpness) will determine the generalization ability. Extending to deep homogenous models is definitely a good working direction, but it could be more interesting and important to focus on more general DNN models that the "local property" of the found solutions is more important.

**Limitations:**

Please refer to the weakness part.

**Strengths And Weaknesses:**

Overall, the strengths of this paper are clearly stated in the paper:

1. This paper first proves the implicit regularization of momentum-based optimizers.
2. A new proof technique is developed for studying the algorithmic behavior when momentum is involved, which is of independent interest.

The weaknesses are as follows:

1. This paper does not clearly discuss the difference between GD and GDM (or SGDM). In particular, the authors only prove the implicit regularization/final converging point of GDM, which is the same as that of GD. However, the role that momentum plays in the training process is still not clear. For instance, can you show that the momentum can accelerate the training process (at least in stage 1)?

2. In practice, the training algorithms are typically performed with early stopping, thus the output of the optimizer will not be exactly (or even not close to) the max-margin solution. Can you elaborate on this point and provide some brief discussion regarding the difference between GD, GDM, and ADAM. Perhaps early stopping is one aspect of the empirical generalization differences between these optimizers.

---

> ### Author Response · Authors · 2022-08-02
> **Thanks for positive feedback and constructive comments. The concerns are dually addressed.**
>
> We thank the reviewer for positive feedback and constructive comments. The concerns are dually addressed below.
>
> **Q1**: Can momentum accelerate the training process in this case?
>
> **A1**:  We have discussed the convergence rate of (S)GDM in the second paragraph of Section 7. We provide precise derivation and characterization of (S)GDM’s convergence rate in Appendix E.4.  The results show that the convergence rates for (S)GD and (S)GDM given the same learning rate are in the same order (which also matches the results in the literature  ([Sun et al. 2019], [Yan et al. 2018], etc.)) and are exactly the same asymptotically. This is also verified by our experiments (e.g. Figure 1), where the training curve of (S)GD aligns with (S)GDM well. Furthermore, to the best of our knowledge, there still lacks theoretical justification for the acceleration effect of the momentum except in the strongly convex case (note that the exponential-tailed loss considered in this paper is not strongly convex).   As this work mainly focuses on the linear model, whether momentum can accelerate the training of the non-linear model is an exciting future direction and deserves further investigation. We also invite the reviewer to the second paragraph of Section 7 for a complete discussion.
>
> **Q2**: What happens if we consider early stopping?
>
> **A2**: Good question and thanks for asking. As illustrated by the previous answer, the optimization trajectories of (S)GD and (S)GDM, in this case, are pretty similar. However, as shown in Figure 1. (b), deterministic Adam and stochastic RMSProp have different optimization trajectories with (S)GDM: the angle gaps do not decrease until the middle of the training, while those of (S)GD(M) keep decreasing. Therefore, early stopping may influence the implicit regularization of Adam and RMSProp.   This can also be explained by the proof of Adam’s implicit regularization: when the training time is large enough, the gradient is small and the adaptive learning rate $\frac{1}{\sqrt{\nu(t)+\varepsilon}}$ is dominated by $\frac{1}{\sqrt{\varepsilon}}$, and then Adam behaves like GDM and thus converges to the max-margin solution. Therefore, early stopping may prevent the gradient from being too small, which makes Adam’s optimization trajectory different from GDM and may bring a worse generalization (in terms of margin). We will include such a discussion in Section 7.
>
> **Q3**: It would be better to discuss more regarding the nonconvex model.
>
> **A3**: Thanks for the suggestion!   It is an important research topic to find the implicit regularization of cases where finite-root local minima exist (e.g. exponential-tailed loss with $L^2$ regularization) and early stopping is applied. In such a circumstance, the quality of the local minima (e.g., sharpness) will come into play and may require additional analysis techniques. We will investigate it in future work.
>
> **References**
>
> Sun et al. "Non-ergodic convergence analysis of heavy-ball algorithms" (2019).
>
> Yan et al. "A unified analysis of stochastic momentum methods for deep learning" (2018).

---

> > ### Comment · Reviewer_HjYT · 2022-08-09
> > **Thanks for the response**
> >
> > I am generally satisfied with the authors' comments and explanations regarding my questions. I would like to keep my evaluation as "weak accept".

---

### Meta-Review · Area_Chair_M87B · 2022-08-27

**Recommendation:** Accept
**Confidence:** Certain

**Metareview:**

The reviewers agree that the paper provides a nice technical contribution to understanding the effects of momentum on the generalization error of optimization methods. The results of the paper are solid and interesting, while the presentation is clear and easy to follow.

**Award:**

No

---

### Decision · Program_Chairs · 2022-09-14

Accept